# Fast Asymptotically Optimal Algorithms for Non-Parametric Stochastic Bandits

**Dorian Baudry**
Ecole Polytechnique, CREST
Palaiseau, France
dorian.baudry@ensae.fr

**Fabien Pesquerel**
Univ. Lille, Inria, CNRS, Centrale Lille,
UMR 9189-CRIStAL, F-59000 Lille, France
fabien.pesquerel@inria.fr

**Rémy Degenne**
Univ. Lille, Inria, CNRS, Centrale Lille,
UMR 9189-CRIStAL, F-59000 Lille, France
remy.degenne@inria.fr

**Odalric-Ambrym Maillard**
Univ. Lille, Inria, CNRS, Centrale Lille,
UMR 9189-CRIStAL, F-59000 Lille, France
odalric.maillard@inria.fr

## Abstract

We consider the problem of regret minimization in non-parametric stochastic bandits. When the rewards are known to be bounded from above, there exists asymptotically optimal algorithms, with asymptotic regret depending on an infimum of Kullback-Leibler divergences (KL). These algorithms are computationally expensive and require storing all past rewards, thus simpler but non-optimal algorithms are often used instead. We introduce several methods to approximate the infimum KL which reduce drastically the computational and memory costs of existing optimal algorithms, while keeping their regret guaranties. We apply our findings to design new variants of the `MED` and `IMED` algorithms, and demonstrate their interest with extensive numerical simulations.

## 1 Introduction

A Multi-Armed Bandit (MAB) is a sequential decision-making problem where at each time step $t \in \mathbb{N}$ a learner collects a reward from an arm $A_t$ chosen among $K \in \mathbb{N}$ alternatives. We consider the stochastic case, in which all rewards collected from an arm $k \in [K]$ are i.i.d. and drawn from a fixed distribution $F_k$, of expectation $\mu_k$. For a time horizon $T$ we define the number of pulls of an arm $k$ by $N_k(T) = \sum_{t=1}^{T} \mathbb{I}(A_t = k)$, and their sub-optimality gap by $\Delta_k = \mu^\star - \mu_k$, with $\mu^\star = \max_{k \in [K]} \mu_k$. The goal of the learner is to maximize their expected sum of rewards, or equivalently to minimize the *regret*, defined as

$$\mathcal{R}_T = \mathbb{E}\left[\sum_{t=1}^{T}(\mu^\star - \mu_{A_t})\right] = \sum_{k=1}^{K} \Delta_k \mathbb{E}\left[N_k(T)\right] \ . \tag{1}$$

To achieve this goal, the learner can leverage the rewards collected up to time $t$ and their knowledge on the family of distributions $\mathcal{F}$ to which $F_1, \ldots, F_K$ belong. The definition of $\mathcal{F}$ determines the complexity of the bandit problem, since any uniformly efficient algorithm[1] satisfies the lower bound [Lai and Robbins, 1985, Burnetas and Katehakis, 1996]: for all $k \in [K]$ with $\Delta_k > 0$,

$$\liminf_{T \to \infty} \frac{\mathbb{E}[N_k(T)]}{\log(T)} \geqslant \frac{1}{\mathcal{K}_{\inf}^{\mathcal{F}}(F_k, \mu^\star)} \ , \quad \mathcal{K}_{\inf}^{\mathcal{F}}(F_k, \mu^\star) = \inf_{G \in \mathcal{F}}\left\{\text{KL}(F_k, G) : \mathbb{E}_G[X] > \mu^\star\right\} \ , \tag{2}$$

---

[1]$\forall(F_1, \ldots, F_K) \in \mathcal{F}^K : \forall \alpha > 0, \mathbb{E}[N_k(T)] = o(T^\alpha)$ for all $k$ satisfying $\Delta_k > 0$.

37th Conference on Neural Information Processing Systems (NeurIPS 2023).

where KL$(.,.)$ denotes the Kullback-Leibler divergence between two distributions. We call an algorithm *asymptotically optimal* (sometimes shortened to optimal) if its regret upper bound matches this lower bound. That is, if for all $k$, $\limsup_{T\to\infty} \mathbb{E}[N_k(T)]/\log(T) \leq 1/\mathcal{K}_{\inf}^{\mathcal{F}}(F_k, \mu^\star)$.

For some parametric families (e.g. Gaussian), $\mathcal{K}_{\inf}^{\mathcal{F}}$ has a convenient closed-form expression. However, for more general non-parametric families of distributions this may not be the case. In this work we consider the most famous example of such model, where the learner only knows that the distributions are bounded. Given a range $[b, B]$, we formally define

$$\mathcal{F}_{[b,B]} = \{F \in \mathcal{P}(\mathbb{R}) : \ \mathrm{supp}(F) \subset [b, B] \subset \mathbb{R}\}, \tag{3}$$

but in the rest of the paper we keep the notation $\mathcal{F}$ (for $\mathcal{F}_{[b,B]}$) for simplicity. We always assume that $B$ is known, but in some cases knowing that $b$ is finite will be sufficient. By analyzing a dual optimization problem derived from (2), Honda and Takemura [2010] obtained that

$$\forall F \in \mathcal{F}, \ \mu \leqslant B : \quad \mathcal{K}_{\inf}^{\mathcal{F}}(F, \mu) = \max_{\lambda \in \left[0, \frac{1}{B-\mu}\right]} \mathbb{E}_{X \sim F}\left[\log\left(1 - \lambda\left(X - \mu\right)\right)\right]. \tag{4}$$

**Optimal algorithms for bounded distributions**   The literature on MABs has become diverse in the past years, so in the following we present a non-exhaustive selection of works that focus on asymptotically optimal algorithms for (bounded) stochastic bandits. We refer the reader to [Lattimore and Szepesvári, 2020, Bubeck et al., 2012] for broader surveys.

The most famous optimal algorithm is certainly KL-UCB [Cappé et al., 2013, Agrawal et al., 2021]. Based on the principle of optimism in face of uncertainty [Auer et al., 2002], it uses confidence intervals on empirical $\mathcal{K}_{\inf}^{\mathcal{F}}$ to choose the arm with the largest plausible mean. The optimality of KL-UCB was only proved recently [Agrawal et al., 2021], as the seminal work of Cappé et al. [2013] only proved it for Multinomial distributions. A second family of algorithms, *Minimized Empirical Divergence* (MED) [Honda and Takemura, 2010, 2011, 2015, Baudry et al., 2023], aims at exploiting the lower bound (2): these algorithms are based on the computation of $\mathcal{K}_{\inf}^{\mathcal{F}}$ for the empirical distributions and the current best empirical mean, and hence differ from KL-UCB (they perform only one computation per arm/step). More recently, an optimal algorithm based on Thompson Sampling (TS) [Thompson, 1933] was proposed for general bounded distributions, under the name of *Non-Parametric Thompson Sampling* (NPTS) [Riou and Honda, 2020]. In this work, the authors generalize the TS algorithms for Bernoulli distributions [Agrawal and Goyal, 2012, Kaufmann et al., 2012] by using an improper Dirac prior and a Dirichlet posterior.

Throughout the paper, we use "MED algorithms" (or simply MED) to refer to the family of algorithms. We denote by MED the randomized algorithm presented in [Honda and Takemura, 2010], that we detail in Algorithm 3, and IMED its deterministic counterpart ([Honda and Takemura, 2015], Algorithm 2).

**Cost/performance trade-off**   In practice, the choice of a bandit algorithm may be motivated by its theoretical guarantees, but also by its computation and memory costs. Unfortunately, optimal algorithms are quite costly: MED and KL-UCB need to compute an empirical $\mathcal{K}_{\inf}^{\mathcal{F}}$ (costing $\mathcal{O}(n\log(n))$ for $n$ samples, solving (4) with precision $1/n$ for each arm/round, KL-UCB being more costly (several $\mathcal{K}_{\inf}^{\mathcal{F}}$ per step), while the cost of sampling in NPTS is linear in the number of samples of each arm. For that reason, cheaper sub-optimal alternatives are often considered in place of optimal algorithms. For instance, UCB [Auer et al., 2002] has constant run time and memory, and achieves a logarithmic regret with multiplicative constant $\mathcal{O}\left(\sum_{k:\Delta_k>0} \Delta_k^{-1}\right)$. More generally, all the algorithms designed for $1/4$-sub-gaussian distributions can be used on $\mathcal{F}$ if the rewards are rescaled in $[0, 1]$. A finer approximation consists in using the KL divergence of Bernoulli distributions, that lower bounds $\mathcal{K}_{\inf}^{\mathcal{F}}$ [Cappé et al., 2013]. However, it is important to note that these approximations are sensitive to the value (and knowledge) of the lower bound of the support $b$ (as a rescaling in $[0, 1]$ is necessary), contrarily to asymptotically optimal algorithms. Pushing further this idea, a procedure proposed by Riou and Honda [2020], inspired by Agrawal and Goyal [2012], consists in *discretizing* the rewards: a reward $X \in [b, B]$ is transformed in $Y \in \mathcal{X} \coloneqq \{x_1 = b, \ldots, x_M = B\}$ for some finite grid $\mathcal{X}$, such that $\mathbb{E}[Y|X] = X$. Hence, the expectations of the arms are unchanged while the memory is reduced to $\mathcal{O}(MK)$ and the computation time of $\mathcal{K}_{\inf}^{\mathcal{F}}$ for discretized distributions is proportional to $M$. Unfortunately, even if all these techniques lead to logarithmic regret, the multiplicative constant before $\log(T)$ can be arbitrarily large compared with the optimal one. More precisely, we show in Lemma 6 that for a small gap $\Delta$ the ratio $\mathcal{K}_{\inf}^{\mathcal{F}}/\Delta^2$ can be of order $\Delta^{-1}$. We detail this result and the description of the discretization procedure in Appendix D.

We illustrate this gap with an example from [Baudry et al., 2021a], that consider a problem of crop-management optimization in agriculture. In Figure 1, we represent four distributions of crop yields generated from the DSSAT[2] simulator [Hoogenboom et al., 2019], corresponding to different crop-management policies. The distributions are naturally bounded due to the physical constraints of the problem and are non-parametric. In Table 1 we compare $\mathcal{K}_{\inf}^{\mathcal{F}}$ to the Bernoulli KL-divergence, denoted by kl (see (16) for a definition), and to $\Delta^2$ for each arm, and we obtain on the 4-armed bandit that a regret bound scaling with kl or $2\Delta_k^2$ is almost 10 times larger than the asymptotically optimal regret defined by (2). We include this problem in our experiments of Section 4 to check the practical consequences of this asymptotic property.

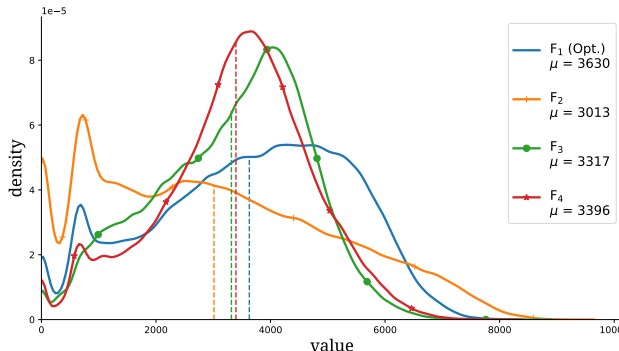

| Dist. | $\dfrac{\mathcal{K}_{\inf}^{\mathcal{F}}(F_k,\mu^\star)}{\text{kl}(\mu_k,\mu^\star)}$ | $\dfrac{\mathcal{K}_{\inf}^{\mathcal{F}}(F_k,\mu^\star)}{2\Delta_k^2}$ |
|---|---|---|
| $F_2$ | 4.4 | 4.72 |
| $F_3$ | 11.73 | 12.44 |
| $F_4$ | 12.04 | 12.73 |

Table 1: Comparison of $\mathcal{K}_{\inf}^{\mathcal{F}}$, kl and $2\Delta_k^2$ for each sub-optimal arm.

Figure 1: Four yield distributions from DSSAT, and comparison of $\mathcal{K}_{\inf}^{\mathcal{F}}$, kl, and $2\Delta^2$.

| Algorithm | Run time | Memory | Optimality |
|---|---|---|---|
| KL-UCB [Cappé et al., 2013] | $\mathcal{O}(n\log(n)^2)$ | $n$ | Opt. |
| kl-UCB | $\mathcal{O}(\log(n))$ | $\mathcal{O}(1)$ | Sub-opt. (kl) |
| UCB [Auer et al., 2002] | $\mathcal{O}(1)$ | $\mathcal{O}(1)$ | Sub-opt. ($2\Delta_k^2$) |
| NPTS [Riou and Honda, 2020] | $\mathcal{O}(n)$ | $n$ | Opt. |
| MED/IMED [Honda and Takemura, 2015] | $\mathcal{O}(n\log(n))$ | $n$ | Opt. |
| Mult. MED/IMED ($M$ items) | $\mathcal{O}(M\log(n))$ | $\mathcal{O}(M)$ | Sub-opt. ($\mathcal{K}_{\inf}^{\mathcal{F}}$ mult.) |
| FMED/FIMED (this paper) | $\mathcal{O}(n\log(n))$ if pulled, $\mathcal{O}(1)$ otherwise. | $n$ | Opt. (Theorem 1) |
| OMED/OIMED (this paper) | $\mathcal{O}(1)$ | $\mathcal{O}(K)$ | Opt. under assumptions of Theorem 2 |

Table 2: Comparison of memory and run time needed per step and for an arm $k$ with $n$ observations

**Outline and contributions**   The results presented in previous paragraphs motivate the search for novel cheap and asymptotically optimal non-parametric bandit algorithms. We build on MED to propose two novel approaches that achieve this goal. We first propose FMED (resp. FIMED) as a fast variant of MED (resp. IMED), that computes $\mathcal{K}_{\inf}^{\mathcal{F}}$ only for the arm that is pulled while for the other arms a first-order Taylor expansion is used. This simple change translates to a considerable speed-up of the algorithms, as for large enough horizons the best empirical arm (for which $\mathcal{K}_{\inf}^{\mathcal{F}}$ is 0) is pulled most of the time, and preserves the theoretical guarantees of the two algorithms. However, FMED and FIMED still require to store all rewards and to fully compute $\mathcal{K}_{\inf}^{\mathcal{F}}$ (sometimes). Hence we propose another approach, in which estimates of $\mathcal{K}_{\inf}^{\mathcal{F}}$ are computed using an *online portfolio selection* algorithm. We highlight a property that would ensure that such an algorithm also keeps the guarantees of MED and IMED, while having much faster computation time and at most $\mathcal{O}(K^2)$ memory.

---

[2]Decision Support System for Agrotechnology Transfer

The novel algorithms are presented in Section 2, while we discuss their guarantees in Section 3. Finally, in Section 4 we perform numerical simulations that confirm the benefits of our novel algorithms in terms of computation time, and show their strong empirical performance. Table 2 summarizes our results. We detail all computations in Appendix F.1.

## 2 Fast MED algorithms

In the following, we denote by $F_k(t)$, $\mu_k(t)$ and $\mu^\star(t) = \max_k \mu_k(t)$ respectively the empirical distribution and mean of arm $k$ and the best empirical mean at time $t$. We propose fast variants of MED and IMED, that avoid computing $\mathcal{K}_{\inf}^{\mathcal{F}}(F_k(t), \mu^\star(t))$ for each arm and at each time.

### 2.1 On-access update and linearization: FMED and FIMED

We use that $\mathcal{K}_{\inf}^{\mathcal{F}}$ is non-decreasing and convex in its second argument (Lemma 2, prop. 1). Given $F \in \mathcal{F}$ and two thresholds $(\mu', \mu) \in [b, B]^2$, and assuming that $\mathcal{K}_{\inf}^{\mathcal{F}}(F, \mu)$ and its derivative (w.r.t $\mu$) have already been computed and are available in memory, we can use that

$$\mathcal{K}_{\inf}^{\mathcal{F}}(F, \mu') \geq \mathcal{K}_{\inf}^{\mathcal{F}}(F, \mu) + \frac{\partial \mathcal{K}_{\inf}^{\mathcal{F}}}{\partial \mu}(F, \mu)(\mu' - \mu) . \tag{5}$$

This inequality is a cheap approximation of $\mathcal{K}_{\inf}^{\mathcal{F}}$ from below when only the second argument changes: as long as $k \in [K]$ is not pulled its empirical distribution $F_k(t)$ is constant. Furthermore, the derivative is equal to the maximizer of (4) [Honda and Takemura, 2015], which is provided at no additional cost when computing $\mathcal{K}_{\inf}^{\mathcal{F}}$. Using this result, we propose FMED and FIMED as fast variants of MED and IMED respectively, for which at each time step $\mathcal{K}_{\inf}^{\mathcal{F}}(F_k(t), \mu^\star(t))$ is replaced by

$$\mathcal{K}_k(t) = \max \left\{ 0, \mathcal{K}_{\inf}^{\mathcal{F}}(F_k(t), \mu^\star(s_k(t))) + \lambda_k(s_k(t)) \times (\mu^\star(t) - \mu^\star(s_k(t))) \right\} , \tag{6}$$

where $s_k(t) = \sup \{ s \leqslant t : A_s = k \}$ is the time of last pull of arm $k$ before $t$, and for $s \in \mathbb{N}$,

$$\lambda_k(s) = \underset{\lambda \in [0, (B - \mu^\star(s))^{-1}]}{\operatorname{argmax}} \mathbb{E}_{F_k(s)} \left[ \log(1 - \lambda(X - \mu^\star(s))) \right] .$$

We detail their implementation in Appendix A.2. Despite its simplicity, this method already permits a huge gain of computation time. Indeed, in Theorem 1 (see Section 3) we prove that each sub-optimal arm is pulled only $\mathcal{O}(\log T)$ in expectation, which is thus the expected number of full $\mathcal{K}_{\inf}^{\mathcal{F}}$ computations. Additionally, at most one $\mathcal{K}_{\inf}^{\mathcal{F}}$ is computed at each stage, while for MED/IMED $K - 1$ computations are required. However, FMED/FIMED still require to keep every sample in memory. This motivates us to investigate an alternative approach in the next section.

### 2.2 Sequential update: OMED and OIMED

We now propose bandit algorithms that update $\mathcal{K}_{\inf}^{\mathcal{F}}(F_k(t), \mu^\star(t))$ in a purely sequential fashion, for any arm $k$. We use that its formulation as a maximization problem (4) can be reformulated as an *online portfolio selection* problem, a class of online convex optimization problems. In the following we drop the index $k$ to simplify the presentation, and thus use the notation $N(t)$ and $F(t)$ for the sample size and empirical cdf of the selected arm at time $t$, and by $X_n$ the $n$-th observation collected by this arm.

**Formulation** Online portfolio selection is a sequential interaction between a learner and an adversary. Before the interaction, the adversary selects $(x_n)_{n \in [N]} \in \mathbb{R}^{N \times d}$ for a dimension $d \geq 2$. At iteration $n \in \mathbb{N}$, the learner chooses $\lambda_n \in \triangle_d \subseteq \mathbb{R}^d$ (the simplex of dimension $d - 1$) and receives the reward $\log(\lambda_n^\top x_n)$. The regret of the learner after $N$ iterations is defined by

$$R_N(x_1, \ldots, x_n) = \max_{\lambda \in \triangle_d} \sum_{n=1}^{N} \log(\lambda^\top x_n) - \sum_{n=1}^{N} \log(\lambda_n^\top x_n) . \tag{7}$$

The objective of the learner is then to find a sequence $(\lambda_n)_{n \in \mathbb{N}}$, where $\lambda_n$ depends only on $(\lambda_1, x_1, \ldots, \lambda_{n-1}, x_{n-1})$, that minimizes $R_N$. We now express the computation of $\mathcal{K}_{\inf}^{\mathcal{F}}(F(t), \mu^\star(t))$

as a portfolio selection algorithm in dimension 2,

$$N(t)\mathcal{K}_{\inf}^{\mathcal{F}}(F(t),\mu^{\star}(t)) = \max_{\lambda\in\Delta_2} \sum_{n=1}^{N(t)} \log\left(\lambda^{\top} x_n(t)\right) , \qquad \text{with } x_n(t) = \left(1, \frac{B-X_n}{B-\mu^{\star}(t)}\right) .$$

However, this expression is unpractical since $x_1(t),\ldots,x_n(t)$ all depend on $\mu^{\star}(t)$, that is not revealed before time $t$. At the $n$-th iteration, we can only use the current best empirical mean, denoted by $\mu_n^{\star}$ for simplicity: a portfolio algorithm can only try to minimize $R_N(x_1,\ldots,x_N)$, with $x_n = (1,(B-X_n)/(B-\mu_n^{\star}))$. We call that quantity *portfolio regret* in the rest of the paper.

On the other hand, if $\mu_n^{\star}$ diverges too often from $\mu^{\star}(t)$ the estimation of $\mathcal{K}_{\inf}^{\mathcal{F}}$ will not be accurate. We define the *bias* of the portfolio algorithm (or portfolio bias) by

$$B_{N(t)}(\mu^{\star}(t)) = \max_{\lambda\in\triangle_2} \sum_{n=1}^{N_k(t)} \log(\lambda^{\top} x_n) - \max_{\lambda\in\Delta_2} \sum_{n=1}^{N(t)} \log\left(\lambda^{\top} x_n(t)\right) . \tag{8}$$

This term can be studied independently of the portfolio regret and only depends on the variations of the best empirical mean. It is not negligible: we will have to modify the structure of the MED algorithms and assume that the best mean is not too close to $B$ to control its magnitude.

In summary, and using all the previously introduced notation, at each stage $t$ we propose to approximate $N(t)\mathcal{K}_{\inf}^{\mathcal{F}}(F(t),\mu^{\star}(t))$ by $L_{N(t)} := \sum_{n=1}^{N(t)} \log(\lambda_n^{\top} x_n)$, where $\lambda_n$ is updated sequentially by a portfolio algorithm. Furthermore, the accuracy of the estimate can be expressed in terms of the portfolio bias and regret as follows,

$$N(t)\mathcal{K}_{\inf}^{\mathcal{F}}(F(t),\mu^{\star}(t)) = L_{N(t)} + R_{N(t)} + B_{N(t)}(\mu^{\star}(t)) , \tag{9}$$

the theoretical results presented in Section 3 are hence obtained by bounding these two quantities.

**Portfolio algorithms** See [Tsai et al., 2023] for a recent review of portfolio selection algorithms. Different methods vary based on their computational complexities and their regret. Some have regret upper bounds as low as $\mathcal{O}(d \log T)$ but are computationally expensive, like UPS [Cover, 1991, Cover and Ordentlich, 1996, Kalai and Vempala, 2002]. Our goal is to use portfolio methods to obtain a computationally efficient $\mathcal{K}_{\inf}^{\mathcal{F}}$, hence we cannot use those. On the other hand, we can afford a looser regret upper bound since we will use it on $\mathcal{O}(\log T)$ samples from the sub-optimal arms. Other algorithms have $\mathcal{O}(d)$ computational complexity per round, which is cheap in our case where $d = 2$. This is, for example, the case for Soft-Bayes [Orseau et al., 2017], which has regret $\mathcal{O}(\sqrt{dT})$. Other algorithms achieve intermediate trade-offs: see [Zimmert et al., 2022, Tsai et al., 2023]. Among the computationally cheap algorithms, we chose Soft-Bayes for its simple implementation and the fact that Orseau et al. [2017] provide bounds on the regret valid for an adaptive step-size parameter, which we need since we don't known the total number of samples we will see for the arms.

While we are the first to use a portfolio algorithm to estimate $\mathcal{K}_{\inf}^{\mathcal{F}}$ in a bandit algorithm, the link between $\mathcal{K}_{\inf}^{\mathcal{F}}$ and portfolio selection was previously used in [Agrawal et al., 2021, Lemma E.1], where the authors use the existence of a portfolio algorithm with logarithmic regret to obtain a concentration inequality on $\mathcal{K}_{\inf}^{\mathcal{F}}$. However, they only use that observation to obtain a bound for the analysis and they don't explore any algorithmic use of the portfolio formulation.

Furthermore, our theoretical analysis is based on a non-standard assumption, that the regret of the portfolio algorithm admits sub-linear upper and lower bounds. We discuss this assumption in the next section.

**Algorithm structure of** `OMED` **and** `OIMED` We introduce the `OMED` and `OIMED` algorithms for a generic portfolio algorithm. As discussed above, using an online estimate of empirical $\mathcal{K}_{\inf}^{\mathcal{F}}$ makes the MED algorithms highly sensitive to variations of the best empirical mean. For this reason, we apply several structural changes to the algorithms in order to prevent the portfolio bias to be large, and assume the knowledge of an upper bound $\mu_{\max}$ on $\mu^{\star}$ with $\mu_{\max} < B$.

First, we use a *duel-based* algorithm: inspired by Chan [2020], at each *round* $t$ we define a *leader* as $\ell_t \in \text{argmax } N_k(t)$[3]. Then, the other arms (called challengers) compete against the leader in pairwise

---

[3]ties are broken in favor of the arm with best empirical mean, then at random if several candidates remain

---

**Algorithm 1** Online Indexed Minimum Empirical Divergence `OIMED`

---

**Input:** $K$ arms, $B$, portfolio selection algorithm `ALG` (and its parameters), function $f$, $\mu_{\max}$.

**Initialization:** Pull each arm once. $\forall (k,\ell) \in [K]^2$ set $N_k = 1$, $\widetilde{N}_{k,\ell} = 0$, $\widehat{\mu}_k = X_{k,1}$, $L_{k,\ell} = 0$,
  $\widehat{\lambda}_{k\ell} = 1/2$ (init. for `ALG`).

**for** $t \geq K$ **do**

  Set $\mathcal{A} = \{\}$, choose leader $\ell \in \arg\max_{k \in [K]} N_k$ ;          ▷ `Break ties comparing` $(\widehat{\mu}_k)$

  **for** $k \neq \ell$ **do**

    **if** $N_\ell \leqslant f(\widetilde{N}_{k,\ell})$ or $\widehat{\mu}_\ell \geqslant \mu_{\max}$ **then** (**if** $\widehat{\mu}_k \geqslant \widehat{\mu}_\ell$ **then** add $k$ to $\mathcal{A}$, set $Z_k = 0$) ;  ▷ `Greedy`

    **else** (**if** $L_{k,\ell} + \log(\widetilde{N}_{k,\ell}) \leqslant \log(N_\ell)$ **then** add $k$ to $\mathcal{A}$, set $Z_k = 1$) ;          ▷ `IMED duel`

  **if** $\mathcal{A} = \{\}$ **then** add $l$ to $\mathcal{A}$, set $Z_l = 0$ ;          ▷ `Pull leader if no challenger`

  **for** $k \in \mathcal{A}$ **do**

    Pull $k$, observe reward $X$, update $\widehat{\mu}_k$, $N_k$. ;      ▷ `General update (even if` $Z_k = 0$`)`

    **if** $Z_k = 1$ **then**

      Update $\widehat{\lambda}_{k,\ell}$ with `ALG`, knowing $\widetilde{N}_{k,\ell}$ and $\frac{B-X}{B-\widehat{\mu}_\ell}$ ;          ▷ `Portfolio update`

      $\widetilde{N}_{k,\ell} \leftarrow \widetilde{N}_{k,\ell} + 1$, $L_{k,\ell} \leftarrow L_{k,\ell} + \log\left(1 - \widehat{\lambda}_{k,\ell}\frac{X-\widehat{\mu}_\ell}{B-\widehat{\mu}_\ell}\right)$ .

---

comparisons called *duels*. At the end of the round, all winning challengers (if any) are pulled. If there are none, the leader is pulled. Hence several arms can be pulled per round. The main ideas are to replace $\mu^\star(t)$ by $\mu_{\ell_t}(t)$ for the reference value used by the challengers, and to implement a different $\mathcal{K}_{\inf}^{\mathcal{F}}$ estimate $L_{k,\ell_t}(t)$ for each possible pair $(k, \ell_t)$.

We now introduce some notation and terminology to describe the duel between a challenger $k$ and a leader $\ell$. In some cases (described below) we perform a *greedy duel*, for which the winner is $\arg\max\{\mu_k(t), \mu_\ell(t)\}$. We introduce a variable $Z_k(t)$, that indicates if the duel played by arm $k$ was greedy ($Z_k(t) = 0$) or not ($Z_k(t) = 1$), and then $\widetilde{N}_{k,\ell}(t) = \sum_{s=1}^{t-1} \mathbb{I}(k \in \mathcal{A}_{s+1}, \ell_s = \ell, Z_k(s) = 1)$, the number of observations of a challenger $k$ collected *against the leader* $\ell$ after a *non-greedy duel*.

Using this notation and considering a function $f : \mathbb{N} \mapsto \mathbb{R}^+$ satisfying $n = o(f(n))$, the duels of `OMED`/`OIMED` are implemented as follow: if $N_\ell(t) \leqslant f(\widetilde{N}_{k,\ell}(t))$ or $\mu_\ell(t) \geqslant \mu_{\max}$ the duel is greedy, otherwise `MED` and `IMED` are respectively adapted as follows.

Non-greedy `OIMED` duel: $\qquad\qquad k$ wins if $\quad L_{k,\ell}(t) + \log(\widetilde{N}_{k,\ell}(t)) \leqslant \log(N_\ell(t))$ .    (10)

Non-greedy `OMED` duel: $\qquad k$ wins if $\quad W_k(t) = 1$, with $W_k(t) \sim \text{Ber}\left(e^{-L_{k,\ell}(t)}\right)$ .    (11)

Then, we update $L_{k,\ell}(t)$ using the portfolio selection algorithm *only* if the duel was non-greedy, i.e. $Z_k(t) = 1$, providing $\mu_\ell(t)$ and the reward collected from $k$: this is the main ingredient to control the portfolio bias in our analysis. We provide a condensed implementation of `OIMED` in Algorithm 1, and the detailed implementation of `OMED` in Appendix A.2 (Algorithm 6).

## 3   Theoretical guarantees

We present our theoretical results on the MED algorithms introduced in Section 2, and some insights from their analysis. We start with the theoretical guarantees of `FMED` and `FIMED`.

**Theorem 1** (Regret bound for fast MED). *Consider* $(F_1, \ldots, F_K) \in \mathcal{F}^K$ *and* $\mu^\star = \max_{k \in [K]} \mu_k$. *Then, for any time horizon* $T \in \mathbb{N}$ *and* $\varepsilon > 0$, `MED`, `IMED`, `FMED` *and* `FIMED` *all satisfy*

$$\mathbb{E}\left[N_k(T)\right] \leqslant \frac{\log(T)}{\mathcal{K}_{inf}^{\mathcal{F}}(F_k, \mu^\star) - \varepsilon} + o_\varepsilon(\log(T)) , \qquad (12)$$

*where* $o_\varepsilon(\log(T))$ *denotes a term that is asymptotically dominated by* $\log(T)$ *for a fixed* $\varepsilon$, *but with a polynomial dependency in* $\varepsilon^{-1}$. *Furthermore, all the algorithms are* ***asymptotically optimal***.

We prove Theorem 1 in Appendix B. The main result is that `FMED` and `FIMED` both preserve the theoretical guarantees of their original algorithm. We apply our analysis to `MED` and `IMED` too, in

order to exhibit more precisely the new terms induced by the approximation, and their scaling in $\varepsilon$. Note that the components of the $o_\varepsilon(\log(T))$ terms are explicit in the proof. When $\varepsilon$ is small, we obtain more precisely a scaling in $\varepsilon^{-6}$, that allows to obtain a sub-linear problem-independent bound of order $T^{5/6}$, though not the optimal $\sqrt{T}$ bound. Note however that the same scaling can be obtained for MED with the proof techniques presented in [Baudry et al., 2023], so the fast implementation do not deteriorate that bound. The strong empirical performance of MED/IMED hints that this result is likely to be not tight, but to the best of our knowledge it remains open to prove that an optimal[4] (instance-dependent) algorithm achieves the $\mathcal{O}(\sqrt{T})$ problem-independent bound for general bounded distributions. Finally, we emphasize that the analysis of IMED presented in Appendix B is drastically simplified compared to Honda and Takemura [2015], although their result is more general (they allow $b = -\infty$).

*Proof sketch.* First, the inequality $\mathcal{K}_k(t) \geqslant \mathcal{K}_{\inf}^{\mathcal{F}}(F_k(t), \mu^\star(t))$ (Eq. (5) and (6)) makes FMED and FIMED at least as exploratory as the vanilla algorithms. Their regret is thus smaller in a *pre-convergence* regime, defined by the time steps for which $\mu^\star(t) \leqslant \mu^\star - \varepsilon$. The main challenge is then to prove that using $(\mathcal{K}_k(t))_{k \in [K]}$ do not lead to over-exploration of the sub-optimal arms in a *post-convergence* regime, where $\mu^\star(t) \geqslant \mu^\star - \varepsilon$. The main ingredient of the proof consists in showing that this kind of scenario can only be caused by events of the form

$$\exists j \in [K]: \{A_{t+1} = j, N_j(t) = n, \mu_{j,n} \geqslant \mu^\star + \varepsilon\} \text{ or } \{A_{t+1} = j, N_j(t) = n, \mu_{j,n} \leqslant \mu^\star - \varepsilon\} .$$

Each of them can cause *at most* 1 *pull* of a sub-optimal arm. With union bounds and Hoeffding's inequality we obtain an additional $\mathcal{O}(\varepsilon^{-2})$ term in the regret bound compared to MED and IMED. $\square$

We now provide a very similar result on OMED and OIMED, under some additional assumptions: first, we need the largest mean to be well-separated from $B$ to control the portfolio bias. This is a mild assumption, since the decision-maker can also choose to slightly increase the value of $B$ used by the algorithm if it does not clearly hold. Then, we require strong guarantees on the portfolio regret, that we discuss at the end of this section.

**Theorem 2** (Regret bound for online MED algorithms). *Consider* $(F_1, \ldots, F_K) \in \mathcal{F}^K$, *and* $\mu^\star = \max_{k \in [K]} \mu_k < \mu_{max}$, *for a known* $\mu_{max} < B$. *Assume that* **OMED** *and* **OIMED** *use a portfolio selection algorithm* **ALG** *that satisfies for any* $(k, \ell) \in [K]^2$ *and* $n \in \mathbb{N}$ *the deterministic guarantee*

$$|R_{k,\ell,n}| = o(n) \quad \text{(sub-linear absolute portfolio regret) .} \tag{13}$$

*Then, both* **OMED** *and* **OIMED** *satisfy Equation* (12) *(with a different second-order term) and are **asymptotically optimal**: they enjoy the same asymptotic guarantees as* MED *and* IMED.

Strikingly, the performance of OMED/OIMED only depend on the regret bounds of the portfolio regret. As detailed below, this is largely due to our modified algorithm's structure.

*Proof sketch.* The general proof scheme is inspired by [Chan, 2020], that proposed a similar duel-based approach. The main difficulty consists in controlling the deviation of $L_{k,\ell_t}(t)$ from $\mathcal{K}_{\inf}^{\mathcal{F}}(F_{k,\ell_t}(t), \mu_{\ell_t}(t))$, denoting by $F_{k,\ell_t}(t)$ the empirical distribution of the $\widetilde{N}_{k,\ell_t}(t)$ observations used to update $L_{k,\ell_t}(t)$. Both over and under-estimation are bad: one may prevent sufficient exploration of the best arm, while the other may lead to over-exploration of sub-optimal arms. Building on the discussion of Section 2, our proof relies on the following crucial result: for fixed arms $(k, \ell)$, a sample size $\widetilde{N}_{k,\ell}(t) = n$, and **any threshold** $\mu$ it holds that

$$n\mathcal{K}_{\inf}^{\mathcal{F}}(F_{k,\ell,n}, \mu) = L_{k,\ell,n} + R_{k,\ell,n} + B_{k,\ell,n}(\mu) , \tag{14}$$

where $B_{k,\ell,n}(\mu)$ is the portfolio bias with respect to a fixed threshold $\mu$, defined in (8), and where $L_{k,\ell,n}$ and $F_{k,\ell,n}$ are used to denote $L_{k,\ell}(t)$ and $F_{k,\ell}(t)$ when $\widetilde{N}_{k,\ell}(t) = n$, for simplicity. In the proofs, the portfolio regret is controlled by assumption, but the bias requires more careful examination. To analyze it, we prove the following result (details in Appendix E). For $i \leqslant n$, we denote by $t_{k,\ell,i}$ the time when the $i$-th iteration of $L_{k,\ell,n}$ occurred.

---

[4]i.e. with bounds depending on $\mathcal{K}_{\inf}^{\mathcal{F}}$ and not $\Delta_k^2$ or kl

**Lemma 1** (Deviations of $B_{k,\ell,n}(\mu)$). *Let $\mu \in (0,1)$, and set $C_{k,\ell,i} = \frac{B-\mu_\ell(t_{k,\ell,i})}{B-\mu}$ it holds that*

$$-\sum_{i=1}^{n} \log\left(C_{k,\ell,i}^{-1}\right) \mathbb{I}(\mu_\ell(t_{k,\ell,i}) \geqslant \mu) \leqslant B_{k,\ell,n}(\mu) \leqslant \sum_{i=1}^{n} \log\left(C_{k,\ell,i}\right) \mathbb{I}(\mu \geqslant \mu_\ell(t_{k,\ell,i})). \quad (15)$$

We always use this result for $\mu \in (\max_{k:\mu_k < \mu^\star} \mu_k, \mu^\star)$, and $\mu_\ell(t_{k,\ell,i}) < \mu_{\max}$. Hence, the upper bound (resp. lower bound) of (15) is expected to be sub-linear when $\ell = 1$ (resp. $\ell \neq 1$), thanks to the use of different estimates according to the leader's identity. The rate of convergence of these sums is also important: our proof techniques work if $n = o(N_\ell(t_{k,\ell,n}))$, which justifies the greedy duels. Indeed, we need $e^{an}\mathbb{P}(B_{k,\ell,n} < -nx)$ to be small enough (for some $a, x > 0$ and large $n$), and with this trick we obtain $e^{an - f(n)y_x} \to 0$ (for some $y_x$ depending on $x$). $\qquad \square$

The modifications introduced in `OMED` and `OIMED` guarantee a small absolute bias with large probability. It is natural to ask whether they are necessary. We cannot formally prove it, but our experiments in Section 4 suggest that the regret of `OIMED`/`OMED` may be linear without them.

**Assumption on the portfolio regret** As detailed in Section 2 there are several candidates for the implementation of `OIMED`/`IMED`. The literature on regret *lower* bounds is however scarce. In [Gofer and Mansour, 2016], the authors characterize algorithms with non-negative regret, but for linear losses. Guzmán et al. [2021] showed that the regret of FTRL algorithms is bounded as $\mathcal{O}(\sqrt{T})$ both from above and below, in a setting that encompasses portfolio selection. Unfortunately, FTRL is computationally inefficient, which makes it unsuitable for our application. We used Soft-Bayes for its simplicity and computational efficiency, but it might not have the desired lower bound. Our problem seems to be among the first where a large negative regret is detrimental, and obtaining a portfolio algorithm with both small computational complexity and a regret lower bound is an open question. On the other hand, it may be that this lower bound requirement could be relaxed in the analysis, and that for example a high-probability lower bound could be enough. Our experimental study in the next section supports the hypothesis that using Soft-Bayes in `OIMED` gives good regret bounds for the bandit problem.

## 4 Experiments

We numerically assess the regret and run time of the two novel approaches presented in this paper. We benchmark our approaches with the known asymptotically optimal algorithms: IMED [Honda and Takemura, 2015], KL-UCB [Cappé et al., 2013] and NPTS [Riou and Honda, 2020]; and the more efficient but sub-optimal UCB [Auer et al., 2002], kl-UCB [Cappé et al., 2013] and kl-IMED (that denotes the binarized version of IMED). We provide the pseudo-code of KL-UCB and NPTS in Appendix A.2 for completeness. Our code is available in the supplementary material of the paper.

We showcase our main findings by presenting a selection of experiments, focusing on FIMED and OIMED, while we present a broader set of experiments and more detailed results in Appendix F. In each case, we plot the average run time and regret of each algorithm, along with their quantiles 10%-90%. The run times are dependent of the Python implementation of our algorithms, and are thus only indicative. We refer to Appendix F.1 for discussions on their theoretical complexity.

**Distributions from DSSAT** We first consider the crop-management optimization problem introduced in Section 1 and detailed for instance in [Baudry et al., 2021a], as it illustrates the theoretical advantage of asymptotically optimal algorithms (Figure 1). A learner needs to select a planting date for maize grains among seven possible options. For each distribution, the rewards are drawn uniformly at random among $10^4$ points sampled from the DSSAT simulator, in order to emulate the distributions at a reduced cost.[5] Our results, displayed in Figure 2, show that on this problem optimal algorithms also achieve better empirical performance. IMED, NPTS and KL-UCB perform similarly, while kl-IMED and UCB achieve significantly larger regret. Furthermore, FIMED is 10 times faster than IMED for $T = 10^4$ (and as fast as kl-UCB), with almost no difference in terms of regret. As expected, OIMED is as fast as UCB and kl-IMED. It achieves better regret than those algorithms, but its performance is deteriorated compare to IMED. This can be seen as the cost of the ability to forget

---

[5]The dataset is available in the supplementary material of the paper

past observations. In summary, this experiment illustrates the respective benefits of our two novel approaches: FIMED is the fastest algorithm among those with the smallest regret, while OIMED has the smallest regret among the fastest algorithm.

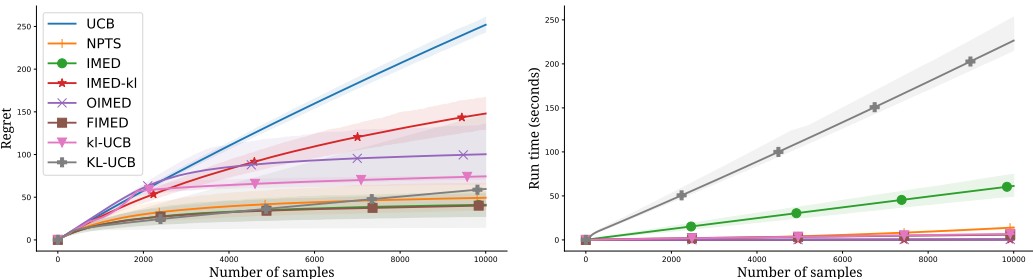

Figure 2: Average regret (left) and run time (right) of the algorithms on the DSSAT bandit problem

**Bernoulli bandit** For our second experiment we consider a Bernoulli bandit with several means close to $0.5$, for which all optimal algorithms match their implementation with the Bernoulli KL-divergence. Intuitively, sequences of $0$ and $1$ with high variance may lead to the most potentially confusing inputs for portfolio algorithms, so our objective is to check the performance of OIMED in that case. Our results, summarized in Figure 3, are promising: the average regret of OIMED is on par with other algorithms, while it still is among the fastest. In that experiment only UCB is sub-optimal, and performs much worse than the other algorithms.

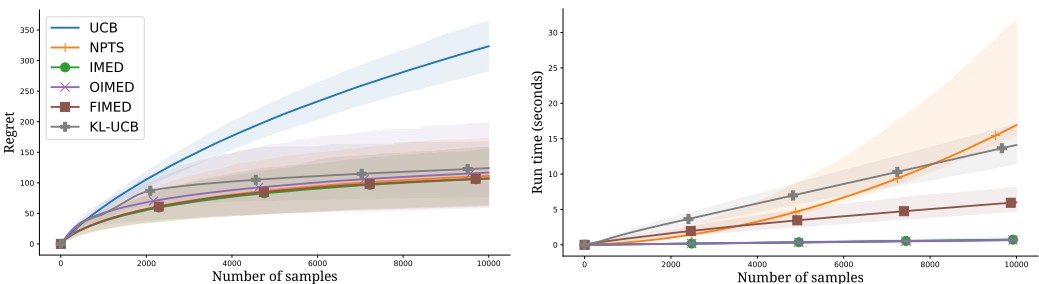

Figure 3: Average regret (left) and run time (right) of the algorithms on a 6-arms Bernoulli bandit problem with means $\{0.3, 0.4, 0.45, 0.5, 0.52, 0.55\}$.

**Importance of the duel-based structure** In Section 3 we detailed our theoretical motivations for introducing a duel-based approach when approximating $\mathcal{K}_{\inf}^{\mathcal{F}}$ with a portfolio algorithm. We now provide empirical evidence that this change may prevent OIMED from suffering a linear regret. In Figure 4, we compare the regret of OIMED and a variant of OIMED that does not use the duel-based structure of Algorithm 1 for the two experiments introduced in previous paragraphs. In that algorithm, the online estimates of $\mathcal{K}_{\inf}^{\mathcal{F}}$ are updated at each pull of an arm with the current best empirical mean. We observe that the regret of the "no duel" variant of OIMED may be better in some cases: for the DSSAT experiment, its average regret matches the best algorithms of Figure 2. However, on the Bernoulli experiment its regret is linear. Overall, OIMED needs a modification like the duels to ensure that it has sub-linear regret.

**Further experiments** In Appendix F we show that FMED and OMED exhibit the same characteristics as FIMED and OIMED, as expected. We also consider other Bernoulli bandits, for instance with means close to the boundaries of the support. We then show that the conclusions of this section still hold on problems with distributions of various shapes, by testing our algorithms on several examples of Beta distributions. We also investigate the sensitivity of OIMED to the learning rate of Soft-Bayes, which is the only hyper-parameter of the algorithm. Finally, we perform experiments with the discretized IMED algorithms that is briefly presented in the introduction and discussed more thoroughly in Appendix D.

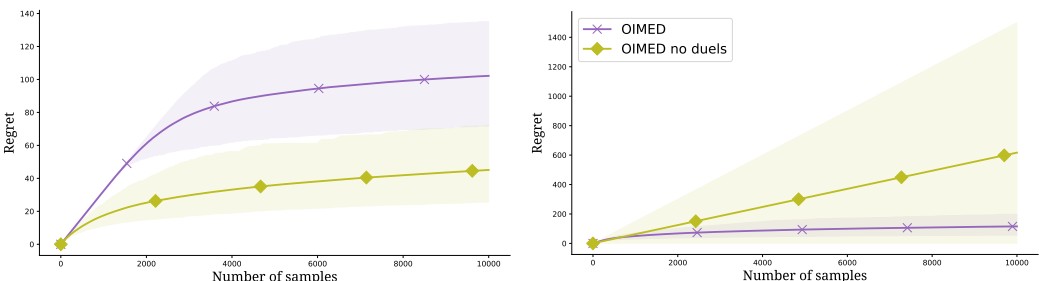

Figure 4: Average regrets of `OIMED` and "OIMED no duels" on the previous 7-armed DSSAT bandit (left) and 6-armed Bernoulli bandit (right).

# 5    Conclusion

We introduced methods to compute efficiently approximations of $\mathcal{K}_{\text{inf}}^{\mathcal{F}}$ and demonstrated their use in algorithms for regret minimization in stochastic bandits. The `FMED` and `FIMED` variants have the same asymptotic optimality properties as the base MED algorithms, but have a much reduced computational complexity. `OMED` and `OIMED` push the computational gains further and are also memory efficient. They can also be asymptotically optimal, under the hypothesis that the portfolio algorithm they use satisfies a deterministic regret lower bound.

While our experiments show the good practical performance of `OIMED` and `OMED` with the Soft-Bayes portfolio algorithm, this question is however still open: can we have a portfolio algorithm which is computationally efficient, does not store all past gains in memory and has sub-linear regret upper and lower bounds?

Finally, our work towards enabling the use of $\mathcal{K}_{\text{inf}}^{\mathcal{F}}$ in a computationally efficient way has potential applications beyond regret minimization in stochastic bandits. First, similar quantities are used in RL algorithms like IMED-RL [Pesquerel and Maillard, 2022]. Second, other bandit tasks like best arm identification also have complexities that depend on a $\mathcal{K}_{\text{inf}}^{\mathcal{F}}$ quantity, and could benefit from faster variants of the algorithms that need to compute it [Jourdan et al., 2022].

## Acknowledgements

R. Degenne acknowledges the funding of the French National Research Agency under the project FATE (ANR-22-CE23-0016-01). This work beneficiated from the support of the French Ministry of Higher Education and Research, of Inria and of the Hauts-de-France region.

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

# A Index, notation and algorithms

## A.1 Index and notation

In this section we summarize the notation used in the proofs of Theorem 1 and 2, and recall some of the definitions presented in the paper.

- $[K] = \{1, \ldots, K\}$.
- $\mathcal{F}$ the family of distributions supported on $[b, B] \subset \mathbb{R}$.
- $(F_1, \ldots, F_K) \in \mathcal{F}^K$ denote the distribution of the $K$ arms of a bandit problem. Their expectations are denoted by $\mu_1, \ldots, \mu_K$. Without loss of generality we assume in the proofs that $\mu_1 = \max_{k \in [K]} \mu_k$, and also define $x_k = \frac{\mu_1 + \mu_k}{2}$.
- For any $F \in \mathcal{F}$, $\mu \in [b, B]$, $\mathcal{K}_{\inf}^{\mathcal{F}}(F, \mu) = \max_{\lambda \in [0,1]} \mathbb{E}_F \left[ \log \left( 1 - \lambda \frac{X - \mu}{B - \mu} \right) \right]$.
- $A_{t+1}$: arm selected by the learner at the beginning of round $t + 1$.
- $N_k(t)$: number of pulls of arm $k$ up to (including) time step $t$.
- $(X_{k,n})_{k \in [K], n \in \mathbb{N}}$: sequences of rewards: $X_{k,n}$ is the $n$-th reward collected from arm $k$.
- $F_k(t)$: empirical distribution of arm $k$ *at time* $t$. We also denote for convenience by $F_{k,n}$ the empirical distribution of arm $k$ *after collecting $n$ observations*: $F_{k,N_k(t)} = F_k(t)$.
- $\mu_k(t)$ and $\mu_{k,n}$ denote the empirical mean of arm $k$ respectively at time $t$ and after collecting $n$ observations.
- $\mu^\star(t) = \max_{k \in [K]} \mu_k(t)$, $k^\star(t) = \underset{k \in [K]}{\text{argmax}}\, \mu_k(t)$.
- For a randomized algorithm, we use the shorthand notation $p_k(t) := \mathbb{P}(A_{t+1} = k | \mathcal{F}_t)$, where $\mathcal{F}_t$ denotes the $\sigma$-algebra generated by the rewards collected by the learner up to (and including) time $t$. We call $p_k(t)$ the *sampling probability* of arm $k$ at time $t$.
- We use "MED" a general terminology to call some existing bandit algorithms based on the computation of $\mathcal{K}_{\inf}^{\mathcal{F}}(F_k(t), \mu^\star(t))$. Some of these algorithms are DMED [Honda and Takemura, 2010], MED [Honda and Takemura, 2011] and IMED [Honda and Takemura, 2015]. Hence, MED denotes a specific algorithm and MED a general family of algorithms in this paper.

FMED and FIMED:

- $s_k(t)$: time of the last pull of arm $k$ at time $t$, formally $s_k(t) = \inf\{s \leqslant t : N_k(s) = N_k(t)\}$.
-
$$\mathcal{K}_k(t) = \mathcal{K}_{\inf}^{\mathcal{F}}(F_k(s_k(t)), \mu^\star(s_k(t))) + \lambda_k(s_k(t))(\mu^\star(t) - \mu^\star(s_k(t))),$$
  is a first order approximation of $\mathcal{K}_{\inf}^{\mathcal{F}}(F_k(t), \mu^\star(t))$ satisfying $\mathcal{K}_k(t) \leqslant \mathcal{K}_{\inf}^{\mathcal{F}}(F_k(t), \mu^\star(t))$, where for any $s \in \mathbb{N}$
$$\lambda_k(s) = \underset{\lambda \in [0, \frac{1}{B - \mu^\star(s)}]}{\text{argmax}} \mathbb{E}_{F_k(s)} \left[ \log(1 - \lambda(X - \mu^\star(s))) \right].$$

OMED and OIMED:

- $\ell_t$: leader at round $t$, defined as $\ell_t \in \underset{k \in [K]}{\text{argmax}}\, N_k(t)$. Ties are broken in favor of the arm with the best empirical mean. If there are still several candidates choose one of them randomly.
- Challengers: all arms except the leader $\ell_t$
- $\mathcal{A}_{t+1}$: set of arms to pull at the beginning of round $t + 1$
- Duel: comparison between a challenger arm $k$ and the leader $\ell_t$ at a time $t$.
- $Z_k(t) = 0$ if the duel for challenger $k$ is a greedy comparison, $Z_k(t) = 1$ otherwise (e.g. if the index comparison of IMED is used). We use the expressions "greedy duel" and "non-greedy duel" to distinguish the two cases.

- $\widetilde{N}_k(t) = \sum_{s=1}^{t-1} \mathbb{I}(k \in \mathcal{A}_{t+1}, \ell_s = \ell_t, Z_k(s) = 1)$: number of observations collected by arm $k$ as a challenger against the current leader $\ell(t)$ and thanks to a non-greedy duel. We denote by $\widetilde{F}_k(t)$ the empirical distribution associated with these observations, and use the shorthand notation $\widetilde{F}_{k,n}$ for $\widetilde{N}_k(t) = n$.

- For any $k \in [K], \ell \in [K], n \in \mathbb{N}$, $X_{k,\ell,n}$ is the $n$-th observation collected from arm $k$ against the leader $\ell$ after a non-greedy duel

- The online estimate used by arm $k$ against leader $\ell$ with $\widetilde{N}_k(t) = n$ is

$$L_{k,\ell,n} := \sum_{i=1}^{n} \log\left(1 - \lambda_{k,i}\left(\frac{X_{k,\ell,i} - \mu^\star(t_{k,\ell,i})}{B - \mu^\star(t_{k,\ell,i})}\right)\right) ,$$

where $t_{k,\ell,i}$ is the time step satisfying $\{\widetilde{N}_k(t+1) = \widetilde{N}_k(t) + 1, \widetilde{N}_k(t) = n\}$, and $(\lambda_{k,i})_{i \in \mathbb{N}}$ is provided by a portfolio selection algorithm.

- 
$$R_{k,\ell,n} := \max_{\lambda \in [0,1]} \sum_{i=1}^{n} \log\left(1 - \lambda\left(\frac{X_{k,\ell,i} - \mu^\star(t_{k,\ell,i})}{B - \mu^\star(t_{k,\ell,i})}\right)\right) - L_{k,\ell,n}$$

is the portfolio regret.

- For any $\mu \in [b, B)$,

$$B_{k,\ell,n}(\mu) := \max_{\lambda \in [0,1]} \sum_{i=1}^{n} \log\left(1 - \lambda\left(\frac{X_{k,\ell,i} - \mu^\star(t_{k,\ell,i})}{B - \mu^\star(t_{k,\ell,i})}\right)\right) - n\mathcal{K}_{\inf}^{\mathcal{F}}(\widetilde{F}_n, \mu)$$

is the bias of the portfolio selection sub-routine with respect to $n\mathcal{K}_{\inf}^{\mathcal{F}}(\widetilde{F}_n, \mu)$.

- We also use $L_{k,\ell_t}(t) = L_{k,\ell_t,\widetilde{N}_k(t)}$, $B_k(t,\mu) = B_{k,\ell_t,\widetilde{N}_k(t)}(\mu)$.

### A.2 Detailed implementations

In this section we detail the implementations of the algorithms presented in the paper, along with their main competitors. We start with the implementation of IMED and MED, presented respectively in Algorithms 2 and 3. We then present FIMED and FMED, respectively in Algorithms 4 and 5. Finally, we detail OMED in Algorithm 6, and recall that the implementation of OIMED is presented in Section 2 (Algorithm 1).

In Algorithm 7, we also detail the implementation of the discretization process proposed by Riou and Honda [2020], that allows to map the rewards in a grid $\mathcal{X} = (x_1, \ldots, x_M)$ without changing the means of the arms.

Finally, for comparison we also detail the implementation of the two other existing optimal algorithms for general bounded rewards: KL-UCB (Algorithm 8), implemented with parameter $f : t \mapsto \log(t) + \log\log(t)$, and NPTS (Algorithm 9).

---

**Algorithm 2** Indexed Minimized Empirical Divergence (IMED)

---

**Input:** $K$ arms, support upper bound $B$
1 **Initialization:** $\forall k \in [K]$: $N_k = 0$ (nb. of pulls), $\widehat{\mu}_k = 0$ (emp. mean), $\mathcal{K}_k = 0$ (emp. $\mathcal{K}_{\inf}^{\mathcal{F}}$).
2 **for** $t \in \mathbb{N}$ **do**
3     **if** $t < K$ **then**
4         Set $a = t$
5     **else**
6         Set $a = \underset{k \in [K]}{\arg\min}\ N_k \mathcal{K}_k + \log(N_k)$ ;    ▷ Index comparison, break ties at random
7     Pull arm $a$, observe reward $X_{a,N_a+1}$, update (or define) $\widehat{\mu}_a$, $N_a$, $\widehat{F}_a$ (emp. distrib.).
        If $t > K$: $\forall k \in [K]$, set $\mathcal{K}_k = \mathcal{K}_{\inf}^{\mathcal{F}}(\widehat{F}_k, \max_{k \in [K]} \widehat{\mu}_k)$. ;    ▷ $\mathcal{K}_{\inf}^{\mathcal{F}}$ update

---

---

**Algorithm 3** Minimized Empirical Divergence (MED)

---

**Input:** $K$ arms, support upper bound $B$

8  **Initialization:** $\forall k \in [K]$: $N_k = 0$ (nb. of pulls), $\widehat{\mu}_k = 0$ (emp. mean), $\mathcal{K}_k = 0$ (emp. $\mathcal{K}_{\text{inf}}^{\mathcal{F}}$).

9  **for** $t \in \mathbb{N}$ **do**

10     **if** $t < K$ **then**

11         Set $a = t$

12     **else**

13         Sample $a \sim \text{Mult}\left(\frac{e^{-N_1 \mathcal{K}_1}}{S}, \dots, \frac{e^{-N_K \mathcal{K}_K}}{S}\right)$, with $S = \sum_{j=1}^{K} e^{-N_k \mathcal{K}_k}$ ;     ▷ Sampling

14     Pull arm $a$, observe reward $X_{a, N_a + 1}$, update (or define) $\widehat{\mu}_a$, $N_a$, $\widehat{F}_a$ (emp. distrib.).
        If $t > K$: $\forall k \in [K]$, set $\mathcal{K}_k = \mathcal{K}_{\text{inf}}^{\mathcal{F}}(\widehat{F}_k, \max_{k \in [K]} \widehat{\mu}_k)$. ;     ▷ $\mathcal{K}_{\text{inf}}^{\mathcal{F}}$ update

---

---

**Algorithm 4** Fast Indexed Minimized Empirical Divergence (FIMED)

---

**Input:** $K$ arms, support upper bound $B$

15  **Initialization:** $\forall k \in [K]$: $N_k = 0$ (nb. of pulls), $\widehat{\mu}_k = 0$ (emp. mean), $\mathcal{K}_k = 0$ (emp. $\mathcal{K}_{\text{inf}}^{\mathcal{F}}$).

16  **for** $t \in \mathbb{N}$ **do**

17     **if** $t < K$ **then**

18         Set $a = t$

19     **else**

20         Set $a = \underset{k \in [K]}{\arg\min} \; N_k \mathcal{K}_k + \log(N_k)$ ;   ▷ Index comparison, break ties at random

21     Pull arm $a$, observe reward $X_{a, N_a + 1}$, update (or define) $\widehat{\mu}_a$, $N_a$, $\widehat{F}_a$ (emp. distrib.).
        **for** $k \in [K]$ **do**

22         **if** $a = k$ **then**

23             If $N_k \geqslant 1$, set $\mathcal{K}_k = \mathcal{K}_{\text{inf}}^{\mathcal{F}}(\widehat{F}_k, \max_{k \in [K]} \widehat{\mu}_k)$ and $\widetilde{\mathcal{K}}_k = \mathcal{K}_k$,
            where $\widehat{F}_k$ is the empirical distribution of $(X_{k1}, \dots, X_{kN_k})$.;   ▷ $\mathcal{K}_{\text{inf}}^{\mathcal{F}}$ computation

24         Keep in memory $\widetilde{\mu}_k^\star = \max \widehat{\mu}_k$, and $\widetilde{\lambda}_k$ the maximizer of (4) when computing $\mathcal{K}_k$.
        **else**

25             Set $\mathcal{K}_k = \widetilde{\mathcal{K}}_k + \widetilde{\lambda}_k \times (\max_{k \in [K]} \widehat{\mu}_k - \widetilde{\mu}_k^\star)$ ;   ▷ Fast update if $k$ not pulled

---

---

**Algorithm 5** Fast Minimized Empirical Divergence (FMED)

---

**Input:** $K$ arms, support upper bound $B$

26  **Initialization:** $\forall k \in [K]$: $N_k = 0$ (nb. of pulls), $\widehat{\mu}_k = 0$ (emp. mean), $\mathcal{K}_k = 0$ (emp. $\mathcal{K}_{\text{inf}}^{\mathcal{F}}$).

27  **for** $t \in \mathbb{N}$ **do**

28     **if** $t < K$ **then**

29         Set $a = t$

30     **else**

31         Sample $a \sim \text{Mult}\left(\frac{e^{-N_1 \mathcal{K}_1}}{S}, \dots, \frac{e^{-N_K \mathcal{K}_K}}{S}\right)$, with $S = \sum_{j=1}^{K} e^{-N_k \mathcal{K}_k}$ ;     ▷ Sampling

32     Pull arm $a$, observe reward $X_{a, N_a + 1}$, update (or define) $\widehat{\mu}_a$, $N_a$, $\widehat{F}_a$ (emp. distrib.).
        **for** $k \in [K]$ **do**

33         **if** $a = k$ **then**

34             If $N_k \geqslant 1$, set $\mathcal{K}_k = \mathcal{K}_{\text{inf}}^{\mathcal{F}}(\widehat{F}_k, \max_{k \in [K]} \widehat{\mu}_k)$ and $\widetilde{\mathcal{K}}_k = \mathcal{K}_k$,
            where $\widehat{F}_k$ is the empirical distribution of $(X_{k1}, \dots, X_{kN_k})$.;   ▷ $\mathcal{K}_{\text{inf}}^{\mathcal{F}}$ computation

35         Keep in memory $\widetilde{\mu}_k^\star = \max \widehat{\mu}_k$, and $\widetilde{\lambda}_k$ the maximizer of (4) when computing $\mathcal{K}_k$.
        **else**

36             Set $\mathcal{K}_k = \widetilde{\mathcal{K}}_k + \widetilde{\lambda}_k \times (\max_{k \in [K]} \widehat{\mu}_k - \widetilde{\mu}_k^\star)$ ;   ▷ Fast update if $k$ not pulled

---

---

**Algorithm 6** Online Minimum Empirical Divergence (`OMED`)

---

**Input:** $K$ arms, upper bound $B$, portfolio selection algorithm `ALG` (and its parameters), function $f$.

37 **Initialization:** $\forall k \in [K]$: $N_k = 0$ (total nb. of pulls), $\widetilde{N}_{k\ell} = 0$ (nb of pulls as a challenger after a non-greedy duel against leader $\ell$), $\widehat{\mu}_k = 0$ (emp. mean), $L_{k\ell} = 0$ (emp. $\mathcal{K}_{\inf}^{\mathcal{F}}$ as a challenger against leader $\ell$), $\widehat{\lambda}_{k\ell} = 1/2$ (initialization for `ALG`).

38 **for** $t \in \mathbb{N}$ **do**
39   **if** $t = 1$ **then**
40     Set $\mathcal{A} = \{1, \dots, K\}$ ;                                         ▷ Pull each arm once
41   **else**
42     Set $\mathcal{A} = \{\}$ ;                     ▷ Initialize the set of arms to pull
43     Set $\mathcal{S} = \operatorname{argmax}_{j \in [K]} N_j$ ;               ▷ Candidates for leadership
44     If $|\mathcal{S}| > 1$, set $\mathcal{S} \leftarrow \operatorname{argmax}_{j \in \mathcal{S}} \widehat{\mu}_j$ ;         ▷ Tie breaking
45     Choose leader $\ell$ uniformly at random in $\mathcal{S}$
      **for** $k \neq \ell$ ;         ▷ Organize duels between $\ell$ and all challengers
46     **do**
47       **if** $\widetilde{N}_{k,\ell} \geqslant f(N_\ell)$ *or* $\widehat{\mu}_\ell \geqslant \mu_{max}$ **then**
48         If $\widehat{\mu}_k \geqslant \widehat{\mu}_\ell$ : add $k$ to $\mathcal{A}$, set $Z_k = 0$ ;   ▷ Greedy duel if $N_\ell$ is too small
49       **else**
50         Draw $W \sim \text{Ber}(e^{-L_{k,\ell}})$, **if** $W_k = 1$ **then**
51           Add $k$ to $\mathcal{A}$, set $Z_k = 1$ ;        ▷ MED duel if $N_\ell$ large enough

52   **for** $k \in \mathcal{A}$ **do**
53     Pull arm $k$, observe a reward $X$, update $\widehat{\mu}_k$, $N_k$.
      **if** $Z_k = 1$ **then**
54       $\widetilde{N}_{k,\ell} \leftarrow \widetilde{N}_{k,\ell} + 1$
      Update $\widehat{\lambda}_{k,\ell}$ with `ALG`, knowing $\widetilde{N}_{k,\ell}$ and $\frac{B-X}{B-\widehat{\mu}_\ell}$ ;     ▷ Portfolio update
55       $L_{k,\ell} \leftarrow L_{k,\ell} + \log\left(1 - \widehat{\lambda}_{k,\ell}\frac{X-\widehat{\mu}_\ell}{B-\widehat{\mu}_\ell}\right)$ .

---

---

**Algorithm 7** Discretization algorithm from [Riou and Honda, 2020]

---

**Input:** Reward $X$, grid $\mathcal{X} = (x_1, \dots, x_M)$

56 Set $m = \max\{j \in [M] : x_j < X\}$ if $X > x_1$, $m = 1$ otherwise.

  Sample $Z \sim \text{Ber}\left(\frac{X - x_j}{x_{j+1} - x_j}\right)$

**Return:** $Y = x_j \mathbb{I}(Z = 0) + x_{j+1}\mathbb{I}(Z = 1)$

---

---

**Algorithm 8** KL-UCB [Cappé et al., 2013]

---

**Input:** $K$ arms, support upper bound $B$

57 **Initialization:** $\forall k \in [K]$: $N_k = 0$ (nb. of pulls), function $f$.

58 **for** $t \in \mathbb{N}$ **do**
59   **if** $t < K$ **then**
60     Set $a = t$
61   **else**
62     **for** $k \in [K]$ **do**
63       Set $\widehat{\mu}_k = \max\{\mu \in [b, B] : N_k \mathcal{K}_{\inf}^{\mathcal{F}}(\widehat{F}_k, \mu) \leqslant f(t)\}$
64     Set $a = \operatorname*{argmax}_{k \in [K]} \widehat{\mu}_k$ ;        ▷ Index comparison, break ties at random

65   Pull arm $a$, observe reward $X_{a, N_a+1}$, update (or define) $\mu_a$, $N_a$, $\widehat{F}_a$ (emp. distrib.).

---

**Algorithm 9** Non-Parametric Thompson Sampling [Riou and Honda, 2020]

**Input:** $K$ arms, support upper bound $B$

66    **Initialization:** $\forall k \in [K]$: $N_k = 0$ (nb. of pulls), $\mathcal{H}_k = \{\}$ (history).

67    **for** $t \in \mathbb{N}$ **do**

68      **if** $t < K$ **then**

69        Set $a = t$

70      **else**

71        **for** $k \in [K]$ **do**

72          Sample weights $w \sim \mathrm{Dir}(1, \ldots, 1)$ ($N_k + 1$ ones)

           Set $I_k = \sum_{i=1}^{N_k} w_i X_{ki} + w_{N_k+1} B$ ;            ▷ Re-weighted mean

73        Set $a = \underset{k \in [K]}{\arg\min} \, I_k$ ;           ▷ Index comparison, break ties at random

74      Pull arm $a$, observe reward $X_{a,N_a+1}$, set $\mathcal{H}_a = \mathcal{H}_a \cup \{X_{a,N_a+1}\}$

# B    Proof of Theorem 1: regret analysis of the index policies

Before proving the theorem, we recall some useful properties from previous works.

## B.1    Toolbox for bounded distributions.

In the proofs we use the following results, that come from several works on bounded distributions in bandits (e.g. [Honda and Takemura, 2015, Cappé et al., 2013]), and are detailed in Appendix D.3 of Baudry et al. [2023].

**Lemma 2** (Useful properties of $\mathcal{K}_{inf}^{\mathcal{F}}$)**.** *Let $F \in \mathcal{F}$ be a distribution of mean $\mu_F$. We denote by $F_n$ and $\mu_n$ respectively the empirical cdf and empirical mean corresponding to $n$ i.i.d. observations collected from $F$. It holds that*

1. *$\mathcal{K}_{inf}^{\mathcal{F}}$ is continuous, non-decreasing and convex in its second argument (Theorem 7 of [Honda and Takemura, 2010]).*

2. *There exists constants $c > 0$, $\delta_0 > 0$ such that $\forall \mu > \mu_F$, and $\delta \in (0, \delta_0]$ it holds that*
$$\mathbb{P}\left(\mathcal{K}_{inf}^{\mathcal{F}}(F_n, \mu) \leqslant \mathcal{K}_{inf}^{\mathcal{F}}(F, \mu) - \delta\right) \leqslant e^{-nc\delta^2} \ .$$

3. *For any $\varepsilon > 0$ and $\mu > \mu_n + \varepsilon$, it holds that*
$$\mathcal{K}_{inf}^{\mathcal{F}}(F_n, \mu) - \mathcal{K}_{inf}^{\mathcal{F}}(F_n, \mu - \varepsilon) \geqslant \delta_\varepsilon := \frac{2\varepsilon^2}{(B-b)^2} \ .$$

4. *For any $x > 0$, it holds that (Lemma 6 from Cappé et al. [2013])*
$$\mathbb{P}(\mathcal{K}_{inf}^{\mathcal{F}}(F_n, \mu_1) > x) \leqslant e(n+2)e^{-nx}.$$

5. *For any $\varepsilon > 0$ (Hoeffding's inequality),*
$$\mathbb{P}(\mu_n \leqslant \mu_F - \varepsilon) \leqslant e^{-2n\varepsilon^2} \quad \text{and} \quad \mathbb{P}(\mu_n \geqslant \mu_F + \varepsilon) \leqslant e^{-2n\varepsilon^2} \ .$$

*Proof.* Property 3. can be deduced from Lemma 13 of Honda and Takemura [2010]. Using the convexity of $\mathcal{K}_{inf}^{\mathcal{F}}$ and Pinsker inequality we obtain that

$$\mathcal{K}_{inf}^{\mathcal{F}}(F_n, \mu) - \mathcal{K}_{inf}^{\mathcal{F}}(F_n, \mu - \varepsilon) \geqslant \mathcal{K}_{inf}^{\mathcal{F}}(F_n, \mu_n + \varepsilon) \geqslant \mathrm{kl}(\mu_n, \mu_n + \varepsilon) \geqslant 2\varepsilon^2 \ ,$$

if the range is $[0, 1]$. The factor $\frac{1}{(B-b)^2}$ comes from using this result on rescaled distributions for general ranges.

Then, property 2. can be obtained with a straightforward adaptation of the proof of Lemma 6 of Agrawal et al. [2021] (in their Appendix B.1). Indeed, in this paper the authors consider a non-parametric family of distributions for which $\mathcal{K}_{inf}^{\mathcal{F}}$ can be expressed in a very analogous way to Equation (4).     □

## B.2 Proof of Theorem 1

We introduce the notation $s_k(t) = \inf\{s \leqslant t : N_k(s) = N_k(t)\}$, which is simply the time step corresponding to the last pull of arm $k$ before time $t$, and $k^\star(t) = \operatorname*{argmax}_{k \in [K]} \mu_k(t)$. We recall Theorem 1.

**Theorem 1** (Regret bound for fast MED). *Consider $(F_1, \ldots, F_K) \in \mathcal{F}^K$ and $\mu^\star = \max_{k \in [K]} \mu_k$. Then, for any time horizon $T \in \mathbb{N}$ and $\varepsilon > 0$, MED, IMED, FMED and FIMED all satisfy*

$$\mathbb{E}\left[N_k(T)\right] \leqslant \frac{\log(T)}{\mathcal{K}_{inf}^{\mathcal{F}}(F_k, \mu^\star) - \varepsilon} + o_\varepsilon(\log(T)) , \tag{12}$$

*where $o_\varepsilon(\log(T))$ denotes a term that is asymptotically dominated by $\log(T)$ for a fixed $\varepsilon$, but with a polynomial dependency in $\varepsilon^{-1}$. Furthermore, all the algorithms are **asymptotically optimal**.*

*Proof.* Equation (12) can be proved for all the algorithms with the same general proof scheme. Hence, we propose a proof that tackle them all at once, explicitly mentioning which parts are specific to a given algorithm. We start by proving a first general upper bound.

**Lemma 3** (Generic regret upper bound). *For each sub-optimal arm $k$ and any $\varepsilon > 0$, FMED and FIMED both satisfy*

$$\mathbb{E}[N_k(T)] \leqslant u_\varepsilon(T) + \underbrace{\mathbb{E}\left[\sum_{t=u}^{T-1} \mathbb{I}(A_{t+1} = k, N_k(t) > u_\varepsilon(T), \mathcal{G}_k(t), \mathcal{J}_k(t))\right]}_{\text{Post-CV}}$$

$$+ \underbrace{\mathbb{E}\left[\sum_{t=u}^{T-1} \mathbb{I}(A_{t+1} = k, N_k(t) > u, \mu^\star(t) \leqslant \mu_1 - \varepsilon_0)\right]}_{\text{Pre-CV}} + \mathcal{O}_\varepsilon(1) ,$$

*where $u_\varepsilon(T) = \left\lceil \frac{\log(T)}{\mathcal{K}_{inf}^{\mathcal{F}}(F_k, \mu_1) - \varepsilon} \right\rceil$, and using the notation $\varepsilon_0 = \frac{B - \mu_1}{4}\varepsilon$ and $\varepsilon_1 = \varepsilon_2 = \frac{\varepsilon}{2}$ we define*

$$\mathcal{J}_k(t) = \{\mathcal{K}_{inf}^{\mathcal{F}}(F_k(t), \mu_1 - \varepsilon_0) \geqslant \mathcal{K}_{inf}^{\mathcal{F}}(F_k, \mu_1) - \varepsilon_1\},$$

*and*

- *For MED and IMED, $\mathcal{G}_k(t) = \{\mu^\star(t) \geqslant \mu_1 - \varepsilon_0\}$.*

- *For FMED and FIMED, $\mathcal{G}_k(t) = \left\{\mu^\star(t) \geqslant \mu_1 - \varepsilon_0, \mu^\star(s_k(t)) \geqslant \mu_1 - \varepsilon_0, \frac{\mu^\star(t) - \mu^\star(s_k(t))}{B - \mu^\star(s_k(t))} \geqslant -\varepsilon_2\right\}$.*

*Proof.* Following for instance the regret analysis of the vanilla MED presented in [Baudry et al., 2023], we upper bound the number of pulls by distinguishing several cases,

$$\mathbb{E}\left[N_k(T)\right] \leqslant u_\varepsilon(T) + \underbrace{\mathbb{E}\left[\sum_{t=1}^{T-1} \mathbb{I}(A_{t+1} = k, N_k(t) > u_\varepsilon(T), \mathcal{G}(t), \mathcal{J}_k(t))\right]}_{\text{Post-CV}}$$

$$+ \underbrace{\mathbb{E}\left[\sum_{t=1}^{T-1} \mathbb{I}(A_{t+1} = k, N_k(t) > u_\varepsilon(T), \mu^\star(t) \leqslant \mu_1 - \varepsilon_0)\right]}_{\text{Pre-CV}}$$

$$+ \underbrace{\mathbb{E}\left[\sum_{t=1}^{T-1} \mathbb{I}(A_{t+1} = k, N_k(t) > u_\varepsilon(T), \overline{\mathcal{J}}_k(t))\right]}_{\text{CV-Emp}}$$

$$+ \underbrace{\mathbb{E}\left[\sum_{t=1}^{T-1} \mathbb{I}\left(A_{t+1} = k, \mu^\star(t) \geqslant \mu_1 - \varepsilon_0, \frac{\mu^\star(t) - \mu^\star(s_k(t))}{B - \mu^\star(s_k(t))} \leqslant -\varepsilon_2\right)\right]}_{\text{Var-Best (FMED and FIMED only)}}$$

$$+ \underbrace{\mathbb{E}\left[\sum_{t=1}^{T-1} \mathbb{I}\left(A_{t+1} = k, \mu^\star(t) \geqslant \mu_1 - \varepsilon_0, \mu^\star(s_k(t)) \leqslant \mu_1 - \varepsilon_0\right)\right]}_{\text{Transition-Best (FMED and FIMED only)}}.$$

This upper bound already exhibits $u_\varepsilon(T)$, Post-CV and Pre-CV. It remains to prove that the additional terms are upper bounded by constants depending on $\varepsilon$. The term CV-Emp is necessary for all algorithms, while the two last terms are specific to FMED and FIMED. Indeed, they tackle the cases where the sub-optimal arm $k$ is pulled due to a value of $\mu^\star(s_k(t))$ that deviates from $\mu_1$ and/or $\mu^\star(t)$. We start by upper bounding CV-Emp, writing

$$\text{CV-Emp} \leqslant \sum_{t=1}^{T-1} \sum_{n=u_\varepsilon(T)}^{T-1} \mathbb{E}\left[\mathbb{I}(A_{t+1} = k, N_k(t) = n, \overline{\mathcal{J}}_{k,n})\right]$$

$$\leqslant \sum_{n=u_\varepsilon(T)}^{T-1} \mathbb{P}\left(\mathcal{K}_{\inf}^{\mathcal{F}}(F_{k,n}, \mu_1 - \varepsilon_0) \leqslant \mathcal{K}_{\inf}^{\mathcal{F}}(F_k, \mu_1) - \varepsilon_1\right)$$

$$\leqslant \sum_{n=u_\varepsilon(T)}^{T-1} \mathbb{P}\left(\mathcal{K}_{\inf}^{\mathcal{F}}(F_{k,n}, \mu_1) \leqslant \mathcal{K}_{\inf}^{\mathcal{F}}(F_k, \mu_1) + \frac{\varepsilon_0}{B - \mu_1} - \varepsilon_1\right),$$

where the last line comes from Lemma 4 of Cappé et al. [2013]. We choose $\varepsilon_1 = 2\frac{\varepsilon_0}{B-\mu_1}$, and obtain that CV-Emp= $\mathcal{O}_{\varepsilon_0}(1)$ thanks to property 2. of Lemma 2. This concludes the proof of Lemma 3 regarding MED and IMED. We now consider the terms specific to FMED and FIMED.

We start with Var-Best, and analyse more precisely the implications of the event

$$\mathcal{B}_k(t) = \left\{\mu^\star(t) \geqslant \mu_1 - \varepsilon_0, \frac{\mu^\star(t) - \mu^\star(s_k(t))}{B - \mu^\star(s_k(t))} \leqslant -\varepsilon_2\right\}.$$

We first prove that the combination of these two events ensure that $\mu^\star(s_k(t)) > \mu_1$ with a proper tuning of $\varepsilon_0$ and $\varepsilon_2$, since

$$\frac{\mu^\star(t) - \mu^\star(s_k(t))}{B - \mu^\star(s_k(t))} \leqslant -\varepsilon_2 \Rightarrow \mu^\star(t) - \mu^\star(s_k(t)) \leqslant -\varepsilon_2(B - \mu^\star(s_k(t)))$$

$$\Rightarrow \mu^\star(s_k(t)) \geqslant \frac{\mu^\star(t) + \varepsilon_2 B}{1 + \varepsilon_2} \geqslant \frac{\mu_1 - \varepsilon_0 + \varepsilon_2 B}{1 + \varepsilon_2}$$

$$\Rightarrow \mu^\star(s_k(t)) \geqslant \mu_1 + \frac{\varepsilon_2(B - \mu_1) - \varepsilon_0}{1 + \varepsilon_2}.$$

Hence, we choose $\varepsilon_2 = 2\frac{\varepsilon_0}{B-\mu_1}$ to obtain $\mu^\star(s_k(t)) \geqslant \mu_1 + \frac{\varepsilon_0}{3}$, if it holds that $\varepsilon_0 \leqslant B - \mu_1$. Then, we also remark that this event also implies some variation of the best empirical mean between $s_k(t)$ and $t$. Hence, only two scenarios can make $\mathcal{B}_k(t)$ hold:

- $k^\star(s_k(t))$ is pulled between $s_k(t)$ and $t$, and $\mu^\star(s_k(t)) \geqslant \mu_1 + \frac{\varepsilon_0}{3}$.

- $k^\star(s_k(t))$ is not pulled between $s_k(t)$ and $t$, $k^\star(t) = j$ for some $j \neq k^\star(s_k(t))$. In that case, $j$ has been pulled between $s_k(t)$ and $t$ and it holds that $\mu_j(t) = \mu^\star(t) \geqslant \mu^\star(s_k(t)) \geqslant \mu_1 + \frac{\varepsilon_0}{3}$.

As each of these scenario can cause at most one pull of arm $k$, we can upper bound Var-Best by simply counting the number of times an arm $j \in [K]$ is pulled (at time $t$) and either $\mu_j(t) \geqslant \mu_1 + \varepsilon_0$ or $\mu_j(t+1) \geqslant \mu_1 + \varepsilon_0$. We hence obtain that

$$
\begin{aligned}
\text{Var-Best} &\leqslant \sum_{j=1}^{K} \sum_{t=1}^{T-1} \mathbb{E}\left[\mathbb{I}\left(A_{t+1} = j, \left\{\mu_j(t) \geqslant \mu_1 + \frac{\varepsilon_0}{3}\right\} \cup \left\{\mu_j(t+1) \geqslant \mu_1 + \frac{\varepsilon_0}{3}\right\}\right)\right] \\
&\leqslant \sum_{j=1}^{K} \sum_{t=1}^{T-1} \sum_{n=1}^{T-1} \mathbb{E}\left[\mathbb{I}\left(A_{t+1} = j, N_j(s) = n, \left\{\mu_{j,n} \geqslant \mu_1 + \frac{\varepsilon_0}{3}\right\} \cup \left\{\mu_{j,n+1} \geqslant \mu_1 + \frac{\varepsilon_0}{3}\right\}\right)\right] \\
&\leqslant 2 \sum_{j=1}^{K} \sum_{n=1}^{+\infty} \mathbb{P}\left(\mu_{j,n} \geqslant \mu_1 + \frac{\varepsilon_0}{3}\right) \\
&= \mathcal{O}_{\varepsilon_0}(1) .
\end{aligned}
$$

We can apply the exact same arguments for transition-best, except that this time we deduce that $k^\star(t)$ was pulled between $s_k(t)$ and $t$, and that (if $N_{k^\star(t)} = n$) $\mu_{j,n} \leqslant \mu_1 - \varepsilon_0$ **and** $\mu_{j,n+1} \geqslant \mu_1 + \varepsilon_0$. Depending on the identity of $k^\star(t)$ one of these two events has exponentially decreasing probability. Formally, we obtain that

$$
\begin{aligned}
\text{Transition-best} &\leqslant \sum_{j:\mu_j < \mu_1} \sum_{n=1}^{+\infty} \mathbb{P}(\mu_{j,n} \geqslant \mu_1 - \varepsilon_0) + \sum_{j:\mu_j = \mu_1} \sum_{n=1}^{+\infty} \mathbb{P}(\mu_{j,n} \leqslant \mu_1 - \varepsilon_0) \\
&= \mathcal{O}_{\varepsilon_0}(1) .
\end{aligned}
$$

This concludes the proof of Lemma 3 by choosing $\varepsilon_0 = \frac{B-\mu_1}{4}\varepsilon$. $\qquad\square$

Building on Lemma 3, we finish to prove Theorem 1 by upper bounding the post-convergence and pre-convergence terms for all algorithms.

**Upper bounding the post-convergence terms** Under $\mathcal{G}_k(t)$ and $\mathcal{J}_k(t)$, it holds thanks to property 1. of Lemma 2 that $\mathcal{K}_{\inf}^{\mathcal{F}}(F_k(t), \mu^\star(t)) \geqslant \mathcal{K}_{\inf}^{\mathcal{F}}(F_k, \mu_1) - \varepsilon$. Hence, MED satisfies

$$
\begin{aligned}
\text{Post-CV}_{\text{MED}} &\leqslant \mathbb{E}\left[\sum_{t=1}^{T-1} \mathbb{I}(N_k(t) > u_\varepsilon(T), \mathcal{G}_k(t), \mathcal{J}_k(t)) \times p_k(t)\right] \\
&\leqslant \mathbb{E}\left[\sum_{t=1}^{T-1} \mathbb{I}(N_k(t) > u_\varepsilon(T), \mathcal{G}_k(t), \mathcal{J}_k(t)) \exp\left(-N_k(t)\mathcal{K}_k(t)\right)\right] \\
&\leqslant T \exp\left(-u_\varepsilon(T)(\mathcal{K}_{\inf}^{\mathcal{F}}(F_k, \mu_1) - \varepsilon)\right) \leqslant 1 ,
\end{aligned}
$$

by definition of $u_\varepsilon(T) \left\lceil \frac{\log(T)}{\mathcal{K}_{\inf}^{\mathcal{F}}(F_k,\mu_1)-\varepsilon} \right\rceil$. Similarly, the design of $\mathcal{G}_k(t)$ for FMED also ensure that $\mathcal{K}_k(t) \geqslant \mathcal{K}_{\inf}^{\mathcal{F}}(F_k,\mu_1) - \varepsilon$ and Post-CV$_{\text{FMED}} \leqslant 1$ by the same arguments. For IMED, we obtain that

$$\text{Post-CV}_{\text{IMED}} \leqslant \mathbb{E}\left[\sum_{t=1}^{T-1}\mathbb{I}(I_k(t) \leqslant I_{k^\star(t)}(t), N_k(t) > u_\varepsilon(T), \mathcal{G}_k(t), \mathcal{J}_k(t))\right]$$

$$\leqslant \mathbb{E}\left[\sum_{t=1}^{T-1}\mathbb{I}(N_k(t)\mathcal{K}_k(t) \leqslant \log(N_{k^\star(t)}(t)) < \log(T), N_k(t) > u_\varepsilon(T), \mathcal{G}_k(t), \mathcal{J}_k(t))\right]$$

$$\leqslant \mathbb{E}\left[\sum_{t=1}^{T-1}\mathbb{I}(u_\varepsilon(T)(\mathcal{K}_{\inf}^{\mathcal{F}}(F_k,\mu_1)-\varepsilon) < \log(T))\right] = 0 \,,$$

so arm $k$ is never pulled in the post-convergence regime with IMED. Again, these arguments directly translate to Post-CV$_{\text{FIMED}}$.

**Upper bounding pre-convergence term** Interestingly, the design of $\mathcal{K}_k(t)$, satisfying $\mathcal{K}_k(t) \geqslant \mathcal{K}_{\inf}^{\mathcal{F}}(F_k(t),\mu^\star(t))$, makes FMED and FIMED more exploratory than MED and IMED. Hence, we can unify the proof for the vanilla algorithms and their fast update in this part.

We start with MED, following Baudry et al. [2023] we first use that for any sampling probability $p_k(t)$ it holds that $p_k(t) \leqslant \frac{p_k(t)}{p_1(t)}p_1(t) \leqslant \frac{1-p_1(t)}{p_1(t)}p_1(t)$, and obtain that

$$\text{Pre-CV}_{\text{MED}} := \mathbb{E}\left[\sum_{t=u}^{T-1}\mathbb{I}(N_k(t) > u, A_{t+1} = k, \mu^\star(t) \leqslant \mu_1 - \varepsilon_0)\right]$$

$$= \mathbb{E}\left[\sum_{t=u}^{T-1}\mathbb{I}(N_k(t) > u, \mu^\star(t) \leqslant \mu_1 - \varepsilon_0)p_k(t)\right]$$

$$\leqslant \mathbb{E}\left[\sum_{t=u}^{T-1}\mathbb{I}(\mu_1(t) \leqslant \mu_1 - \varepsilon_0)\left(\frac{1}{p_1(t)} - 1\right)p_1(t)\right]$$

$$= \mathbb{E}\left[\sum_{t=u}^{T-1}\mathbb{I}(A_{t+1} = 1, \mu_1(t) \leqslant \mu_1 - \varepsilon_0)\left(\frac{1}{p_1(t)} - 1\right)\right] \,.$$

Let us denote by $p_1^{\text{FMED}}(t)$ the sampling probability of arm 1 under FMED. Using that $\sum_{k=1}^{K} e^{N_k(t)\mathcal{K}_k(t)}$ we obtain that

$$p_1^{\text{FMED}}(t) \geqslant \frac{1}{K}\exp\left(-N_1(t)\mathcal{K}_1(t)\right) \geqslant \frac{1}{K}\exp\left(-N_1(t)\mathcal{K}_{\inf}^{\mathcal{F}}(F_1(t),\mu^\star(t))\right) := p_1^{\text{MED}}(t) \,.$$

where $p_1^{\text{MED}}(t)$ is itself a lower bound on the sampling probability of arm 1 under MED (we use this notation with a slight abuse). If follows that Pre-CV$_{\text{FMED}}$ admits the same upper bound as Pre-CV$_{\text{MED}}$. The rest of the proof can be found in Baudry et al. [2023], we briefly recall it from completeness. We first obtain that

$$\text{Pre-CV}_{\text{MED}} \leqslant K\sum_{n=1}^{T-1}\mathbb{E}\left[e^{n\mathcal{K}_{\inf}^{\mathcal{F}}(F_{1,n},\mu_1-\varepsilon_0)} - 1\right] + (K-1)\sum_{n=1}^{T-1}\mathbb{P}(\mu_{1,n} \leqslant \mu_1 - \varepsilon_0)$$

$$\leqslant K\sum_{n=1}^{T-1}\mathbb{E}\left[e^{n\mathcal{K}_{\inf}^{\mathcal{F}}(F_{1,n},\mu_1-\varepsilon_0)} - 1\right] + \mathcal{O}_{\varepsilon_0}(1) \,.$$

Then using properties 3. and 4. from Lemma 2 and $\mathcal{K}_{\inf}^{\mathcal{F}}(F_{1,n}, \mu_1) \leqslant \mathcal{K}^+ := \log\left(\frac{B-b}{B-\mu_1}\right)$ we conclude that

$$
\begin{aligned}
\text{Pre-CV}_{\texttt{MED}} &\leqslant K \sum_{n=1}^{T-1} \int_0^{e^{n\mathcal{K}^+}-1} \mathbb{P}\left(\mathcal{K}_{\inf}^{\mathcal{F}}(F_{1,n}, \mu_1) \geqslant \delta_{\varepsilon_0} + \frac{\log(1+x)}{n}\right) \mathrm{d}x \\
&\leqslant K \sum_{n=1}^{T-1} e(n+2) \int_0^{e^{n\mathcal{K}^+}-1} \frac{e^{-n\delta_{\varepsilon_0}}}{1+x} \mathrm{d}x \\
&= K e \log\left(\frac{B-b}{B-\mu_1}\right) \times \sum_{n=1}^{T-1} n(n+2) e^{-n\delta_{\varepsilon_0}} \\
&= \mathcal{O}_{\varepsilon_0}(1),
\end{aligned}
$$

with $\delta_{\varepsilon_0} = \frac{2\varepsilon_0^2}{(B-b)^2}$. As a consequence, $\text{Pre-CV}_{\texttt{FMED}} = \mathcal{O}_{\varepsilon_0}(1)$ too. For $\texttt{FIMED}$ the relationship with the pre-convergence term of $\texttt{IMED}$ is even more straightforward,

$$
\begin{aligned}
\text{Pre-CV}_{\texttt{FIMED}} &\leqslant \mathbb{E}\left[\sum_{t=u_\varepsilon(T)}^{T-1} \mathbb{I}(N_1(t)\mathcal{K}_1(t) + \log(N_1(t)) \geqslant \log(N_k(t)), N_k(t) > u_\varepsilon(T), \mu^\star(t) \leqslant \mu_1 - \varepsilon_0)\right] \\
&\leqslant \mathbb{E}\left[\sum_{t=u_\varepsilon(T)}^{T-1} \mathbb{I}(N_1(t)\mathcal{K}_{\inf}^{\mathcal{F}}(F_1(t), \mu^\star(t)) + \log(N_1(t)) \geqslant \log(N_k(t)), N_k(t) > u_\varepsilon(T), \mu^\star(t) \leqslant \mu_1 - \varepsilon_0)\right] \\
&= \text{Pre-CV}_{\texttt{IMED}} = \mathcal{O}_{\varepsilon_0}(1).
\end{aligned}
$$

The last statement can be deduced from the regret analysis presented in [Honda and Takemura, 2015], but we now propose a simpler proof of independent interest. We use that if $A_{t+1} = k$, then the index of arm $k$ is smaller than the index of arm 1, and that $N_k(t) > u_\varepsilon(T)$ to first write that

$$
\begin{aligned}
\text{Pre-CV}_{\texttt{IMED}} &\leqslant \mathbb{E}\left[\sum_{t=1}^{T} \mathbb{I}(A_{t+1} = k, N_k(t) \geqslant u_\varepsilon(T), I_1(t) \geqslant \log(N_k(t)))\right] \\
&\leqslant \mathbb{E}\left[\sum_{t=u_\varepsilon(T)}^{T} \sum_{n_1=1}^{T} \sum_{n_k=u_\varepsilon(T)}^{T} \mathbb{I}(A_{t+1} = k, N_k(t) = n_k, N_1(t) = n_1, I_1(t) \geqslant \log(n_k))\right] \\
&\leqslant \mathbb{E}\left[\sum_{n_1=1}^{T} \sum_{n_k=u_\varepsilon(T)}^{T} \mathbb{I}\left(n_1 \mathcal{K}_{\inf}^{\mathcal{F}}(F_{1,n}, \mu_1 - \varepsilon_0) + \log(n_1) \geqslant \log(n_k)\right)\right].
\end{aligned}
$$

To further upper bound this term, we observe that since $\mathcal{K}_{\inf}^{\mathcal{F}}(F_{1,n}, \mu_1 - \varepsilon_0) \leqslant \log\left(\frac{B-b}{B-\mu_1+\varepsilon_0}\right) := \mathcal{K}_{\varepsilon_0}^+$, the events considered above cannot happen if $n_1 \mathcal{K}_{\varepsilon_0}^+ + \log(n_1) \leqslant \log(u_\varepsilon(T)) \leqslant \log(n_k)$. Hence, there exists a function $g : T \mapsto \mathbb{N}$ satisfying $g(T) \to +\infty$ ($g(T)$ would be of order $\log\log(T)$ without the additional $\log(n_1)$) such that

$$
\text{Pre-CV}_{\texttt{IMED}} \leqslant \mathbb{E}\left[\sum_{n_1=g(T)}^{T} \sum_{n_k=u_\varepsilon(T)}^{T} \mathbb{I}\left(n_1 \mathcal{K}_{\inf}^{\mathcal{F}}(F_{1,n}, \mu_1 - \varepsilon_0) + \log(n_1) \geqslant \log(n_k)\right)\right].
$$

Then, using again properties 3. and 4. of Lemma 2 we obtain that

$$
\text{Pre-CV}_{\texttt{IMED}} \leqslant \sum_{n_1=g(u)}^{T} \sum_{n_k=u}^{T} e(n_1+2)e^{-(n_1\delta_{\varepsilon_0}+\log(n_k)-\log(n_1))}
$$

$$
\leqslant e\left(\sum_{n_1=g(T)}^{T} n_1(n_1+2)e^{-n_1\delta_{\varepsilon_0}}\right)\left(\sum_{n_k=u}^{T} \frac{1}{n_k}\right)
$$

$$
\leqslant e\left(\sum_{n_1=g(T)}^{+\infty} n_1(n_1+2)e^{-n_1\delta_{\varepsilon_0}}\right)\log\left(\frac{T}{u_\varepsilon(T)}\right)
$$

$$
= o_{\varepsilon_0}(\log(T)) ,
$$

where the conclusion comes from the fact that the remaining sum is the remainder of a convergent series, or in other words $\text{Pre-CV}_{\texttt{IMED}} \leqslant a_T \log(T)$ where $a_T \to 0$. This is sufficient to conclude the proof, by noting that a careful tuning of $\varepsilon_0$ (for instance as $\log\log(T)$ allows to obtain that $\limsup \frac{\mathbb{E}[N_k(T)]}{\log(T)} \leqslant \frac{1}{\mathcal{K}_{\text{inf}}^{\mathcal{F}}(F_k,\mu_1)}$.

$\square$

## C   Proof of Theorem 2: regret analysis of `OMED` and `OIMED`

We first recall the theorem.

**Theorem 2** (Regret bound for online MED algorithms). *Consider* $(F_1,\ldots,F_K) \in \mathcal{F}^K$, *and* $\mu^\star = \max_{k\in[K]} \mu_k < \mu_{max}$, *for a known* $\mu_{\textbf{max}} < B$. *Assume that* `OMED` *and* `OIMED` *use a portfolio selection algorithm* `ALG` *that satisfies for any* $(k,\ell) \in [K]^2$ *and* $n \in \mathbb{N}$ *the deterministic guarantee*

$$
|R_{k,\ell,n}| = o(n) \quad \text{(sub-linear absolute portfolio regret) .} \tag{13}
$$

*Then, both* `OMED` *and* `OIMED` *satisfy Equation* (12) *(with a different second-order term) and are **asymptotically optimal**: they enjoy the same asymptotic guarantees as* `MED` *and* `IMED`.

*Proof.* As for the proof of Theorem 1 we start by proving a first regret upper bound that holds for both algorithms.

**Lemma 4** (Generic upper bound). *For any sub-optimal arm* $k$, *for any* $\varepsilon > 0$ *it holds that*

$$
\mathbb{E}[N_k(t)] \leqslant u_\varepsilon(T) + \underbrace{\mathbb{E}\left[\sum_{t=1}^{T} \mathbb{I}(k \in \mathcal{A}_{t+1}, \ell_t = 1, \widetilde{N}_{k,1}(t) > u_\varepsilon(T), Z_k(t) = 1, \mathcal{G}_k(t))\right]}_{\text{Post-CV}}
$$

$$
+ 4\log(4)\underbrace{\mathbb{E}\left[\sum_{t=1}^{T} \mathbb{I}(1 \notin \mathcal{A}_{t+1}, \ell_t \neq 1, Z_1(t) = 1)\right]}_{\text{Pre-CV}} + \mathcal{O}_\varepsilon(1) ,
$$

*where* $u_\varepsilon(T) = \frac{\log(T)}{\mathcal{K}_{\text{inf}}^{\mathcal{F}}(F_n,\mu_1)-\varepsilon}$, *and* $\mathcal{G}_k(t) = \left\{L_{k,1}(t) \geqslant \widetilde{N}_{k,1}(t)(\mathcal{K}_{\text{inf}}^{\mathcal{F}}(F_k,\mu_1) - \varepsilon)\right\}$.

Before proving this result, let us detail some intuitions. First, we remark that the two expectations need to be second order terms to make the algorithms asymptotically optimal. The first expectation corresponds to pulls of arm $k$ in a *post-convergence regime*, where arm 1 is the leader and the empirical distribution of arm $k$ is close to $F_k$. On the contrary, the second term is the expected number of duels lost by arm 1 as a challenger, against a sub-optimal leader. For this reason, we name these two terms respectively *Post-CV* and *Pre-CV* (where CV abbreviates convergence). Note that Lemma 4 actually holds for any bandit algorithms that would use the same duel-based structure, independently of what they do during the non-greedy duels.

*Proof.* We start the proof by introducing some notation. Due to the specific structure of the duel-based algorithm we define for a challenger/leader pair $(k, \ell)$ a pseudo-count $\widetilde{N}_{k,\ell}(t)$ and a corresponding empirical cdf $\widetilde{F}_{k,\ell}(t)$: they are computed by considering only the observations of arm $k$ that have been collected *when arm $k$ was pulled after a non-greedy duel ($Z_k(t) = 1$) performed against the leader* $\ell_t = \ell$. Hence, $\widetilde{N}_{k,\ell}(t) \leqslant N_k(t)$. In several parts of the proof we also use constants $x_k \in (\mu_k, \mu_1)$, that we arbitrarily set to $\frac{\mu_k + \mu_1}{2}$. Finally, we introduce two ways to denote the rewards: we continue to call $X_{k,n}$ the $n$-th reward received by arm $k$, but we also introduce the notation $X_{k,\ell,n}$ for the $n$-th reward received by arm $k$ after a non-greedy duel against the leader $\ell$.

We start the analysis by considering the case when the best arm is the leader, and the alternative. For each sub-optimal arm $k$, it holds that

$$\mathbb{E}[N_k(T)] = 1 + \underbrace{\mathbb{E}\left[\sum_{t=1}^{T-1} \mathbb{I}(k \in \mathcal{A}_{t+1}, \ell_t = 1)\right]}_{A_1} + \underbrace{\mathbb{E}\left[\sum_{t=1}^{T-1} \mathbb{I}(k \in \mathcal{A}_{t+1}, \ell_t \neq 1)\right]}_{A_2}$$

We then consider the favorable case where arm 1 the leader, splitting the cases according to the value of $Z_k(t)$.

$$A_1 \leqslant \underbrace{\mathbb{E}\left[\sum_{t=1}^{T-1} \mathbb{I}(k \in \mathcal{A}_{t+1}, \ell_t = 1, \mu_1(t) \leqslant \mu_k(t))\right]}_{B_1 \text{ (greedy duels)}} + \underbrace{\mathbb{E}\left[\sum_{t=1}^{T-1} \mathbb{I}(k \in \mathcal{A}_{t+1}, \ell_t = 1, Z_k(t) = 1)\right]}_{B_2 \text{ (non-greedy duels)}}.$$

We now upper bound $B_1$ using that if $\ell_t = 1$ then $N_1(t) \geqslant t/K$, and that if $\mu_k(t) \geqslant \mu_1(t)$ then either $\mu_k(t) \geqslant x_k$ or $\mu_1(t) \leqslant x_k$. By starting $N_k(t)$ at 1 we cover all possible scenarios for $Z_k(t) = 0$ (including $\mu_{\ell_t}(t) \geqslant \mu_{\max}$). We then obtain

$$B_1 \leqslant \mathbb{E}\left[\sum_{t=1}^{T-1} \mathbb{I}(\ell_t = 1, \mu_1(t) \leqslant x_k)\right] + \mathbb{E}\left[\sum_{t=1}^{T-1} \mathbb{I}(k \in \mathcal{A}_{t+1}, \mu_k(t) \geqslant x_k)\right]$$

$$\leqslant \sum_{t=1}^{+\infty} \sum_{n=t/K}^{+\infty} \mathbb{P}(\mu_{1,n} \leqslant x_k) + \sum_{n=1}^{+\infty} \mathbb{P}(\mu_{k,n} \geqslant x_k) = \mathcal{O}(1).$$

We then consider $B_2$. We use that for any $n$, $\{k \in \mathcal{A}_{t+1}, Z_k(t) = 1, \widetilde{N}_{k,1}(t) = n\} = \{\widetilde{N}_{k,1}(t+1) = \widetilde{N}_{k,1}(t) + 1, \widetilde{N}_{k,1}(t) = n\}$ and can happen only once. For any $u_\varepsilon(T) \in \mathbb{N}$, we hence obtain that

$$B_2 \leqslant u_\varepsilon(T) + \sum_{t=1}^{T-1} \mathbb{E}\left[\mathbb{I}(\widetilde{N}_{k,1}(t+1) = \widetilde{N}_{k,1}(t) + 1, \widetilde{N}_{k,1}(t) > u_\varepsilon(T))\right].$$

We then upper bound $B_2$ by considering separately cases when $\mathcal{G}_k(t)$ holds or not,

$$B_2 \leqslant u + \underbrace{\mathbb{E}\left[\sum_{t=1}^{T} \mathbb{I}(\widetilde{N}_{k,1}(t+1) = \widetilde{N}_{k,1}(t) + 1, \widetilde{N}_{k,1}(t) > u, \mathcal{G}_k(t))\right]}_{\text{Post-CV}}$$

$$+ \underbrace{\mathbb{E}\left[\sum_{t=1}^{T} \mathbb{I}(\widetilde{N}_{k,1}(t+1) = \widetilde{N}_{k,1}(t) + 1, \widetilde{N}_{k,1}(t) > u, \bar{\mathcal{G}}_k(t))\right]}_{B_2'}.$$

We remark the first term is exactly the post-convergence term that we introduced in the lemma. Hence, we leave this expression as it is for this first result. We then consider $B_2'$. We use Equation (14), for any $\varepsilon_0 > 0$ it holds that

$$L_{k,1,\widetilde{N}_{k,1}(t)} = \widetilde{N}_{k,1}(t)\mathcal{K}_{\inf}^{\mathcal{F}}(\widetilde{F}_k(t), \mu_1 - \varepsilon_0) + R_{k,1,\widetilde{N}_{k,1}(t)} + B_{k,1,\widetilde{N}_{k,1}(t)}(\mu_1 - \varepsilon_0).$$

Hence, we obtain that

$$\bar{\mathcal{G}}_k(t) \subset \left\{ \max\left\{ \frac{B_{k,1,\widetilde{N}_{k,1}(t)}(\mu_1 - \varepsilon_0)}{\widetilde{N}_{k,1}(t)}, \frac{R_{k,1,\widetilde{N}_{k,1}(t)}}{\widetilde{N}_{k,1}(t)}, \mathcal{K}_{\inf}^{\mathcal{F}}(F_k, \mu_1) - \mathcal{K}_{\inf}^{\mathcal{F}}(\widetilde{F}_k(t), \mu_1 - \varepsilon_0) \right\} \geqslant \frac{\varepsilon}{3} \right\}.$$

Note that we could have stated different thresholds for each term, but we choose $\varepsilon/3$ in each case for simplicity.

We use the first side of our assumption on the regret of the portfolio algorithm: if $R_n = o(n)$, then there exists $n_0 \in \mathbb{N}$ large enough such that for $n \geqslant n_\varepsilon$, $R_n \leqslant \frac{n\varepsilon}{3}$. Hence, by defining $u_\varepsilon(T) \geqslant n_\varepsilon$ we ensure that $\bar{\mathcal{G}}_k(t)$ is not due to a large portfolio regret.

We now consider the term involving $\mathcal{K}_{\inf}^{\mathcal{F}}(\widetilde{F}_k(t), \mu_1 - \varepsilon_0)$, which is analogous to the CV-Emp term of the proof of Theorem 1 (Appendix B). We hence directly write that

$$\mathbb{E}\left[ \sum_{t=1}^{T} \mathbb{I}\left( \widetilde{N}_{k,1}(t+1) = \widetilde{N}_{k,1}(t) + 1, \widetilde{N}_{k,1}(t) > u_\varepsilon(T), \mathcal{K}_{\inf}^{\mathcal{F}}(\widetilde{F}_k(t), \mu_1 - \varepsilon_0) \leqslant \mathcal{K}_{\inf}^{\mathcal{F}}(F_k, \mu_1) - \frac{\varepsilon}{3} \right) \right]$$

$$\leqslant \sum_{n=u_\varepsilon(T)}^{+\infty} \mathbb{P}\left( \mathcal{K}_{\inf}^{\mathcal{F}}(\widetilde{F}_{k,n}, \mu_1) \leqslant \mathcal{K}_{\inf}^{\mathcal{F}}(F_k, \mu_1) + \frac{\varepsilon_0}{B - \mu_1} - \frac{\varepsilon}{3} \right) = \mathcal{O}_\varepsilon(1),$$

by choosing $\varepsilon_0 = (B - \mu_1)\frac{\varepsilon}{6}$. Finally, we use that $B_{k,1,\widetilde{N}_{k,1}(t)}(\mu_1 - \varepsilon_0) \geqslant \widetilde{N}_{k,1}(t)\varepsilon/3$ is possible only if at least one empirical mean computed with a "reasonable" sample size deviates. More precisely,

$$B_{k,1,\widetilde{N}_{k,1}(t)}(\mu_1 - \varepsilon_0) \geqslant \widetilde{N}_{k,1}(t)\frac{\varepsilon}{3} \Rightarrow \exists n \geqslant \widetilde{N}_{k,1}(t)\frac{\varepsilon}{3} \; : \; \mu_1(t_{k,1,n}) \leqslant \mu_1 - \varepsilon_0$$

$$\Rightarrow \exists n \geqslant f\left( \widetilde{N}_{k,1}(t)\frac{\varepsilon}{3} \right) \; : \; \mu_{1,n} \leqslant \mu_1 - \varepsilon_0,$$

where we used that, thanks to our algorithm, at any time $s$ for which the estimate $L_{k,1}(t)$ was incremented $N_1(s) \geqslant f(\widetilde{N}_{k,1}(s))$ was satisfied (check to decide that the duel is non-greedy). Interestingly, we now get an event that only depends on $\widetilde{N}_{k,1}(t)$. Using Lemma 1 we finally get that

$$\mathbb{E}\left[ \sum_{t=1}^{T} \mathbb{I}\left( \widetilde{N}_{k,1}(t+1) = \widetilde{N}_{k,1}(t) + 1, \widetilde{N}_{k,1}(t) > u_\varepsilon(T), B_{k,1,\widetilde{N}_{k,1}(t)}(\mu_1 - \varepsilon_0) \geqslant \widetilde{N}_{k,1}(t)\frac{\varepsilon}{3} \right) \right]$$

$$\leqslant \sum_{t=u_\varepsilon(T)}^{+\infty} \mathbb{P}\left( \widetilde{N}_{k,1}(t+1) = \widetilde{N}_{k,1}(t) + 1, \widetilde{N}_{k,1}(t) \geqslant u_\varepsilon(T), \exists s \geqslant f\left( \widetilde{N}_{k,1}(t)\frac{\varepsilon}{3} \right) : \mu_{1,s} \leqslant \mu_1 - \varepsilon_0 \right)$$

$$\leqslant \sum_{n=u_\varepsilon(T)}^{+\infty} \mathbb{P}\left( \exists s \geqslant f\left( n\frac{\varepsilon}{3} \right) : \mu_{1,s} \leqslant \mu_1 - \varepsilon_0 \right)$$

$$\leqslant \sum_{n=u_\varepsilon(T)}^{+\infty} \sum_{s=f\left( n\frac{\varepsilon}{3} \right)}^{+\infty} \mathbb{P}\left( \mu_{1,s} \leqslant \mu_1 - \varepsilon_0 \right) = \mathcal{O}_\varepsilon(1),$$

thanks to Hoeffding's inequality. This last step allows to conclude that $B_2' = \mathcal{O}_\varepsilon(1)$, using that $f(s) \geqslant s$ for any $s \in \mathbb{N}$.

**Remark 1.** *We can see that at this step $N_1(s) \geqslant N_k(s)$ was sufficient, the enforcement of $N_1(s) \geqslant f(N_k(s))$ is necessary only in the "pre-convergence" analysis.*

**Upper bounding $A_2$, $\ell_t \neq 1$:** For this part of the proof, we mainly use techniques from [Chan, 2020, Baudry et al., 2021b]. In particular, we use that if the current leader is a sub-optimal arm then either 1 has already been leader and has lost leadership or 1 has never been leader. Formally, we define the event

$$\mathcal{D}_t = \left\{ \exists r \in \left[ \frac{t}{4}, t \right] : \ \ell_r = 1 \right\} .$$

Then, we first upper bound the term

$$C_1 := \sum_{t=1}^{T-1} \mathbb{E}\left[ \mathbb{I}\left( k \in \mathcal{A}_{t+1}, \ell_t \neq 1, \mathcal{D}_t \right) \right] .$$

We use that if $\mathcal{D}_t$ holds and $\ell_t \neq 1$ then arm 1 was the leader at some point between $t/4$ and $t$ and lost its leadership. Furthermore, a change of leadership from 1 to $j$ at time $s$ can only happen if (1) arm $j$ has been pulled at the previous round, (2) the two arms now satisfy $N_1(s) = N_j(s) = n$ for some $n \geq s/K \geq t/(4K)$, and (3) $\mu_{1,n} \leq \mu_{j,n}$. Thanks to these properties, we obtain with some union bounds that

$$C_1 \leq \sum_{j=2}^{K} \sum_{t=1}^{T-1} \sum_{s=\lceil t/4 \rceil}^{t} \sum_{n=\lceil s/K \rceil}^{t} \mathbb{E}\left[ \mathbb{I}\left( j \in \mathcal{A}_s, N_1(s) = N_j(s) = n, \mu_{j,n} \geq \mu_{1,n} \right) \right]$$

$$\leq \sum_{j=2}^{K} \sum_{t=1}^{T-1} \sum_{n=\lceil t/K \rceil}^{t} \mathbb{E}\left[ \mathbb{I}\left( \mu_{j,n} \geq \mu_{1,n} \right) \sum_{s=\lceil t/4 \rceil}^{t} \mathbb{I}(j \in \mathcal{A}_s, N_j(s) = n) \right]$$

$$\leq \sum_{j=2}^{K} \sum_{t=1}^{T-1} \sum_{n=\lceil t/K \rceil}^{t} \mathbb{P}\left( \mu_{j,n} \geq \mu_{1,n} \right)$$

$$= \sum_{j=2}^{K} \sum_{t=1}^{T-1} \sum_{n=\lceil t/K \rceil}^{t} \left( \mathbb{P}\left( \mu_{j,n} \geq x_j \right) + \mathbb{P}\left( \mu_{1,n} \leq x_j \right) \right) = \mathcal{O}(1) .$$

We now upper bound the term

$$C_2 := \sum_{t=1}^{T-1} \mathbb{E}\left[ \mathbb{I}\left( k \in \mathcal{A}_{t+1}, \ell_t \neq 1, \bar{\mathcal{D}}_t \right) \right] .$$

Following Chan [2020], we use that if arm 1 has never been leader between $t/4$ and $t$, then it has necessarily lost at least $t/4$ duels in that time interval. Using Markov inequality, we hence obtain that

$$C_2 \leq \sum_{t=1}^{T-1} \mathbb{P}\left( \mathbb{I}\left( \sum_{s=t/4}^{t} \mathbb{I}(1 \notin \mathcal{A}_{s+1}, \ell_s \neq 1) \geq \frac{t}{4} \right) \right)$$

$$\leq \sum_{t=1}^{T-1} \frac{4}{t} \sum_{s=t/4}^{t} \mathbb{E}\left[ \mathbb{I}(1 \notin \mathcal{A}_{s+1}, \ell_s \neq 1) \right]$$

$$\leq \sum_{s=1}^{T-1} \left( \sum_{t=1}^{T-1} \frac{4}{t} \mathbb{I}(t \in [s, 4s]) \right) \mathbb{E}\left[ \mathbb{I}(1 \notin \mathcal{A}_{s+1}, \ell_s \neq 1) \right]$$

$$\leq 4\log(4) \sum_{t=1}^{T-1} \mathbb{E}\left[ \mathbb{I}(1 \notin \mathcal{A}_{t+1}, \ell_t \neq 1) \right] .$$

Thank to these simple tricks, we are back to upper bounding the total number of duels lost by arm 1 while not being the leader at the cost of a multiplicative constant, which is close to the remaining

term in our statement. The last step of this first generic analysis consists in upper bounding the regret caused by $Z_1(t) = 0$ by a constant. We denote by $f^{-1}$ the function satisfying $f^{-1}(f(s)) = s$ for any $s \in \mathbb{N}$. We use that the greedy duel is caused by either $\mu_{\ell_t}(t) \geqslant \mu_{\max}$ or $\widetilde{N}_{1,\ell_t}(t) \geqslant f^{-1}(t/K)$,

$$D_1 := \sum_{t=1}^{T-1} \mathbb{E}\left[\mathbb{I}\left(1 \notin \mathcal{A}_{t+1}, \ell_t \neq 1, Z_1(t) = 0\right)\right]$$

$$\leqslant \sum_{t=1}^{T-1} \mathbb{E}\left[\mathbb{I}\left(\cup_{j=2}^K \left\{\mu_1(t) \leqslant x_j \cup \mu_j(t) \geqslant x_j, \widetilde{N}_{1,j}(t) \geqslant f^{-1}(t/K), N_j(t) \geqslant t/K\right\}\right)\right]$$

$$+ \sum_{t=1}^{T-1} \mathbb{E}\left[\mathbb{I}\left(\cup_{j=2}^K \left\{\mu_j(t) \geqslant \mu_{\max}\right\}, \ell_t = j\right)\right]$$

$$\leqslant \sum_{t=1}^{T-1} \mathbb{E}\left[\mathbb{I}(\mu_1(t) \leqslant x_j, N_1(t) \geqslant f^{-1}(t/K))\right] + 2\sum_{j=2}^K \mathbb{E}\left[\mathbb{I}(\mu_j(t) \geqslant x_j, N_j(t) \geqslant t/K)\right]$$

$$\leqslant \sum_{t=1}^{T-1} \sum_{n=f^{-1}(t/K)}^{t} \mathbb{P}(\mu_{1,n} \leqslant x_j) + 2\sum_{j=2}^K \sum_{t=1}^{T-1} \sum_{n_j=t/K}^{t} \mathbb{P}(\mu_{j,n} \geqslant x_j) = \mathcal{O}(1) \,,$$

where we grouped upper bounded the terms corresponding to $\mu_j(t) \geqslant \mu_{\max}$ by the terms corresponding to $\mu_j(t) \geqslant x_j$ for simplicity. Finally, the remaining term of our upper bound is exactly

$$\text{Pre-CV} := 4\log(4) \sum_{t=1}^{T-1} \mathbb{E}\left[\mathbb{I}\left(1 \notin \mathcal{A}_{t+1}, \ell_t \neq 1, Z_1(t) = 1\right)\right] \,,$$

as stated in the Lemma 4. This concludes the proof of the lemma,

$$\mathbb{E}[N_k(T)] \leqslant u_\varepsilon(T) + \text{Post-CV} + \text{Pre-CV} + \mathcal{O}_\varepsilon(1) \,.$$

$\square$

We now prove the following lemma, that concludes the proof of Theorem 2.

**Lemma 5.** *OMED and OIMED both satisfy*
$$\text{Post-CV} = \mathcal{O}_\varepsilon(1) \quad \text{and} \quad \text{Pre-CV} = \mathcal{O}(1) \,.$$

*Proof.* We need to upper bound four terms. We start with the upper bounds of the two post-convergence terms, that are straightforward thanks to the tuning of $u_\varepsilon(T)$ and the definition of $\mathcal{G}_k(t)$.

**Upper bounding Post-CV**  We start with OMED,

$$\text{Post-CV}_{\text{OMED}} = \mathbb{E}\left[\sum_{t=1}^T \mathbb{I}(k \in \mathcal{A}_{t+1}, \ell_t = 1, \widetilde{N}_{k,1}(t) > u_\varepsilon(T), Z_k(t) = 1, \mathcal{G}_k(t))\right]$$

$$\leqslant \left[\sum_{t=1}^T e^{-\widetilde{N}_{k,1}(t)(\mathcal{K}_{\inf}^{\mathcal{F}}(F_k,\mu_1)-\varepsilon)} \mathbb{I}(\widetilde{N}_{k,1}(t) > u_\varepsilon(T), Z_k(t) = 1, \mathcal{G}_k(t))\right]$$

$$\leqslant T e^{-u_\varepsilon(T)(\mathcal{K}_{\inf}^{\mathcal{F}}(F_k,\mu_1)-\varepsilon)}$$

$$\leqslant 1 \,,$$

thanks to the definition of $u_\varepsilon(T)$. For OIMED, we obtain

$$\text{Post-CV}_{\text{OIMED}} \leqslant \mathbb{E}\left[\sum_{t=1}^T \mathbb{I}(k \in \mathcal{A}_{t+1}, \ell_t = 1, \widetilde{N}_{k,1}(t) > u_\varepsilon(T), Z_k(t) = 1, \mathcal{G}_k(t))\right]$$

$$\leqslant \left[\sum_{t=1}^T \mathbb{I}(\widetilde{N}_{k,1}(t)(\mathcal{K}_{\inf}^{\mathcal{F}}(F_k,\mu_1) - \varepsilon) < \log(T), \widetilde{N}_{k,1}(t) > u_\varepsilon(T))\right]$$

$$= 0 \,,$$

again thanks to the definition of $u_\varepsilon(T)$.

**Upper bounding Pre-CV, OMED** We recall that

$$\text{Pre-CV}_{\text{OMED}} := \sum_{t=1}^{T-1} \mathbb{E}\left[ \mathbb{I}\left( 1 \notin \mathcal{A}_{t+1}, \ell_t \neq 1, \widetilde{N}_{1,\ell_t}(t) \leqslant f^{-1}(N_j(t)) \right) \right] .$$

We use the notation $p_1(t) = p_{1,j,\widetilde{N}_{1,\ell_t}(t)}$, and $p_{1,j,n} = e^{-L_{1,j,n}}$. Then, with the same arguments as for the regret analysis of the vanilla MED (Baudry et al. [2023], Appendix B) we obtain that

$$
\begin{aligned}
\text{Pre-CV}_{\text{OMED}} &= \sum_{t=1}^{T-1} \mathbb{E}\left[ \mathbb{I}\left( \ell_t \neq 1, \widetilde{N}_{1,\ell_t}(t) \leqslant f^{-1}(N_j(t)) \right) (1 - p_1(t)) \right] \\
&= \sum_{t=1}^{T-1} \mathbb{E}\left[ \mathbb{I}\left( 1 \in \mathcal{A}_{t+1}, \ell_t \neq 1, \widetilde{N}_{1,\ell_t}(t) \leqslant f^{-1}(N_j(t)) \right) \frac{1 - p_1(t)}{p_1(t)} \right] \\
&\leqslant \sum_{j=2}^{K} \sum_{n=1}^{T} \mathbb{E}\left[ \frac{1}{p_{1,j,n}} - 1 \right] \\
&:= \sum_{j=2}^{K} \sum_{n=1}^{T} \mathbb{E}\left[ e^{L_{1,j,n}} - 1 \right] .
\end{aligned}
$$

Now, thanks to Equation (14) we can relate $L_{1,j,n}$ to $\mathcal{K}_{\inf}^{\mathcal{F}}(F_n, x)$ for any $x \in \mathbb{R}$. We again choose $x_j$ for convenience, and obtain that

$$\text{Pre-CV}_{\text{OMED}} \leqslant \sum_{j=2}^{K} \sum_{n=1}^{T} \mathbb{E}\left[ e^{(n\mathcal{K}_{\inf}^{\mathcal{F}}(\widetilde{F}_{1,j,n}, x_j) - R_{1,j,n} - B_{1,j,n}(x_j))_+} - 1 \right] .$$

To relate to the proof for MED we need the portfolio regret and bias to be small enough. We use the second side of our assumption on the regret, which is that $-R_{1,j,n} = o(n)$ for any $j, n$. Hence, for $n$ large enough it holds for instance that $-R_{1,j,n} \leqslant \frac{\delta_j}{3} n$, with

$$\delta_j = \inf_{F \in \mathcal{F}} \mathcal{K}_{\inf}^{\mathcal{F}}(F, \mu_1) - \mathcal{K}_{\inf}^{\mathcal{F}}(F, x_j) > 0 ,$$

where $\delta_j > 0$ is ensured by property 3. of Lemma 2. For the bias, we first obtain with Lemma 1 that

$$-B_{1,j,n}(x_j) \leqslant \sum_{i=1}^{n} \mathbb{I}(\mu_j(t_{1,j,n}) \geqslant x_j) .$$

We then define a "good event" under which the bias is controlled,

$$\mathcal{B}_{j,n} = \left\{ \sum_{i=1}^{n} \mathbb{I}(\mu_j(t_{1,j,n}) \geqslant x_j) \leqslant n \frac{\delta_j}{3} \right\} .$$

We then use a similar proof as for the post-convergence term. We first consider

$$P_1 := \sum_{j=2}^{K} \sum_{n=1}^{T} \mathbb{E}\left[ \left( e^{L_{1,j,n}} - 1 \right) \mathbb{I}(\mathcal{B}_{j,n}) \right] = \sum_{j=2}^{K} \sum_{n=1}^{T} \mathbb{E}\left[ e^{n\mathcal{K}_{\inf}^{\mathcal{F}}(\widetilde{F}_{1,j,n}, x_j) + n\frac{2\delta_j}{3}} - 1 \right] ,$$

that can be upper bounded with the same proof as the vanilla MED from [Baudry et al., 2023]. Using properties 3. and 4. of Lemma 2, we obtain that

$$P_1 \leqslant \sum_{j=2}^{K} \sum_{n=1}^{T} \int_{0}^{e^{n\mathcal{K}_j^+ + \frac{2\delta_j}{3}} - 1} \mathbb{P}\left(\mathcal{K}_{\text{inf}}^{\mathcal{F}}(\widetilde{F}_{1,j,n}, x_j) > \frac{\delta_j}{3} + \frac{1}{n}\log(1+x)\right) dx$$

$$\leqslant \sum_{j=2}^{K} \sum_{n=1}^{T} en(n+1)\left(\mathcal{K}_j^+ + \frac{2\delta_j}{3}\right) e^{-n\frac{\delta_j}{3}}$$

$$= \mathcal{O}(1)\,,$$

with $\mathcal{K}_j^+ = \log\left(\frac{B-b}{B-x_j}\right)$. We then tackle the case where $\mathcal{B}_{j,n}$ does not hold, which is possible only if $\mu_{j,s} \geqslant x_j$ for some $s \geqslant (n\delta_j)^2$. Furthermore, we also use the trivial bound $-B_{1,j,n}(x_j) \leqslant n$ thanks to Lemma 1. We then obtain

$$P_2 := \sum_{j=2}^{K} \sum_{n=1}^{T} \mathbb{E}\left[\left(e^{(n\mathcal{K}_{\text{inf}}^{\mathcal{F}}(\widetilde{F}_{1,j,n}, x_j) - R_{1,j,n} - B_{1,j,n}(x_j))_+} - 1\right)\mathbb{I}(\bar{\mathcal{B}}_{j,n})\right]$$

$$\leqslant \sum_{j=2}^{K} \sum_{n=1}^{T} \mathbb{E}\left[\left(e^{n\mathcal{K}_{\text{inf}}^{\mathcal{F}}(\widetilde{F}_{1,j,n}, x_j) + n(1+\delta_j)} - 1\right)\mathbb{I}(\exists s \geqslant f(n\delta_j) : \mu_{j,s} \geqslant x_j)\right]$$

$$\leqslant \sum_{j=2}^{K} \sum_{n=1}^{T} \mathbb{E}\left[\left(e^{n\mathcal{K}_{\text{inf}}^{\mathcal{F}}(\widetilde{F}_{1,j,n}, x_j) + n(1+\delta_j)} - 1\right) \frac{e^{-f(n\delta_j)I_1(x_j)}}{1 - e^{-I_1(x_j)}}\right]\,,$$

We can conclude at this step using that $e^{n(1+\delta_j) - f(n\delta_j)I_1(x_j)} \to 0$, as we ensured that $n = o(f(n))$, so it simply holds that

$$P_1 = \mathcal{O}(1) \Rightarrow P_2 = \mathcal{O}(1)\,.$$

**Upper bounding Pre-CV, OIMED** We again consider the two cases, depending on if $\mathcal{B}(t)$ holds or not. When this is true, we can use a similar proof to the one of vanilla IMED that we propose in Appendix B,

$$Q_1 := \sum_{j=2}^{K} \sum_{t=1}^{T-1} \mathbb{E}\left[\mathbb{I}\left(\widetilde{N}_{1,j}(t)\mathcal{K}_{\text{inf}}^{\mathcal{F}}(\widetilde{F}_{1,j,n}, x_j) + \log(\widetilde{N}_{1,j}(t)) \geqslant \log(N_j(t)) - \widetilde{N}_{1,j}(t)\frac{2\delta_j}{3}, N_j(t) \geqslant t/K\right)\right]$$

$$\leqslant \sum_{j=2}^{K} \sum_{t=1}^{T-1} \sum_{n=\log(t/K)/\mathcal{K}^+}^{T-1} \mathbb{P}\left(n\mathcal{K}_{\text{inf}}^{\mathcal{F}}(\widetilde{F}_{1,j,n}, x_j) \geqslant \log(t/K) - n\frac{2\delta_j}{3} - \log(n)\right)$$

$$\leqslant \sum_{j=2}^{K} \sum_{t=1}^{T-1} \sum_{n=\log(t/K)/\mathcal{K}^+}^{T-1} \frac{Ken(n+2)}{t} e^{-n\frac{\delta_j}{3}}$$

$$\leqslant K^2 e \sum_{t=1}^{T-1} \sum_{n=\log(t/K)/\mathcal{K}^+}^{T-1} \frac{n(n+2)}{t} e^{-n\frac{\delta_j}{3}}\,,$$

where we again used properties 3. and 4. of Lemma 2 to upper bound the probability. The convergence of this term is ensured by the fact that the sum on $n$ starts at $\frac{\log(t/K)}{\mathcal{K}^+}$. To formally prove this we can for instance use that there exists a constant $C_j$ (that typically scales in $\mathcal{O}(\delta_j^{-2})$ up to logarithm terms) such that $\forall n \in \mathbb{N}, n(n+2) \leqslant C_j e^{n\frac{\delta_j}{6}}$ for any $n \in \mathbb{N}$, so

$$Q_1 \leqslant C_j K^2 \sum_{t=1}^{T-1} \sum_{n=\log(t/K)/\mathcal{K}^+}^{T-1} \frac{e^{-n\frac{\delta_j}{6}}}{t}$$

$$\leqslant C_j K^{2+\frac{\delta_j}{3\mathcal{K}^+}} \frac{1}{1-e^{-\frac{\delta_j}{6}}} \sum_{t=1}^{T-1} \frac{1}{t^{1+\frac{\delta_j}{6\mathcal{K}^+}}}$$

$$= \mathcal{O}(1) .$$

We then consider $\bar{\mathcal{B}}(t)$, under which the bias can be up to $\widetilde{N}_{1,j}(t)$. In that case, we use the same proof techniques as for OMED, with

$$Q_2 \coloneqq \sum_{j=2}^{K} \sum_{t=1}^{T-1} \mathbb{E}\left[\mathbb{I}(\widetilde{N}_{1,j}(t)\mathcal{K}_{\inf}^{\mathcal{F}}(F_{1,j,n},x_j) \geqslant \log(N_j(T)) - \widetilde{N}_{1,j}(t)(1+\delta_j), N_j(t) \geqslant t/K, \bar{\mathcal{B}}(t))\right]$$

$$\leqslant \sum_{j=2}^{K} \sum_{t=1}^{T-1} \sum_{n=\log(t/K)/\mathcal{K}^+}^{T-1} \frac{K}{t} e^{-n\delta_j} \times \frac{e^{n\left(1+\frac{\delta_j}{3}\right)-(n\delta_j)^2 I_1(x_j)}}{1-e^{-I_1(x_j)}} .$$

We then use that the right-hand term converges to 0, so $Q_1 = \mathcal{O}(1) \Rightarrow Q_2 = (1)$. $\qquad\square$

**Remark 2.** *The enforcement of a sufficient sample size for the leader is justified by the analysis of the pre-convergence term. In this regime a linear bias may cost up to $e^n$ for each sample size $n$. Hence, a concentration of the mean with a rate $e^{-cn}$ is not sufficient, especially if $c$ is small (which happens if the gaps are small). Furthermore, this kind of rate cannot be avoided without actively controlling the sample size of the leader at each update.*

This concludes the proof of Theorem 2. $\qquad\square$

# D Accuracy of $\mathcal{K}_{\inf}^{\mathcal{F}}$ approximation and discretization trick

In this section we provide more details on the discussion of Section 1. In particular, we elaborate on the discretization trick proposed in Riou and Honda [2020] and on the sub-optimality of the algorithms that do not target $\mathcal{K}_{\inf}^{\mathcal{F}}$ specifically.

For simplicity we assume that $b = 0$ and $B = 1$, and all results hold for general ranges $[b,B] \subset \mathbb{R}$ up to a rescaling of the rewards: $X \in [b,B] \Rightarrow \frac{X-b}{B-b} \in [0,1]$.

**Sub-optimality of $2\Delta^2$ and kl**  We denote by kl : $[0,1] \times [0,1) \to \mathbb{R}^+$ the KL divergence between two Bernoulli distributions, expressed in terms of their expectations by

$$\forall \mu^\star \geqslant \mu : \quad \text{kl}(\mu,\mu^\star) \coloneqq \mu \log\left(\frac{\mu}{\mu^\star}\right) + (1-\mu) \log\left(\frac{1-\mu}{1-\mu^\star}\right) . \tag{16}$$

Consider $F \in \mathcal{F}$ of mean $\mu \in [0,1)$ and $\mu^\star \in [\mu,1)$, and let us define $\Delta = \mu^\star - \mu$. First, it is known (see e.g. Cappé et al. [2013]) that

$$\mathcal{K}_{\inf}^{\mathcal{F}}(F\mu^\star) \geqslant \text{kl}(\mu,\mu^\star) \geqslant 2\Delta^2 . \tag{17}$$

This result (partially) explains why bandit algorithms adapted for $1/4$-sub-gaussian or Bernoulli distributions work for general bounded distributions (UCB1 [Auer et al., 2002], kl-UCB [Cappé et al., 2013]), since a smaller constant means a more exploratory algorithm. A natural question is to determine how sub-optimal these algorithms can be in the general case. In the following we assume that $\Delta$ is small to perform some first-order approximations.

First, a simple computation shows that $\text{kl}(\mu,\mu^\star) \approx \frac{\Delta^2}{\mu(1-\mu)}$: kl adapts to the true variance of the Bernoulli distributions, exploiting better the geometry of $\mathcal{F}$. The sub-gaussian approximation is tight

only if $\mu$ is close to $1/2$ but not near the boundaries of the support, which is not surprising. We now investigate the gap between $\mathcal{K}_{\inf}^{\mathcal{F}}$ and kl, and recall that the two quantities match for Bernoulli distributions. The following result demonstrates that the ratio between the two can be arbitrarily large if the distributions are in fact "far" from being Bernoulli.

**Lemma 6** (kl vs $\mathcal{K}_{\inf}^{\mathcal{F}}$). *For $\mu \in (0,1)$ and $F \in \mathcal{F}_{[0,1]}$ of mean $\mu - \Delta$, the scaling of $\frac{\mathcal{K}_{\inf}^{\mathcal{F}}(F,\mu)}{kl(\mu_F,\mu)}$ can be as large as $\frac{\mu}{\Delta}$, with the maximum achieved when $F$ is the Dirac distribution with support $\mu - \Delta$.*

*Proof.* First, Jensen inequality and equation (4) provide that

$$\mathcal{K}_{\inf}^{\mathcal{F}}(F,\mu) \leqslant \log\left(1 + \frac{\Delta}{1-\mu}\right) = \mathcal{K}_{\inf}^{\mathcal{F}}(\delta_{\mu-\Delta},\mu) \,,$$

where $\delta_x$ is the Dirac distribution at $x$. Then, approximating $\mathcal{K}_{\inf}^{\mathcal{F}}(F,\mu)$ by $\frac{\Delta}{1-\mu}$ we obtain that

$$\text{for small } \Delta > 0, \quad \frac{\mathcal{K}_{\inf}^{\mathcal{F}}(\delta_{\mu-\Delta},\mu)}{kl(\mu-\Delta,\mu)} \approx \left(\frac{\Delta}{1-\mu}\right) \times \left(\frac{\mu(1-\mu)}{\Delta^2}\right) = \frac{\mu}{\Delta} \xrightarrow{\Delta \to 0} +\infty \,.$$

$\square$

We could argue that Dirac (or highly concentrated) distributions may be unlikely in most applications, however the use-case presented in Section 1 (Figure 1) shows that large ratios $\mathcal{K}_{\inf}^{\mathcal{F}}/kl$ may already be observed with more natural examples. Finally, to compare the impact of the approximation on the asymptotic guarantees of an algorithm on a $K$-armed bandits we can compare $\sum_{k:\Delta_k>0} \frac{\Delta_k}{\mathcal{K}_{\inf}^{\mathcal{F}}(F_k,\mu^\star)}$ and $\sum_{k:\Delta_k>0} \frac{1}{\Delta_k}$.

**Discretization trick** We now detail the trick proposed by Riou and Honda [2020], and inspired for instance by the binarization trick of Agrawal and Goyal [2012]. For any $M \in \mathbb{N}$, we fix an arbitrary grid $\mathcal{X}_M = (x_1 = 0 < x_2 < \cdots < x_M = 1) \in [0,1]^M$. Then, the trick consists in transforming any incoming reward $X \notin \mathcal{X}_M$ in a discrete reward $Y \in \mathcal{X}_M$, as follows:

1. Let $m \in \mathbb{N}$ satisfy $x_m < X < x_{m+1}$ , and define $p = \frac{X-x_m}{x_{m+1}-x_m}$.
2. Sample $Z \sim \text{Ber}(p)$ and define $Y = x_m \mathbb{I}(Z=0) + x_{m+1}\mathbb{I}(Z=1)$.

If this procedure is used with all rewards collected, one can use a bandit algorithm calibrated for multinomial distributions with support $\mathcal{X}_m$. It is easy to see that $\mathbb{E}[Y] = X$, so the order of the arms do not change with this discretization. Furthermore, the bandit problem becomes equivalent to a problem where the distributions would be exactly $(F_1^{\mathcal{X}_M}, \ldots, F_K^{\mathcal{X}_M}) \in \mathcal{F}^K$ where, for each $k \in [K]$, $F_k^{\mathcal{X}_M}$ is the distribution of the discretized rewards (with reference grid $\mathcal{X}_M$) generated with i.i.d. samples from $F_k$. It is hence clear that an asymptotically optimal bandit algorithm for (general bounded distributions) is guaranteed an asymptotic regret scaling in $\sum_{k:\Delta_k>0} \frac{\Delta_k}{\mathcal{K}_{\inf}^{\mathcal{F}}(F_k^{\mathcal{X}_M})} \log(T)$, and be optimal if the distributions $F_1, \ldots, F_K$ are already supported on $\mathcal{X}_M$ only.

We now discuss the performance that can be achieved with such discretization. Let us assume that a constant step $x_{m+1} - x_m = \frac{B-b}{M-1}$ is chosen for all $m \in [M-1]$. First, due to the compactness of $\mathcal{F}$ we can state that there exists a distribution $G_\star$ satisfying $\mathcal{K}_{\inf}^{\mathcal{F}}(F_k,\mu^\star) = \text{KL}(F_k,G_\star)$. We then denote by $G_\star^{\mathcal{X}_M}$ its corresponding discretized distribution on the grid $\mathcal{X}_M$. Thanks to the data processing inequality, we obtain that

$$\mathcal{K}_{\inf}^{\mathcal{F}}(F_k,\mu^\star) = \text{KL}(F_k,G_\star) \geqslant \text{KL}(F_k^{\mathcal{X}_M}, G_\star^{\mathcal{X}_M}) \,,$$

which is itself larger than $\mathcal{K}_{\inf}^{\mathcal{F}}(F_k^{\mathcal{X}_M},\mu^\star)$ by definition, since $\mathbb{E}_{X \sim G_\star^{\mathcal{X}_M}}[X] = \mathbb{E}_{X \sim G_\star}[X] \geqslant \mu^\star$. Finally, using that $F_k^{\mathcal{X}_M}$ cannot be "closer" to $\mu^\star$ than a translation of $F_k$ by $\eta$, and Lemma 4 of Cappé et al. [2013], we obtain that

$$\mathcal{K}_{\inf}^{\mathcal{F}}(F_k,\mu^\star) \geqslant \mathcal{K}_{\inf}^{\mathcal{F}}(F_k^{\mathcal{X}_M},\mu^\star)$$
$$\geqslant \mathcal{K}_{\inf}^{\mathcal{F}}(F_k,\mu^\star - \eta) \geqslant \mathcal{K}_{\inf}^{\mathcal{F}}(F_k,\mu^\star) - \frac{\eta}{B-\mu^\star}$$
$$= \mathcal{K}_{\inf}^{\mathcal{F}}(F_k,\mu^\star) - \frac{1}{M-1} \times \frac{B-b}{B-\mu^\star} \,.$$

Hence, we see that the approximation can be good for large $M$. However, if $\mathcal{K}_{\text{inf}}^{\mathcal{F}}(F_k, \mu^\star) \leqslant \frac{1}{M-1} \times \frac{B-b}{B-\mu^\star}$ (which is possible since $M$ is fixed without knowing the distributions) then the lower bound is vacuous. In fact, we can extend Lemma 6 to *any* discretization process: small gaps (compared to $M^{-1}$) can make the ratio $\mathcal{K}_{\text{inf}}^{\mathcal{F}}(F_k, \mu^\star)/\mathcal{K}_{\text{inf}}^{\mathcal{F}}(F_k^{\mathcal{X}_m}, \mu)$ arbitrarily large. Hence, even if this technique has some convenient properties, it is not entirely satisfying. This justifies our quest for alternatives that would preserve the asymptotic optimality of the algorithms.

# E    Online $\mathcal{K}_{\text{inf}}^{\mathcal{F}}$ computation with portfolio optimization algorithms

In this section we provide additional details regarding `OMED` and `OIMED`. We first prove Lemma 1, that is essential to their regret analysis, then we detail the Soft-Bayes algorithm [Orseau et al., 2017] that we choose in our practical implementation of the algorithms.

## E.1    Proof of Lemma 1

**Lemma 1** (Deviations of $B_{k,\ell,n}(\mu)$). *Let $\mu \in (0,1)$, and set $C_{k,\ell,i} = \frac{B-\mu_\ell(t_{k,\ell,i})}{B-\mu}$ it holds that*

$$-\sum_{i=1}^{n} \log\left(C_{k,\ell,i}^{-1}\right) \mathbb{I}(\mu_\ell(t_{k,\ell,i}) \geqslant \mu) \leqslant B_{k,\ell,n}(\mu) \leqslant \sum_{i=1}^{n} \log\left(C_{k,\ell,i}\right) \mathbb{I}(\mu \geqslant \mu_\ell(t_{k,\ell,i})) . \quad (15)$$

*Proof.* We introduce some notation for convenience. First, we denote $X_{k,\ell,i}$ by $X_i$ and $\mu^\star(t_{k,\ell,i})$ by $\mu^\star(i)$. Then, we define $Y_i = \frac{X_i - \mu^\star(i)}{B - \mu^\star(i)}$, $Z_i = \frac{X_i - \mu}{B - \mu}$, and $\lambda = \underset{\lambda \in [0,1]}{\text{argmax}} \sum_{i=1}^{n} \log(1 - \lambda Z_i)$.

We use two elementary analysis properties. First, $\frac{1 - \lambda Z_i}{1 - \lambda Y_i}$ is positive and non-decreasing if $Y_i \geqslant Z_i$. Then, $Y_i \geqslant Z_i$ only if $\mu^\star(i) \leqslant \mu$. Otherwise, $\log(1 - \lambda Z_i) - \log(1 - \lambda Y_i) \leqslant 0$. Hence, it holds that

$$\begin{aligned}
B_{k,\ell,n}(\mu) &:= \sum_{i=1}^{n} \left(\log(1 - \lambda Z_i) - \log(1 - \lambda Y_i)\right) = \sum_{i=1}^{n} \log\left(\frac{1 - \lambda Z_i}{1 - \lambda Y_i}\right) \\
&\leqslant \sum_{i=1}^{n} \log\left(\frac{1 - Z_i}{1 - Y_i}\right) \mathbb{I}(\mu \geqslant \mu^\star(i)) .
\end{aligned}$$

We then plug the expression of $Y_i$ and $Z_i$ in this bound, obtaining that

$$B_{k,\ell,n}(\mu) \leqslant \sum_{i=1}^{n} \log\left(\frac{\frac{B-X_i}{B-\mu}}{\frac{B-X_i}{B-\mu^\star(i)}}\right) \mathbb{I}(\mu \geqslant \mu^\star(i)) = \sum_{i=1}^{n} \log\left(\frac{B - \mu^\star(i)}{B - \mu}\right) \mathbb{I}(\mu \geqslant \mu^\star(i)) ,$$

which gives the result. Applying the exact same steps provides the other direction,

$$-B_{k,\ell,n} \leqslant \sum_{i=1}^{n} \log\left(\frac{B - \mu}{B - \mu^\star(i)}\right) \mathbb{I}(\mu^\star(i) \geqslant \mu)$$

$\square$

## E.2    Soft-Bayes

In this section we detail the Soft-Bayes algorithm, proposed in Orseau et al. [2017]. We recall that the $\mathcal{K}_{\text{inf}}^{\mathcal{F}}$ estimation is a portfolio optimization problem of dimension 2. At each step $n$, the portfolio algorithm decides an allocation $(1 - \lambda_n, \lambda_n)$ between two assets, that provide a payoff $(1, Y_n)$. We use in this section $Y_n$ to denote for simplification $\frac{B - X_{k,\ell,n}}{B - \mu^\star(t_{k,\ell,n})}$, that is used in our implementation of the estimate $L_{k,\ell,n}$ for the challenger/leader pair $(k, \ell)$.

For the anytime version of the algorithm, a sequence of learning rates $(\eta_n)_{n \in \mathbb{N}}$ is provided as an input of the algorithm. We then define the update rule of the anytime Soft-Bayes in Algorithm 10 below.

---
**Algorithm 10** Anytime Soft-Bayes [Orseau et al., 2017]
---
**Input:** Parameter $\lambda_n$, sequence of learning rates $(\eta_n)_{n \in \mathbb{N}}$, initial parameter $\lambda_1 = 1/2$

75 **Return:** $\lambda_{n+1} = \lambda_n \times \left(1 - \eta_n + \eta_n \frac{Y_n}{1 - \lambda_n(1 - Y_n)}\right) + \left(1 - \frac{\eta_{n+1}}{\eta_n}\right)\lambda_1$

---

Orseau et al. [2017] prove that defining the learning rate as $\eta_n = \sqrt{\frac{\log(2)}{4n}}$ ensures an upper bound on the portfolio regret

$$\mathcal{R}_N = 4\sqrt{\log(2)N} + (1 + \log(2))\log(N + 1) + \log(2) .$$

However, they do not provide a lower bound on the regret.

## F  Supplementary experimental results

### F.1  Detail of the costs of each algorithm

We start this section by detailing the computation of the costs of the algorithms presented in Table 2. We first recall the results that we want to prove.

| Algorithm | Run time | Memory | Optimality |
|---|---|---|---|
| KL-UCB [Cappé et al., 2013] | $\mathcal{O}(n \log(n)^2)$ | $n$ | Opt. |
| kl-UCB | $\mathcal{O}(\log(n))$ | $\mathcal{O}(1)$ | Sub-opt. (kl) |
| UCB [Auer et al., 2002] | $\mathcal{O}(1)$ | $\mathcal{O}(1)$ | Sub-opt. $(2\Delta_k^2)$ |
| NPTS [Riou and Honda, 2020] | $\mathcal{O}(n)$ | $n$ | Opt. |
| MED/IMED [Honda and Takemura, 2010, 2015] | $\mathcal{O}(n \log(n))$ | $n$ | Opt. |
| Mult. MED/IMED ($M$ items) | $\mathcal{O}(M \log(n))$ | $\mathcal{O}(M)$ | Sub-opt. ($\mathcal{K}_{\inf}^{\mathcal{F}}$ mult.) |
| FMED/FIMED (this paper) | $\mathcal{O}(n \log(n))$ if pulled, $\mathcal{O}(1)$ otherwise. | $n$ | Opt. (Theorem 1) |
| OMED/OIMED (this paper) | $\mathcal{O}(1)$ | $\mathcal{O}(K)$ | Opt. w. assumptions of Theorem 2 |

Table 3: Comparison of memory and run time needed per step and for an arm $k$ with $n$ observations

In the table, $n$ is a number of observations for 1 arm, and the costs are computed for this arm only for simplicity.

**Memory cost**  We start with the memory, which is easier to estimate. Most optimal algorithms (KL-UCB, NPTS, MED/IMED, FMED/FIMED) require to store all the observations, for a cost of $n$. On the contrary, UCB and kl-UCB require constant memory, since they just need to store the number of pulls and the empirical means of each arm. Finally, the multinomial version of the algorithms have a memory cost of $M$: they need to store the total number of pulls, the frequency of each item $(x_1, \ldots, x_M)$ on the chosen grid, and the empirical means.

Finally, $K$ appears in the memory cost of OIMED because for each arm $k$ we need to store (at most) $K$ online estimates of $\mathcal{K}_{\inf}^{\mathcal{F}}$ and pseudo-counts, in addition to the total number of pulls and empirical mean of the arm. This is a worst-case memory, in case each arm have been leader at least once. The total memory cost of OIMED (summing over all arms) is then at most $\mathcal{O}(K^2)$, while it is $\mathcal{O}(T)$ for the optimal algorithms that store all observations.

**Computation costs**  First, we assume that any algorithm that use only quantities that can be updated sequentially has a cost of $\mathcal{O}(1)$. We include in this category updates of means, number of pulls, or the output of a portfolio algorithm with a running time independent of $n$. We insist that in practice the cost of these algorithms is not exactly the same, but it can be verified that they remain constant over time.

We now consider NPTS: the main cost comes from the sampling of $n + 1$ weights from a Dirichlet distribution of parameter $(1, \ldots, 1)$ ($n$ ones). This sampling is linear in $n$, since it can be done by sampling $n+1$ i.i.d. samples $R_1, \ldots, R_{n+1}$ from the exponential distribution $\mathcal{E}(1)$, and then defining $w_i = \frac{R_i}{\sum_{j=1}^{n+1} R_j}$ for any $i \in [n+1]$. This justifies the cost in $\mathcal{O}(n)$ in the table.

The remaining algorithms (KL-UCB and MED algorithms) require to solve the optimization problem defined by (4) to compute empirical $\mathcal{K}_{\text{inf}}^{\mathcal{F}}$. This can be done with any search algorithm, which costs $n \log(1/\varepsilon)$ where $\varepsilon$ is the precision asked for the solution. In the implementations we always choose $\varepsilon = 1/n$, so that the precision is enough to preserve the theoretical guarantees of the algorithms. This already provides the cost of the MED algorithms, that require to compute exactly one $\mathcal{K}_{\text{inf}}^{\mathcal{F}}$ for each arm at each time step. For FMED/FIMED, we precise that this computation happens only if the arm wad pulled. Finally, for the discretized version of the algorithms the computation time becomes linear in $M$ instead of $n$, but the $\log(n)$ is kept since the precision is still $1/n$.

For KL-UCB and kl-UCB, another optimization procedure is needed: for a given function $f$ we solve $\max\left\{\mu \in [b, B] : N_k(t)\mathcal{K}_{\text{inf}}^{\mathcal{F}}(F_k(t), \mu) \leqslant f(t)\right\}$. We also set the precision of this search to $1/n$, which gives the supplementary multiplicative $\log(n)$ factor compared with MED algorithms: each $\mathcal{K}_{\text{inf}}^{\mathcal{F}}$ computation costs $\mathcal{O}(n\log(n))$, and we perform $\mathcal{O}(\log(n))$ such computations for a step of KL-UCB. This is also the reason why the cost of kl-UCB is not constant: we perform $\mathcal{O}(\log(n))$ computations of kl.

**Remark 3** (Total computation time). *We detailed the cost for one arm and one step of the algorithms only for simplicity, however for those depending on $n$ there are some differences when considering the total computation time. Indeed, for MED algorithms we do not compute $\mathcal{K}_{\text{inf}}^{\mathcal{F}}$ for the best empirical arm. However, this arm is expected to converge to the true best arm and to be sampled linearly. Hence, if there is only one best arm the costs presented in Table 3 will be paid only for $n = \mathcal{O}(\log(T))$. On the contrary, KL-UCB and NPTS will pay their cost for all arms, including the best empirical arm, so one of them satisfy $n = \Omega(T)$. The analysis of these algorithms suggest that this cost may not be necessary (see Remark 3 of [Riou and Honda, 2020]), nonetheless this is how they are typically implemented in the literature.*

## F.2 DSSAT distributions

The DSSAT problem was presented in Section 1 and the experimental results in the last Section 4. To give the reader the full picture of the problem, we plot in Figure 5 all the densities of the seven distributions.

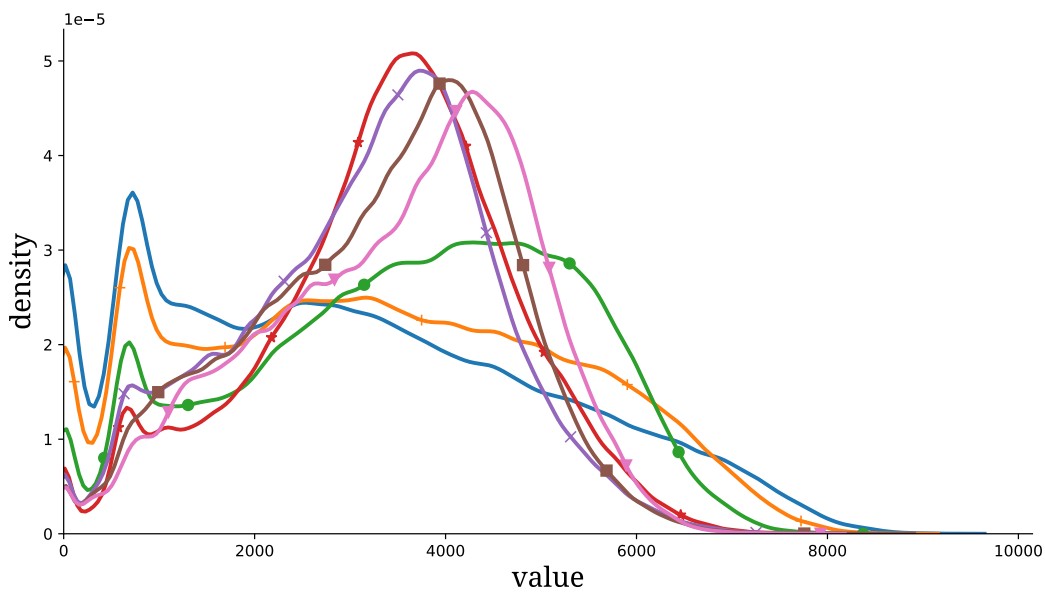

Figure 5: DSSAT distributions

This plot visually confirms the multimodality of all distributions and the difficulty to fit this problem into a parametric framework. We complete the Table 1 presented in the introduction by listing in Table 4 all ratios of the form $\frac{\mathcal{K}_{\inf}^{\mathcal{F}}}{\mathrm{kl}}$ and $\frac{\mathcal{K}_{\inf}^{\mathcal{F}}}{2\Delta^2}$, thus confirming the theoretical importance of using the true $\mathcal{K}_{\inf}^{\mathcal{F}}$ quantities over their usual relaxed versions.

| $\frac{\mathcal{K}_{\inf}^{\mathcal{F}}(F_k,\mu^\star)}{\mathrm{kl}(\mu_k,\mu^\star)}$ | 4.39 | 5.22 | 12.04 | 10.19 | 11.73 | 11.04 |
|---|---|---|---|---|---|---|
| $\frac{\mathcal{K}_{\inf}^{\mathcal{F}}(F_k,\mu^\star)}{2\Delta_k^2}$ | 4.72 | 5.55 | 12.73 | 10.87 | 12.44 | 11.63 |

Table 4: List of all ratios for the DSSAT bandit problem

To ground this theoretical analysis, we plot again the regret curves and average run time of tested algorithms (Figure 6) as well as Table 5 that compiles the average regret and average run time on the DSSAT experiment at the horizon 10 000. Reading this table, it is clear that FIMED improves the run time of IMED without compromising the regret and that OIMED improves the run time even further, close to the one of UCB, but at the cost of loosing a bit in the regret term. However, of the fastest methods (those in $\mathcal{O}(1)$ run time), OIMED clearly has the smallest regret. If the practitioner is willing to pay a bit more time complexity and space complexity, then our proposed FIMED seems to be the best algorithmic method.

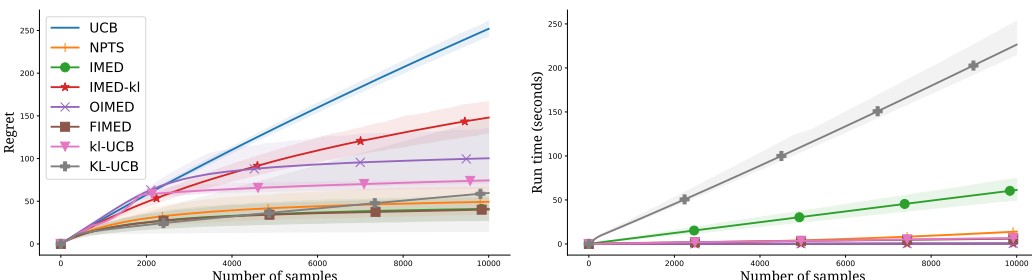

Figure 6: Average regret (left) and run time (right) of the algorithms on the DSSAT bandit problem

| Algorithm | UCB | NPTS | IMED | IMED-kl | OIMED | FIMED | kl-UCB | KL-UCB |
|---|---|---|---|---|---|---|---|---|
| Regret | 252 | 49 | 41 | 148 | 100 | 40 | 74 | 60 |
| Run time (sec.) | 0.38 | 14 | 61 | 0.51 | 0.47 | 6 | 6.8 | 226 |

Table 5: Average regret and run time at horizon 10 000 on DSSAT

### F.3    Additional bandit experiments

Here, we confirm the analysis of the previous section. First, we benchmark our FIMED and OIMED algorithms against IMED using the experimental setting of the original paper by Honda and Takemura [2015] in which the IMED algorithm is introduced. Then, we run two sets of experiments: one with Bernoulli distributions to complete our main experiment presented in Section 4, Figure 3, and another one with Beta distributions.

#### F.3.1    Original testbed of IMED

**Negative exponential**    We consider the negative exponential setting in which the law of an arm $k$ is such that $X_k = 1 - X_k'$ where $X_k'$ is an exponential random variable with parameter $\mu_k'$. In that case, the expected reward of arm $k$ is $\mu_k = 1 - \mu_k'$ and its support is in $(-\infty, 1]$. In Figure 7, we show simulation result of an experiment on a 5-armed negative exponential bandit problem

where, following the setting of Honda and Takemura [2015], $\{\mu'_k\}_k = \{1/5, 1/4, 1/3, 1/2, 1\}$, *i.e.* $\{\mu_k\}_k = \{4/5, 3/4, 2/3, 1/2, 0\}$.

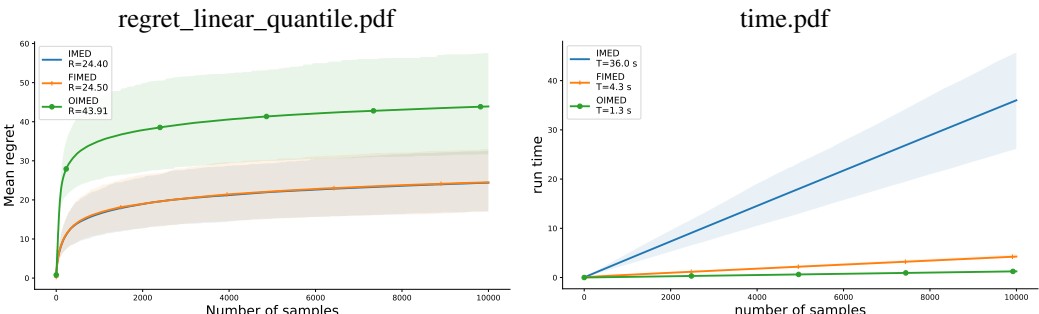

Figure 7: Average regret (left) and run time (right) of the algorithms on a $5$-arms negative exponential bandit problem with means $\{4/5, 3/4, 2/3, 1/2, 0\}$.

On this setting, the benefit of the `FIMED` algorithm over the original `IMED` can easily be read from Figure 7. `FIMED` enjoys the same regret as `IMED` while having an order of magnitude smaller computational time, confirming our theoretical analysis of `FIMED`. `OIMED` further reduce the computational time (and space) complexity, however at the cost of a larger experimental regret. The regret still exhibit the logarithmic shape in a reasonable experimental time and the choice is left to the practitioner whether to trade regret against time and space complexity or not.

**Truncated Gaussian** We continue benchmarking our algorithms using the original testbed of Honda and Takemura [2015], we run two experiments where arms are truncated Gaussian distributions. In the first experiment, Figure 8, the distributions are truncated on $[0, 1]$. In the second experiment, Figure 9, the distributions are truncated on $(-\infty, 1]$. In both experiments, we use the same set of Gaussian distributions, of means (before truncation) $\{0.6, 0.5, 0.5, 0.4, 0.4\}$ and variances $\{0.4, 0.2, 0.4, 0.2, 0.4\}$. After truncation on $[0, 1]$, the expected values are $\{0.519, 0.5, 0.5, 0.465, 0.481\}$, and after truncation on $(-\infty, 1]$, the expected values are $\{0.319, 0.390, 0.265, 0.320, 0.206\}$.

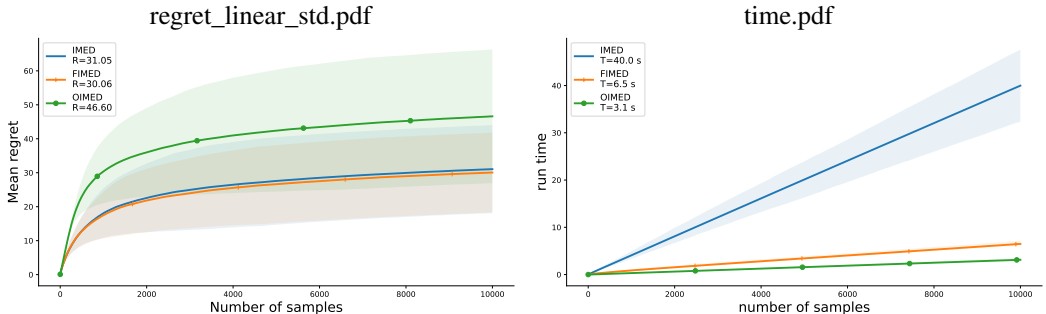

Figure 8: Average regret (left) and run time (right) of the algorithms on a $5$-arms $[0, 1]$-truncated Gaussian bandit problem with means $\{0.519, 0.5, 0.5, 0.465, 0.481\}$.

Quite accordingly to the intuition, the regret of all algorithms are better on the $[0, 1]$-truncated bandit problem than on the $(-\infty, 1]$-truncated one. Indeed, the variance of the former is smaller than the variance on the latter and an increased variance, all things being equal, tend to decrease the divergence between distributions, increasing the need for exploration. Both on Figure 8 and Figure 9, the regret pf `FIMED` is on par with that of `IMED` while being much faster. Following our analysis of running times, `OIMED` indeed is experimentally the fastest but does seem to trade this speed for a larger regret. However, one can see that in Figure 9 that the regret gap is smaller than in Figure 9, probably because the variance is larger. This variance interpretation is confirmed by the experiment on a Bernoulli bandit that is presented in Honda and Takemura [2015].

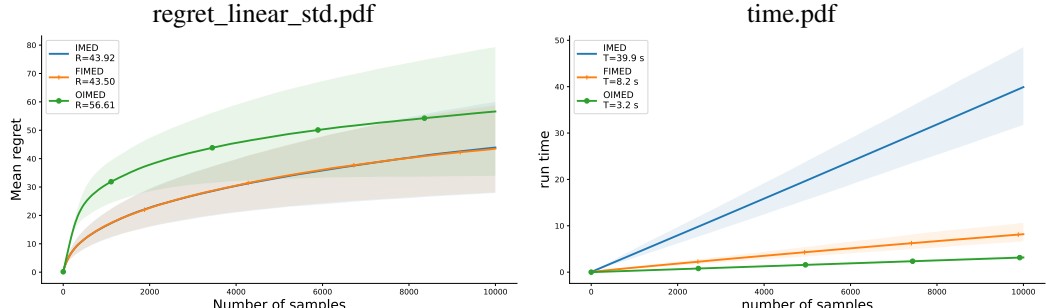

Figure 9: Average regret (left) and run time (right) of the algorithms on a 6-arms $(-\infty, 1]$-truncated Gaussian bandit problem with means $\{0.319, 0.390, 0.265, 0.320, 0.206\}$.

**Bernoulli** In Figure 10, we show simulation results for a 10-arms bandit with Bernoulli rewards with expected values $\{0.1, 0.05, 0.05, 0.05, 0.02, 0.02, 0.02, 0.01, 0.01, 0.01\}$. It can be seen that the performance gap between `OIMED` and `IMED` is larger than in any of the previous experiments. It can be attributed to the fact that all the rewards are all very close to zero and the variance of the distributions are small. However, the learning rate of the portfolio algorithm does not take those information into account, and it takes some time for the portfolio algorithm to converge, a time that can be seen in Figure 10 where the regret of `OIMED` is close to linear with no variance (pure exploration) for the 3 500 first time steps. In contrast, `IMED` and `OIMED`, which do not rely on a learning rate that is independent of the variance, have a much smaller regret.

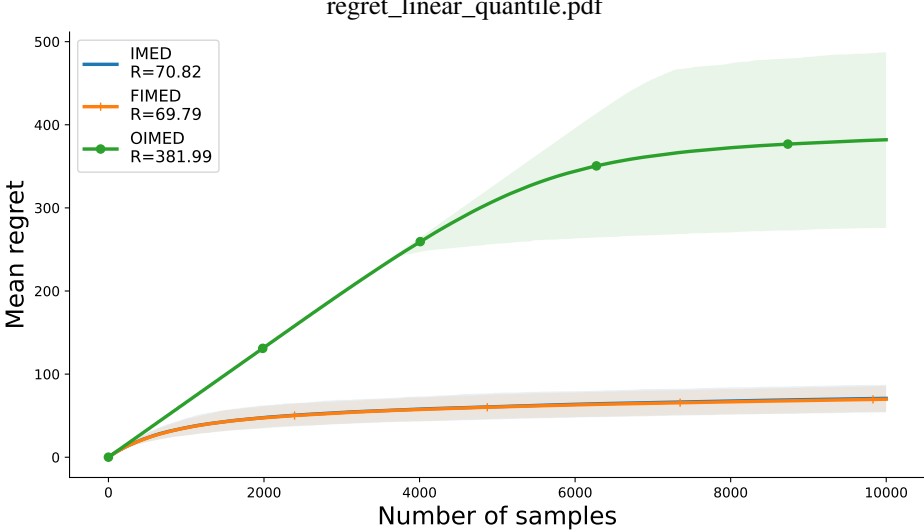

Figure 10: Average regret (left) and run time (right) of the algorithms on a 10-arms Bernoulli bandit problem with means $\{0.1, 0.05, 0.05, 0.05, 0.02, 0.02, 0.02, 0.01, 0.01, 0.01\}$.

Reproducing the experiments of the original `IMED` Honda and Takemura [2015], we confirmed that our theoretical analysis is experimentally supported. `FIMED` always seem to be on par with the original `IMED` algorithm while enjoying an order of magnitude faster numerical complexity, as proven in the analysis of running times. `OIMED` enjoys competitive regret, albeit experimentally larger than the original `IMED`. However, it has the smallest computational time complexity and its space complexity also is minimal. Therefore, it seems like there is currently a trade-off between achieving the smallest empirical computational complexity (and space complexity) and achieving the smallest cumulative regret. The original `IMED` lies at one extreme, `OIMED` at the other extreme and `FIMED` sits in-between. The algorithms studied in this paper allows the practitioner to choose the trade-off that suit her the best. Whether one can find an algorithm as fast as `OIMED` with experimental regret guarantees that exactly match `IMED` is a question for future work.

### F.3.2 Bernoulli bandits

In Section 4, we studied a Bernoulli bandit problem where all the means are close to 0.5 (Figure 3), identified as a "difficult" problem because the variance of the distributions in maximized among those supported in $[0, 1]$. Here, we explore two Bernoulli bandit problems, one where the means are located close to 1 (Figure 11, Table 6) and another hereafter where means are located close to 0 (Figure 12, Table 7). For each experiment, we plot the regret curves, the run time curves, as well as a table compiling the average regrets and run times of tested algorithms at the time horizon $T = 10000$.

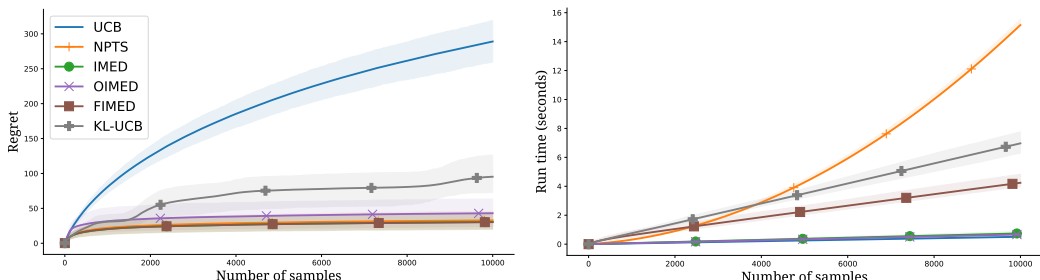

Figure 11: Average regret (left) and run time (right) of the algorithms on a 6-arms Bernoulli bandit problem with means $\{0.4, 0.6, 0.7, 0.85, 0.9, 0.95\}$.

| Algorithm | UCB | NPTS | IMED/IMED-kl | OIMED | FIMED | kl-UCB/KL-UCB |
|---|---|---|---|---|---|---|
| Regret | 289 | 33 | 30 | 43 | 30 | 95 |
| Run time (sec.) | 0.51 | 15 | 0.74 | 0.67 | 4.2 | 7 |

Table 6: Average regret and run time at horizon 10 000 on on a 6-arms Bernoulli bandit problem with means $\{0.4, 0.6, 0.7, 0.85, 0.9, 0.95\}$

We recall that for Bernoulli bandit, `IMED` and `kl-IMED` are the same algorithms (we assume that `IMED` is implemented as `kl-IMED` while only 0s and 1s are observed). For `FIMED` we could have done the same thing to improve its run time without changing its regret in order to emphasize that `FIMED` is always faster (or as fast as) `kl-UCB`, even when using the Bernoulli kl. While the regret and time values change from one experiment to another, the conclusions that can be drawn from them do not, especially when considering the confidence intervals represented by the 10%-90% quantiles. Experimentally, `OIMED` (or `OMED`, see Section F.4) should be the preferred $\mathcal{O}(1)$ method and `FIMED` (or `FMED`, see Section F.4) should be preferred if we really target an empirically optimal regret without compromising too much on the running time.

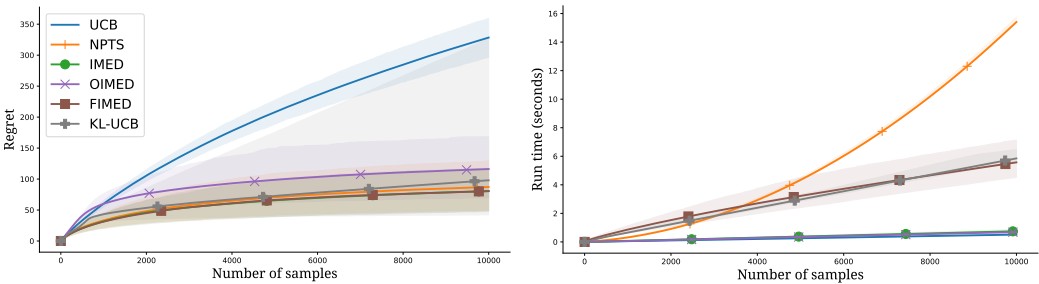

Figure 12: Average regret (left) and run time (right) of the algorithms on a 6-arms Bernoulli bandit problem with means $\{0.05, 0.1, 0.15, 0.2, 0.22, 0.25\}$

| Algorithm | UCB | NPTS | IMED/IMED-kl | OIMED | FIMED | kl-UCB/KL-UCB |
|---|---|---|---|---|---|---|
| Regret | 328 | 87 | 82 | 116 | 81 | 98 |
| Run time (sec.) | 0.51 | 15.4 | 0.75 | 0.69 | 5.6 | 5.9 |

Table 7: Average regret and run time at horizon 10 000 on on a 6-arms Bernoulli bandit problem with means $\{0.05, 0.1, 0.15, 0.2, 0.22, 0.25\}$

### F.3.3 Beta bandits

Beta distribution of a given mean can have different shapes. In particular, a Beta distribution can be close to a Bernoulli distribution (shape parameter close to zero) with most of the density located around 0 and 1, close to a Dirac distribution (shape parameter significantly larger than one) with most of the density located around the mean, and close to a truncated Gaussian distribution (shape parameter larger than one) with the characteristic bell shape distribution around the mean.

In Section 4, we studied a Bernoulli bandit problem where all the means are close to 0.5 (Figure 3). Here, using the same set of means, $\{0.3, 0.4, 0.45, 0.5, 0.52, 0.55\}$, we run two experiments, one on Beta distributions with an intermediate shape parameter of 5 (Figure 13) and another where Beta distributions have a shape parameter of 50 (Figure 15), hence with highly piked distributions. We do so to illustrate the effect of changing the shape of distributions without changing the set of means.

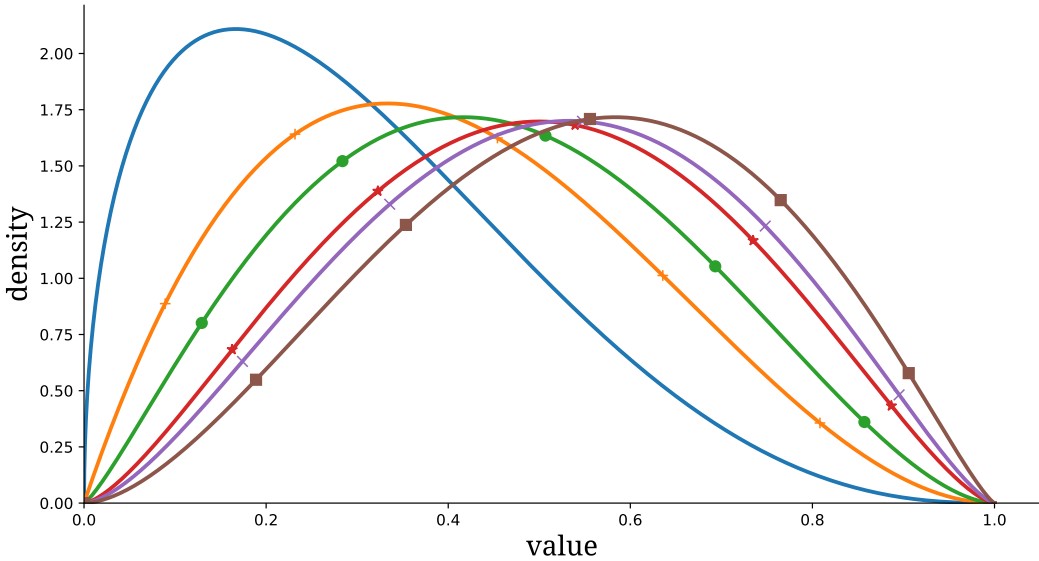

Figure 13: Beta bandit distributions with means $\{0.3, 0.4, 0.45, 0.5, 0.52, 0.55\}$ and shape parameter of 5

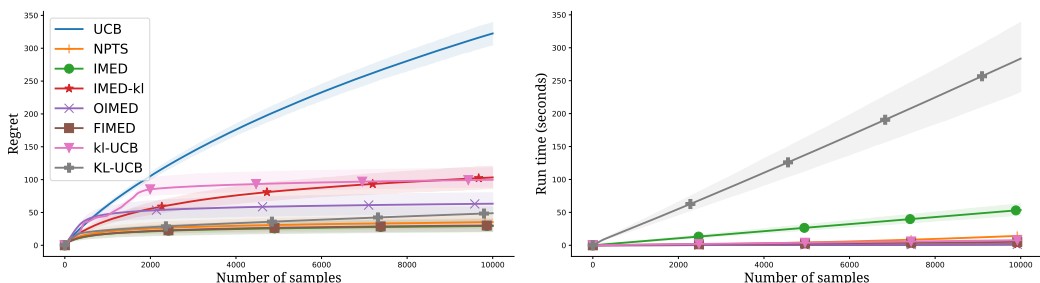

Figure 14: Average regret (left) and run time (right) of the algorithms on a 6-arms Beta bandit problem with means $\{0.3, 0.4, 0.45, 0.5, 0.52, 0.55\}$ and shape parameter of 5

| Algorithm | UCB | NPTS | IMED | IMED-kl | OIMED | FIMED | kl-UCB | KL-UCB |
|---|---|---|---|---|---|---|---|---|
| Regret | 322 | 35 | 29 | 103 | 63 | 30 | 100 | 49 |
| Run time (sec.) | 1.2 | 14.5 | 53.5 | 1.5 | 1.4 | 5.2 | 7.7 | 284 |

Table 8: Average regret and run time at horizon 10 000 on on a 6-arms Beta bandit problem with means $\{0.3, 0.4, 0.45, 0.5, 0.52, 0.55\}$ and shape parameter of 5

Comparing the regret curves of Figure 3, Figure 14, and Figure 16, it is clear that the Bernoulli distributions does induce the larger variance on the regret curves as we identified while the Beta distributions with the largest shape parameter (more concentrated around their means) induce the smallest variance on the regret curve. This, once again, confirms our theoretical findings. While the order of regret curves is globally preserved, one can see on Figure 3 that, except for UCB, the Bernoulli experiment makes all curves to be very similar, which is normal since, except for UCB, all algorithms are roughly solving the same problem (all sample are 0s and 1s). When dealing with Beta distributions, IMED is different than kl-IMED and KL-UCB is different than kl-UCB.

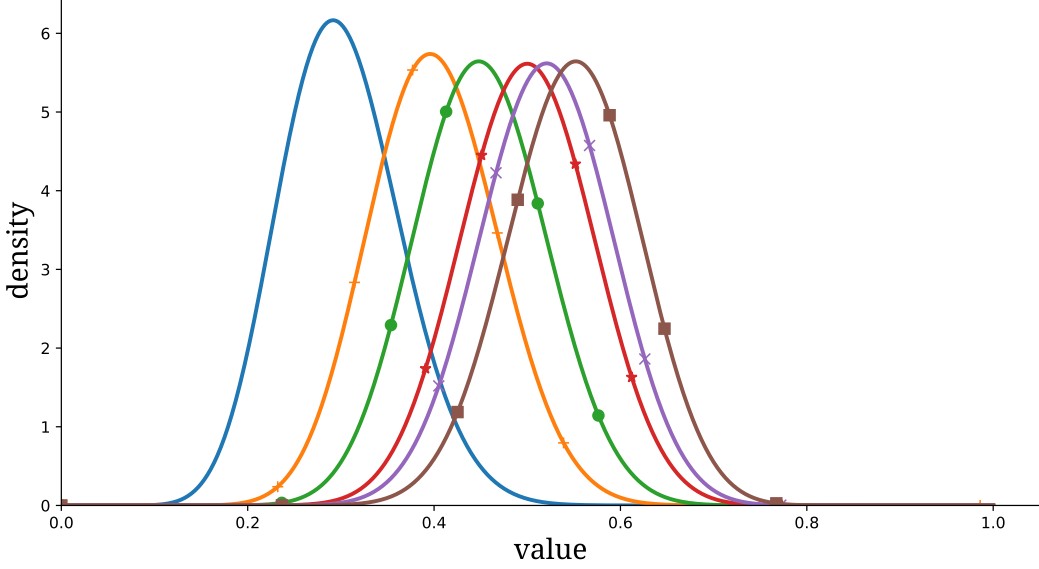

Figure 15: Beta bandit distributions with means $\{0.3, 0.4, 0.45, 0.5, 0.52, 0.55\}$ and shape parameter of 50

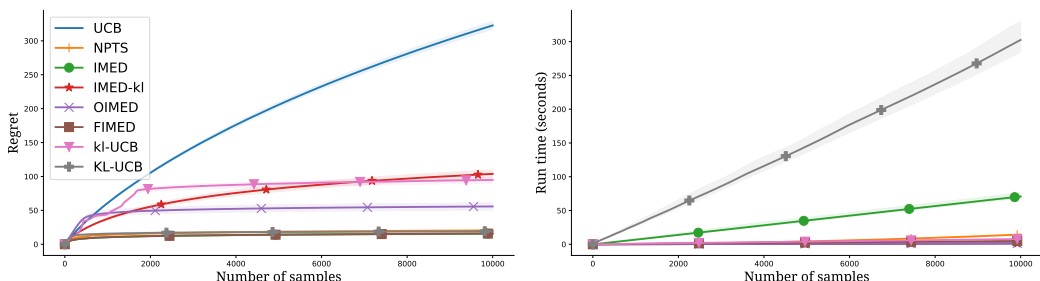

Figure 16: Average regret (left) and run time (right) of the algorithms on a 6-arms Beta bandit problem with means $\{0.3, 0.4, 0.45, 0.5, 0.52, 0.55\}$ and shape parameter of 50

| Algorithm | UCB | NPTS | IMED | IMED-kl | OIMED | FIMED | kl-UCB | KL-UCB |
|---|---|---|---|---|---|---|---|---|
| Regret | 323 | 20 | 15.9 | 103.9 | 55.8 | 15.8 | 95 | 19.2 |
| Run time (sec.) | 1.2 | 14.4 | 70.8 | 1.5 | 1.4 | 5 | 7.8 | 303 |

Table 9: Average regret and run time at horizon 10 000 on on a 6-arms Beta bandit problem with means $\{0.3, 0.4, 0.45, 0.5, 0.52, 0.55\}$ and shape parameter of 50

Those experiments confirm our numerical findings that `OIMED` (or `OMED`, see Section F.4) should be the preferred method with $\mathcal{O}(1)$ computation time, and `FIMED` (or `FMED`, see Section F.4), should be preferred if we really target an empirically optimal regret.

### F.4   Comparison of `MED` and `IMED` versions

In this paper, we derived algorithmic and theoretical results for modification of both `IMED` and `MED` algorithms. In this section, we compare `IMED`, its modifications `FIMED` and `OIMED`, `MED`, and its modifications `FMED` and `OMED`. The comparison is made on three experimental setting, the DSSAT bandit problem and Bernoulli bandit problem presented in the main experimental Section 4, as well as a Beta bandit problem with an intermediate shape parameter of 5 and centered means equal to the Bernoulli setting.

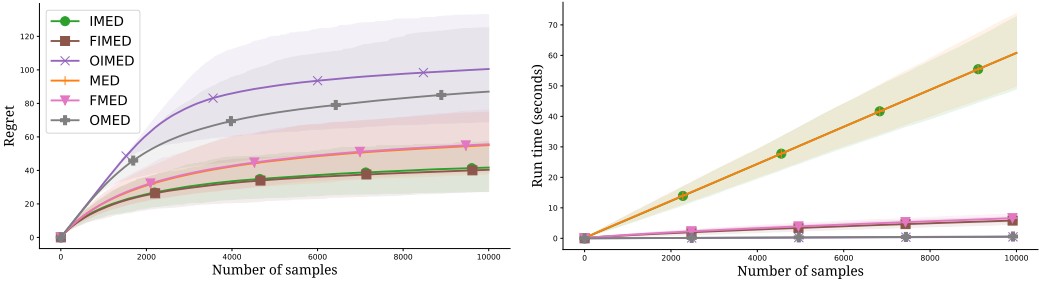

Figure 17: Average regret (left) and run time (right) of the algorithms on DSSAT

| Algorithm | IMED | FIMED | OIMED | MED | FMED | OMED |
|---|---|---|---|---|---|---|
| Regret | 41.7 | 40.4 | 100.5 | 55.1 | 55.6 | 87 |
| Run time (sec.) | 60.8 | 5.9 | 0.5 | 60.8 | 6.6 | 0.6 |

Table 10: Average regret and run time at horizon 10 000 of the algorithms on DSSAT

In Figure 17 and Table 10, we observe that `IMED` slightly overperforms `MED`. Consequently, `FIMED` also slightly overperforms `FMED` for the regret metric. Regarding the running time, the `MED` versions of algorithms are a bit slower than the `IMED` versions due to the numerical complexity of sampling, larger than that of computing the *argmin* of `IMED` indexes. Interestingly, `OMED` overperforms `OIMED` in terms of average regret performance. As we can see in Figure 18 and Table 11 presenting the results of the Bernoulli experiment, this is not a general rule.

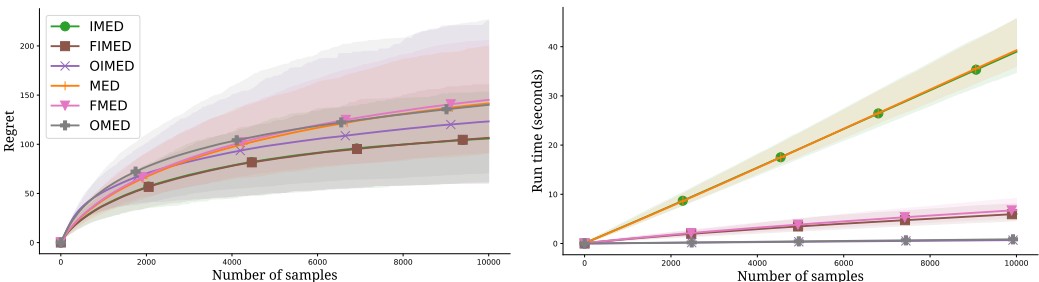

Figure 18: Average regret (left) and run time (right) of the algorithms on a 6-arms Bernoulli bandit problem with means $\{0.3, 0.4, 0.45, 0.5, 0.52, 0.55\}$

| Algorithm | IMED | FIMED | OIMED | MED | FMED | OMED |
|---|---|---|---|---|---|---|
| Regret | 105.8 | 106.6 | 123.3 | 141.5 | 145.2 | 140.1 |
| Run time (sec.) | 39 | 6 | 0.7 | 39.3 | 6.8 | 0.8 |

Table 11: Average regret and run time at horizon 10 000 of the algorithms on a 6-arms Bernoulli bandit problem with means $\{0.3, 0.4, 0.45, 0.5, 0.52, 0.55\}$

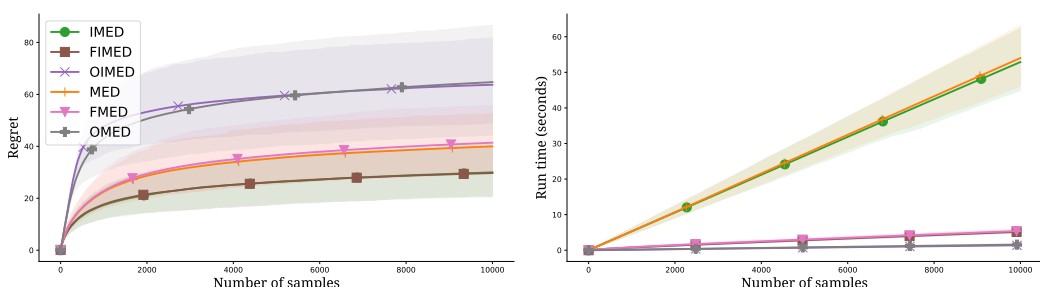

Figure 19: Average regret (left) and run time (right) of the algorithms on a 6-arms Beta bandit problem with means $\{0.3, 0.4, 0.45, 0.5, 0.52, 0.55\}$ and shape parameter of 5

| Algorithm | IMED | FIMED | OIMED | MED | FMED | OMED |
|---|---|---|---|---|---|---|
| Regret | 30 | 29.7 | 63.7 | 40 | 41.4 | 64.7 |
| Run time (sec.) | 53 | 5.2 | 1.4 | 54 | 5.4 | 1.6 |

Table 12: Average regret and run time at horizon 10 000 of the algorithms on a 6-arms Beta bandit problem with means $\{0.3, 0.4, 0.45, 0.5, 0.52, 0.55\}$ and shape parameter of 5

The experiment on Beta distribution, Figure 19 and Table 12, confirms that the regret loss incurred by the transition from `MED` to `OMED` is smaller than the one of transitioning from `IMED` to `OIMED`. It may

be because the sampling strategy of `MED` is better to handle the statistical approximation of portfolio algorithms.

### F.5   Stability of `OIMED` with respect to the learning rate

The portfolio algorithm, Soft Bayes, behind `OIMED` and `OMED` depends on a hyperparameter, $\eta$, the learning rate. A learning rate scheme is prescribed in Theorem 3 of [Orseau et al., 2017] with

$$\eta(n) = \sqrt{\frac{\log(2)}{4n}}$$

where $n$ is the number of collected samples. In section, we test the numerical stability of our `OIMED` algorithm with respect to the learning rate. To do so, we will replace this original $\eta$ by

$$\eta_r(n) = \sqrt{\frac{r\log(2)}{4n}}$$

where $r$ will range from 0.01 to 100. We illustrate the stability of `OIMED` on three bandit settings: the DSSAT bandit problem and Bernoulli problem that were introduced in the main Section 4 and a Beta bandit problem where all the means are centered around 0.5 and the same as in the Bernoulli bandit.

First, we test the stability of our `OIMED` algorithm on the DSSAT bandit problem that we already used in the main experimental Section 4 and Appendix F. On Figure F.5, the average regret ranges from 103.5 for the smallest values of the parameter $r$ to 88.5 for the largest value of $r$ with an intermediate regret of 99.5 for our original value $r = 1$. On this same plot, it is interesting to see that despite the large range of tested parameter size, the quantile tubes remain of roughly the same size with no evident bad behavior. While the effect of changing $r$ may seem important, in the range of tested parameter, the effect of changing $r$ would not have affected the ranking of algorithms in the original experiment, Figure 2 and Table 5. This experiment shows the stability of `OIMED` with respect to the learning rate, in the sense that a multiplicative constant does not seem to be able to dramatically deteriorate its performance.

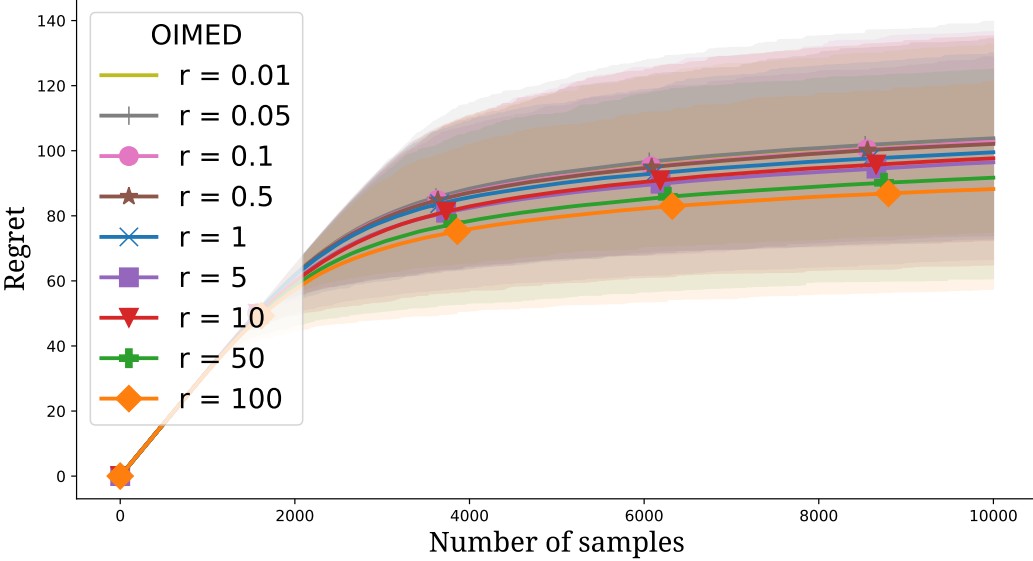

Figure 20: Average regret of the algorithms on DSSAT

Those findings are confirmed on a second experiment performed on a Bernoulli bandit problem with means $\{0.3, 0.4, 0.45, 0.5, 0.52, 0.55\}$. The regret changes from 107 to 124 when varying the parameter $r$ from 0.01 to 100, and, while the upper quantile is larger for small value of $r$, we still observe a logarithmic curve which confirms the stability of `OIMED`.

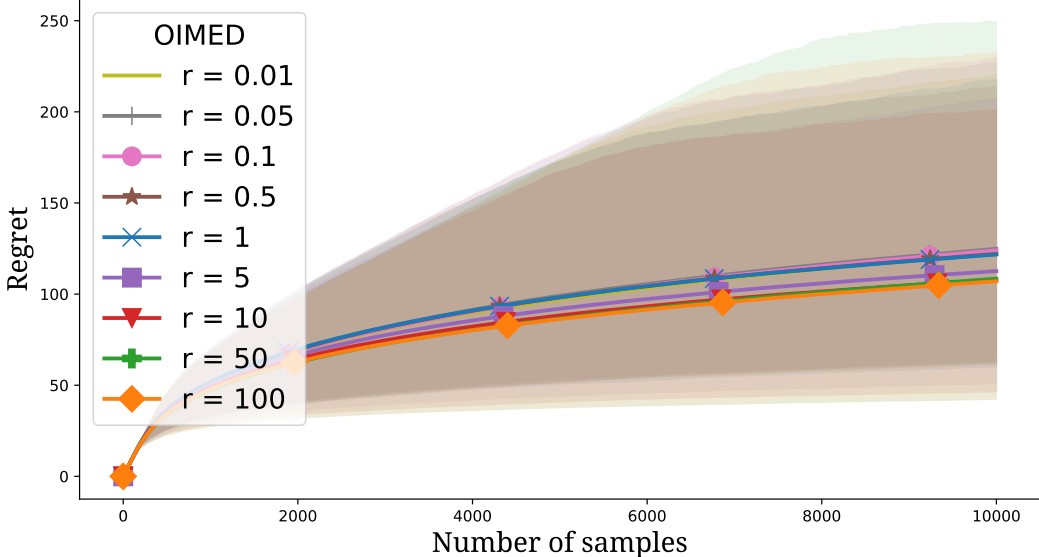

Figure 21: Average regret at horizon 10 000 of the algorithms on a 6-arms Bernoulli bandit problem with means $\{0.3, 0.4, 0.45, 0.5, 0.52, 0.55\}$

Finally, we check the effect of the learning rate on two Beta bandit problems where means are $\{0.3, 0.4, 0.45, 0.5, 0.52, 0.55\}$ and one problem has Beta distributions with shape parameter 5 and the other with shape parameter 50. Those two Beta bandit problems are presented in Appendix F.3.3.

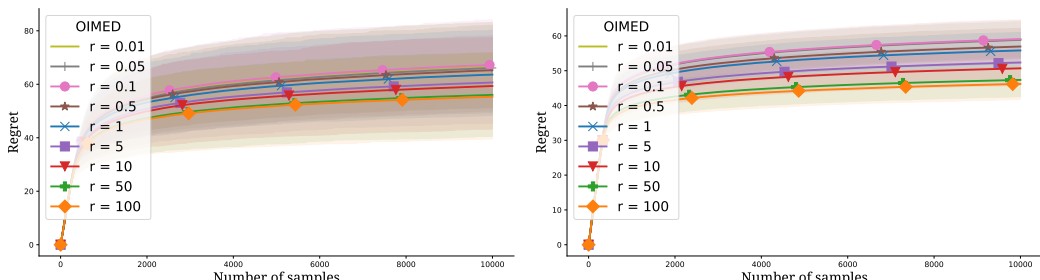

Figure 22: Average regret of the algorithms on two 6-arms Beta bandit problems with means $\{0.3, 0.4, 0.45, 0.5, 0.52, 0.55\}$ and shape parameter of 5 (left) and 50 (right)

When the variance of the distributions is smaller (large shape parameter), we can see that regret curves are better separated and the the bonus of choosing a larger learning rate parameter $r$ is amplified. When the variance is larger (small shape parameter), regret curves are closer to each other and more within each others quantile tubes. This last experiment confirms all our previous findings about the numerical stability of `OIMED`.

### F.6 `IMED` **with discretized rewards**

In Section 1 of this paper we introduced a known trick to reduce the time and memory complexities of `MED/IMED`. It consists in using algorithms designed for multinomial rewards, by using a discretization procedure on the collected rewards. We furthermore detailed this technique in Appendix F.1, and proved its inevitable sub-optimality for some problems. We end this section by experimenting with multinomial `IMED`, comparing the empirical performance and computation time of several instances using different grids. In each case, we fix a number of ticks $M$, and use the grid $\left\{0, \frac{1}{M-1}, \ldots, \frac{M-2}{M-1}, 1\right\}$ ($\{0, 1\}$ if $M = 2$). Our objective is to identify the number of ticks needed

to get a regret close to IMED, and the evolution of the computation time with the number of ticks. We first run an experiment on the DSSAT bandit problem, and then on a Beta bandit problem where means are close to zero.

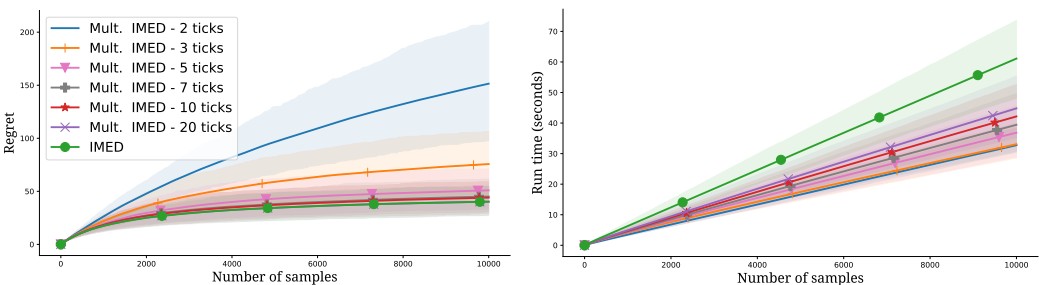

Figure 23: Average regret (left) and run time (right) of the algorithms on DSSAT

| Alg. (ticks) | IMED (2) | IMED (3) | IMED (5) | IMED (7) | IMED (10) | IMED (20) | IMED |
|---|---|---|---|---|---|---|---|
| Regret | 151.5 | 75.7 | 50.9 | 45.2 | 44.1 | 40.25 | 40.25 |
| Run time | 32.8 | 33.1 | 36.8 | 39.4 | 42.2 | 44.8 | 61.1 |

Table 13: Average regret and run time at horizon 10 000 of the algorithms on DSSAT

Unsurprisingly, the more ticks, the better the regret and the larger the running time, with IMED and kl-IMED providing the range for both metrics. For this problem, $M = 20$ (intervals of length $\approx 0.05$ between each tick) seems to be enough to make the algorithm using discretized rewards almost match the true IMED in terms of regret, while being $25\%$ faster.

We remark that the evolution of the time complexity per time step is not linear in the number of ticks. This is likely because it is actually dependent of the "effective" number of ticks used in memory, i.e. those that contain at least 1 samples. Getting at least 1 sample for each tick/arm takes some time, that depends on the shape of the distributions on the $[0, 1]$ support.

We run a final experiments on a Beta bandit problem that confirms our findings. IMED and kl-IMED again provide the range for both computation time and regret, and for this problem $M = 20$ is also the value from which IMED and discretized IMED start to match in terms of regret, while the latter is approximately $20\%$ faster.

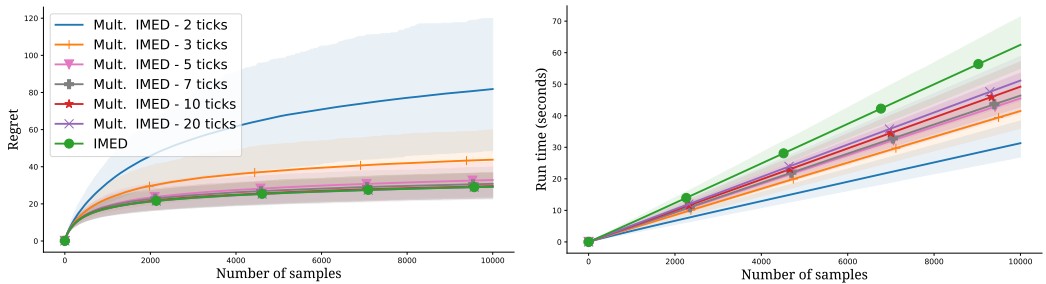

Figure 24: Average regret (left) and run time (right) of the algorithms on a 6-arms Beta bandit problem with means $\{0.05, 0.1, 0.15, 0.2, 0.22, 0.25\}$ and shape parameter of 5

| Alg. (ticks) | IMED (2) | IMED (3) | IMED (5) | IMED (7) | IMED (10) | IMED (20) | IMED |
|---|---|---|---|---|---|---|---|
| Regret | 81.8 | 43.8 | 32.9 | 30.8 | 29.6 | 29.2 | 29.2 |
| Run time | 31.3 | 41.5 | 45.5 | 46.4 | 49.2 | 51.2 | 62.5 |

Table 14: Average regret and run time at horizon 10 000 of the algorithms on a 6-arms Beta bandit problem with means $\{0.05, 0.1, 0.15, 0.2, 0.22, 0.25\}$ and shape parameter of 5

