# OpenReview forum: "Fast Asymptotically Optimal Algorithms for Non-Parametric Stochastic Bandits"
_NeurIPS.cc/2023/Conference — NeurIPS 2023 poster_

### Official Review · Reviewer_zxKa · 2023-06-24

**Soundness:** 3 good
**Presentation:** 3 good
**Contribution:** 3 good
**Rating:** 6
**Confidence:** 3

**Summary:**

This paper considers the classical regret minimization problem in the stochastic multi-armed bandit setting with arms having distributions with support bounded from above by a known constant B, and sometimes also lower bounded by b. Typically, all the known asymptotically optimal algorithms for regret-minimization problem in the bandit setup depend on computing an optimization problem involving the KL divergences. This is known to be computationally expensive in practice. The main object of study of this paper is K-inf. In this paper, authors develop approximations for empirical version of K-inf and study the performance of algorithms using these approximations, instead of exactly computing K-inf at each step.

The first of these approximations simply uses the convexity of K-inf in the second argument, while the second one derives from exploiting a connection between K-inf and online no-regret learning.

While using the first approximation in IMED and MED still leads to asymp. optimal algorithms, the second approximation requires assumptions on the no-regret algorithm used, and finding a no-regret algo. satisfying those assumptions remains open.

Numerical evidence supporting the theory is also provided.



**Strengths:**

I enjoyed reading the paper. It is generally well-written. While theory for asymptotically optimal algorithms for bandits have lately been developed in great generality, these optimal algorithms can be computationally demanding. These optimal algorithms all mostly depend on empirical K-inf. While the dual of this problem is a convex optimization problem, it is well known that its computation cost increases with number of samples (Cappé et al., 2013, Honda et al., 2015, Agrawal et al., 2021).

The paper studies an important practical problem of addressing the computational aspect of these optimal algorithms trying to avoid compromizing their optimality.


It is interesting to see the use of ideas from online portfolio selection algorithms in the bandit setting. As pointed by the authors, while this connection was previously observed in developing concentration inequality for K-inf by Agrawal et al. 2021, exploiting it to get fast algorithms is an interesting step in solving the computational aspect of K-inf, which, in my opinion, can be useful in other bandit settings as well.



**Weaknesses:**

1. The assumption on the no-regret algorithm in the second approach is a bit strange.

2. The paper can be a bit hard to read for someone new in the community.
    - Before introducing the fast MED algorithms in Section 2, it would be good to introduce the basic idea of
    MED algorithms (both IMED and MED) or how K-inf appears in them (or point the reader to relevant
    appendix where it is discussed, if space is a concern).
    - 1 line justification for the re-formulation of K-inf below line 126.
    - Details of why K-inf computations only happen for O(log T) times in line 112 of the paper. Though the
    reader is referred to Section 3 for discussion, I couldn't find a discussion on this there either.

3. In the introduction, the authors mention that KL-UCB requires several computations of K-inf per step, which I believe is not necessary - see, for example, Agrawal et. al, 2021 (Appendix D) where the index is expressed as a single 2D opt. problem. Similar approach can be used for KL-UCB algorithm in the setting considered here.

4. Minor comments:
    - Line 142: is --> are
    - Line 143: This is, for example, ...

**Questions:**

1. Can the ideas be extended beyond bounded distribution class? For example, to the moment-bounded class considered in Agrawal et al., 2021 or even the centered-moment bound class considered in Baudry et al., 2023? What would the challenges be in extending?

2. Could you clarify how you obtained the inequality after the data-processing inequality on Page 30, line 703?

3. In Figure 3, since we are looking at performance of OIMED and FIMED, it will be good to include IMED as well.

4. Also see weakness above.

---

> ### Author Rebuttal · Authors · 2023-08-04
>
> Thank you for your careful reading and your precise questions, as well as your helpful suggestions.
>
> We first answer to the points detailed in the weakness section of your review.
>
> * (1) In our bandit algorithm, a sub-linear upper bound on the portfolio regret avoids over-exploration while a sub-linear lower bound avoids under-exploration. The requirement for a lower bound is indeed unusual in the literature on portfolio selection, making this question an open problem for future work.
>
> * (2) We understand your point, we will follow your suggestions to make some parts of the paper easier for non-experts. We will point to Appendix A.2 where IMED (Algorithm 2) and MED (Algorithm 3) are detailed.
>
> * Still (2): KL-inf computations only happen when a (empirically) sub-optimal arm is pulled, so the upper bound on the number of pulls of Theorem 1 directly translates into an upper bound $\mathcal{O}(\log(T))$ on the number of KL-inf computations. We will clarify this in the revision.
>
> * (3) It is true that for our claim in the introduction we assume a naive implementation of KL-UCB using al linesearch on $\mu$, that may be optimized e.g. using the method of [4]. However, that method makes the cost of KL-UCB similar to the one of MED or IMED, not significantly better. We will change our description l. 57 to account for that.
>
> * (4) Thank you for noticing these typos.
>
> We now answers the questions asked in the dedicated section of your review.
>
> * (1) Extending our approach for the non-parametric families of distributions considered in [3] and [4] is indeed a natural direction for future works. Several challenges arise, making this extension non-trivial. First, the $\mathcal{K}_{\text{inf}}$ is not necessarily convex in its second argument in these settings (we exhibited convex then concave behavior in preliminary experiments for the centered family of [3]). As a consequence, the analysis of the pre-convergence term of FMED and FIMED would not be as simple as in the bounded case, or the algorithms may require modifications. Then, for the adaptation of OIMED and OMED the feasible set of the parameters $ (\lambda_1, \lambda_2)$ of the dual problem is of the form $\mathcal{S}= \{ (\lambda_1, \lambda_2) | \forall x \in \mathbb{R}, \; 1-\lambda_1(x-\mu) - \lambda_2 (B-h(|x|)) \geq 0 \}$, which is more difficult to map to the simplex than $\lambda \in \left[0, \frac{1}{B-\mu}\right]$ in the bounded case.
>
> * (2) p. 30, l. 703: we skipped the following steps, and will add them in the revision to improve clarity. Denote by $G_\star$ the distribution satisfying $$K_{\text{inf}}(F_k, \mu^\star) = \text{KL}(F_k, G_\star),$$
>
> and by $G_\star^{X_M}$ its discretized counterpart. Then, data processing inequality states that
> $$K_{\text{inf}}(F_k, \mu^\star) = \text{KL}(F_k, G_\star) \geq \text{KL}(F_k^{X_M}, G_\star^{X_M}),$$
>
> and we obtain l. 703 using that $\text{KL}(F_k^{X_M}, G_\star^{X_M})$ is itself larger than the KL-inf between $F_k^{X_M}$ and $\mu^\star$, by definition of this function.
>
> * (3) We will include IMED in Figure 3 in the revision. As in Figure 2, its curve is superposed with the one of FIMED. In all our experiments the difference between the two algorithms is negligible while FIMED is much faster, showing the strength of this approach.

---

> > ### Comment · Reviewer_zxKa · 2023-08-19
> > **Response to the rebuttal by authors**
> >
> > I thank the authors for their responses and acknowledge reading the entire thread of rebuttals. I am satisfied with the authors' responses and retain the score.

---

### Official Review · Reviewer_zZCp · 2023-07-05

**Soundness:** 3 good
**Presentation:** 3 good
**Contribution:** 3 good
**Rating:** 7
**Confidence:** 2

**Summary:**

In this paper they provide fast optimal algorithms for (non-parametric) stochastic bandits.
Specifically, they consider the MED family of algorithms that require the computation of $\mathcal{K}_{inf}$.
Based on this family of algorithms they construct new variants that require the computation of $\mathcal{K}_{inf}$ for one arm and approximate $\mathcal{K}_{inf}$ for other arms. Although, these variants (FMED, FIMED) achieve computational speedups they store all the rewards in memory. To address this issue they use portfolio algorithms.


**Strengths:**

- This work addresses the well known issue of optimal bandit algorithms being impractical due to high computational complexity.
I think this is an important step towards practical optimal bandit algorithms.
- Novel (algorithmic) use of portfolio algorithms to estimate $\mathcal{K}_{inf}$.
- Well written
- Experiments prove the applicability of the proposed algorithms.

**Weaknesses:**

Maybe you have to mention the regret assumption for the portfolio algorithms upfront, i.e. in section 2 (instead of the end of section 3).

**Questions:**

- What is kl in line 78? Please clarify.
- How coupled is your approach with the knowledge of $b$ and $B$ the lower and upper bounds for the distribution support?
In case we don't exactly know these bounds, does under- or overestimating this parameters lead to poor perfomance? Specifically how is perfomance affects?

---

> ### Author Rebuttal · Authors · 2023-08-04
>
> Thank you for your encouraging evaluation of our work and for your helpful questions and suggestions.
>
> * **Portfolio regret assumption** We agree that this crucial assumption should be mentioned sooner in the paper, we will follow your suggestion and introduce it already in Section 2.
>
> * **Questions -- $\text{kl}$:** It denotes the Kullback-Leibler divergence of Bernoulli distributions, $$\forall (p, q) \in [0,1]\times (0,1)\;, \\text{kl}(p,q)=p\log(p/q) + (1-p) \log((1-p)/(1-q)).$$
> We indeed forgot to define it, thank you for noticing.
>
> * **Questions -- knowledge of $[b,B]$:** The strength of our approach is that the performance of our algorithms does not require the knowledge of $b$, while approaches relying e.g. on re-scaling are very sensitive to it. It just needs to be finite for our analysis, and $B-b$ multiplies some of the second-order terms of our regret bounds (which would be the case with any other standard algorithm like UCB). On the other hand, the knowledge of $B$ is crucial, but this is true for any bandit algorithm working with bounded distributions. Over-estimation can lead to sub-optimality, but preserves the logarithmic regret. In fact, if the bound used by the algorithm is $B+\gamma$ for some $\gamma>0$ then it would be optimal if the true model was ``the distributions are supported on $[b, B+\gamma]$''. This is a broader family of distributions than the distributions supported in $[0,B]$, which can lead to sub-optimal (but logarithmic) regret. On the other hand, under-estimation may lead to linear regret. Hence, if the practitioner does not know $B$ it is always preferable to over-estimate it.

---

> > ### Comment · Reviewer_zZCp · 2023-08-18
> >
> > Thanks for your answers.

---

### Official Review · Reviewer_Nc1j · 2023-07-09

**Soundness:** 4 excellent
**Presentation:** 4 excellent
**Contribution:** 3 good
**Rating:** 5
**Confidence:** 4

**Summary:**

This paper consideres classic multi armed bandit problem. The focus is on developing asymptotically optimal algorithms with efficient computational and memory complexity. The results for the proposed algorithms are reported in Table 2. FMED and FIMED compute KL divergence for the armed that is pulled while using first order Taylor expansions  for the other arms. In addition an online portfolio selection algorithm is used to estimate the KL divergences that further improves the computational and memory performance.

**Strengths:**

The paper contributes to the bandit literature by improving the computational and memory requirement of asymptotically optimal algorithms with non parameteric distributions (where it is assumed that an upper bound on the rewards is known).

**Weaknesses:**

Although the results are interesting, they rely on known results from Honda and Takemura [2010] on approximating KL diveregnce and results on online portfolio selection. This limits the significance of the contribution of the paper.

**Questions:**

Could authors provide more details on analytical challenges beyond using the results from Honda and Takemura [2010]  and online portfolio selection. This seems important as this is mainly a theoretical paper.

**Limitations:**

The paper is mainly a theoretical paper, targeting the computational and memory requiremets of exisiting algorithmis.

---

> ### Author Rebuttal · Authors · 2023-08-04
>
> Thank you for your comment, in the following we hope to address your concern on the technical contribution of our paper. Though the results from Honda \& Takemura (2010) and the literature on portfolio selection are the starting point of our work, we believe that we solved several significant technical challenges to obtain our results. Hence, we respectfully disagree with the opinion that the technical aspect of the paper may limit the significance of our contribution, and would be happy to discuss more precise points during the discussion phase.
>
> **Outline of our main technical contributions:** For the analysis of FMED and FIMED we had to upper bound new terms involving the variation of the best empirical arms (Var-Best and Transition-Best in the proofs). We believe that this part (l. 479-496) is non-trivial. Furthermore, we point out that our new way to upper bound the term PRE-CV$_{\texttt{IMED}}$ is of independent interest, since it largely simplifies the original proof of Honda \& Takemura (2015).
>
> Then, for the analysis of OMED and OIMED we use existing results on portfolio selection only through the assumption on the portfolio regret. This provides only a small part of the result, and a significant challenge remains in upper bounding the regret due to the portfolio bias. Solving this challenge led us to propose the current form of OMED and OIMED as duel-based algorithms, and their analysis is quite non-standard. Thus, the proof of Theorem 2 (Appendix C) presented a significant technical challenge.

---

### Official Review · Reviewer_ba7Y · 2023-07-13

**Soundness:** 3 good
**Presentation:** 3 good
**Contribution:** 3 good
**Rating:** 5
**Confidence:** 3

**Summary:**

This paper studies non-parametric stochastic bandits. In particular, the author proposes algorithms named Fast (Indexed) Minimum Empirical Divergence (FMED, FIMED), and Online (Indexed) Minimum Empirical Divergence (OMED, OIMED). These algorithms are designed based on Minimum Empirical Divergence (MED). Regret guarantees comparable to that of MED is established for FMED, FIMED, OMED, and OIMED. In addition, these new algorithms have much better computational complexities. Numerical experiments have been conducted to support these claims.


**Strengths:**

This problem is well-motivated and the paper is clearly written and easy to follow. The paper also presents a complete story from motivation to algorithmic design to theoretical guarantees and empirical justifications.

**Weaknesses:**

I have listed a couple questions in the “Questions” part of this review.

**Questions:**

1. It would be great if the authors could elaborate on the result established by Theorem 1. For example, the bound is dependent on $\epsilon$—can $o_{\epsilon}$ be enumerated and this bound be optimized over $\epsilon$ to get a more interpretable bound?
2. In addition, it would be helpful to include a more detailed comparison with Honda and Takemura 2015 on both the proof techniques and the regret bound and computational complexity of the algorithms.
3. Section 4 provides nice empirical justifications on the superior performance and computational complexity of the proposed algorithms. Is it possible to replicate the experiments conducted in Honda and Takemura and compare those algorithms there?

---

> ### Author Rebuttal · Authors · 2023-08-04
>
> Thank you very much for your questions, that we hope to address in the following.
>
> * **Second-order terms** In the revision we will detail the scaling of the $o_{\epsilon}(.)$ term in Theorem 1, and include the following short discussion. In our proof, all its components are explicit, so the scaling in $\epsilon$ can be recovered: we obtain $\epsilon^{-6}$, that allows to get a sub-linear minimax bound of $\mathcal{O}(T^{5/6})$ if e.g. $\epsilon = \Delta_k/2$ is chosen, though not the optimal rate of $\mathcal{O}(\sqrt{T})$. We can make several remarks on this result. First, the same scaling is obtained for the vanilla MED (l. 517) with the proof techniques presented in [3]. Our fast implementations don't deteriorate that bound. This result is likely to be not tight, but improving it may require a much more involved analysis. A recent paper [2] established that MED is minimax optimal for Bernoulli distributions, but the analysis is very technical. Finally, to the best of our knowledge it remains open to prove that there exists an algorithm that is both minimax and asymptotically (instance) optimal in bounded bandits (i.e. with the true $\mathcal{K}_{\text{inf}}$ in the instance-dependent regret, not the $2\Delta^2$ or $\text{kl}$ approximation as for KL-MS [2]).
>
>
> * **Comparison with Honda \& Takemura (2015)** Comparing the steps of the analysis would be intricate since they follow a quite different path. Our analysis is more inspired by the recent works [1] and [3]. We can state that the regret bound of Honda \& Takemura contains a $\mathcal{O}(\Delta^{-10})$ term (proof of their Corollary 4), so our result is tighter for small $\Delta$. However their bound holds for a more general case than ours since semi-bounded supports are allowed ($b=-\infty$ with our notation).
>
> Regarding the experimental aspect, in our revision we will replicate the experiments of Honda \& Takemura and put the results in our Appendix F, already containing additional experiments.

---

### Official Review · Reviewer_gHyp · 2023-07-20

**Soundness:** 3 good
**Presentation:** 3 good
**Contribution:** 4 excellent
**Rating:** 6
**Confidence:** 4

**Summary:**

This papers studies asymptotically optimal algorithms for regret minimization in the multi-armed bandit setting with bounded reward distributions. The main contribution is an efficient approximation of the KL constraint of the lower bound, leading to faster algorithms with (asymptotic) regret guarantees. The proposed approach extends the MED / IMED by optimizing a lower bound on the KL, and using online learning algorithms to estimate the KL constraint. The claims are corroborated with numerical experiments on real-world data for crop management.

**Strengths:**

Asymptotically instance-optimal bandit algorithms are well-studied, however existing approaches are expensive to compute because any optimal approach needs to (approximately) solve a linear program with KL constraints. Developing more efficient and practical algorithms is an import problem and this paper makes significant progress towards this direction. Both the relaxation of the KL and the use of online algorithms to compute K_inf are interesting novel ideas.

The paper is generally well-written, however fairly technical and could use some more guidance/intuition for the reader in a few places (e.g. how is the loss of the online algorithm connect to K_inf? how is the bias of the portfolio algorithm controlled?)

Related work is adequately discussed. Note that the use of online algorithms for approximating/minmizing KL objectives in bandit setting also appears in related context, e.g. [1-3].

[1] Foster, Dylan, and Alexander Rakhlin. "Beyond ucb: Optimal and efficient contextual bandits with regression oracles." In International Conference on Machine Learning, pp. 3199-3210. PMLR, 2020.

[2] Foster, Dylan J., Sham M. Kakade, Jian Qian, and Alexander Rakhlin. "The statistical complexity of interactive decision making." arXiv preprint arXiv:2112.13487 (2021).

[3] Kirschner, Johannes, Tor Lattimore, Claire Vernade, and Csaba Szepesvári. "Asymptotically optimal information-directed sampling." In Conference on Learning Theory, pp. 2777-2821. PMLR, 2021.

**Weaknesses:**

The approach appears to be tailored to bound reward distributions, whereas in general, asymptotically optimal algorithms are known for a much more general class of algorithms and reward distributions. It is unclear if the proposed techniques can be extend beyond the current setting. It would be great if the authors could comment on this point.

As a minor comment, while I appreciate the use of real-world data, the simulation of the crop-management tasks is run over 10000 rounds. Is this realistic in this type of applications? Do the algorithms perform well over much smaller horizons?

Can anything be said about finite-time or worst-case performance of the algorithms?



**Questions:**

* The main result (Theorem 1) uses small-o-notation to suppress the lower-order terms. Are these terms controlled in the analysis? Could you provide a more detailed discussion of the finite-time performance of the algorithm, both from a theoretical and empirical point of view?

**Limitations:**

* The online algorithm used to approximate the KL is assumed to have a bound on the absolute value of the regret. This is in rather unusual requirement and strongly limits the choice of online learning algorithms, as the authors discuss at the end of section 3.
* The approach requires an upper bound on the best reward, mu_max < B. It looks like this can easily be satisfied by replacing B by B + eps for small eps > 0. Does the bound depend on the gap B - mu_max? Is knowing B/mu_max a strong assumption?

I also suggest to add a more detailed explanation/clarify for the following points:

* eq below line 126: why do we get this equality? (How) Is N_k(t) defined?
* line 164-168: The dueling procedure is hard to understand, i.e. what is a "greedy duel"?

---

> ### Author Rebuttal · Authors · 2023-08-04
>
> Thank you for your careful reading and for proposing several ways to further improve the clarity of our paper. In the following we answer to the points raised in your review, in order. Please feel free to ask for any further clarification during the discussion phase.
>
> * **Remarks in the Strengths section:** We will move  Equation (13) in Section 2.2 in order to provide a clear summary of the link between $n \mathcal{K}_{\text{inf}}(F_n, \mu)$ and the quantities of the portfolio selection algorithm in this section.
> Also, thank you for pointing us to additional related works. We will add them to our literature review.
>
> * **Extension beyond bounded distributions:** Our approach is indeed tailored for bounded distributions, but we believe that it can be extended to other settings. Please note that in several usual settings MED/IMED are in fact easy to run and memoryless, so there is no incentive to modify their vanilla implementation: parametric families of distributions [3] (single parameter exponential families, Gaussian distributions), or Sub-Gaussian distributions (Maillard Sampling, [1]). However, as pointed out by reviewer zxKa, our approach may be interesting for some other non-parametric models presented in [3] (in which MED and TS approaches for these settings are analyzed) and [4], with families of distributions characterized by a moment condition of the form $\mathbb{E}[h(|X|)] \leq B$ for some known convex function $h$ and constant $B$. In such setting, $\mathcal{K}_{\text{inf}}$ can be expressed as the solution of an optimization problem, very similarly to the bounded case. There is hope that we can adapt our algorithms to these settings, but this is not straightforward (see our response to reviewer zxKa). This is a direction that we would like to consider in a future work.
>
> * **Real-word data:** We chose the crop yield dataset because it was available and provided realistic examples of bounded distributions for which optimal algorithms are largely superior to sub-optimal ones using $\text{kl}$ or $2\Delta^2$ as a proxy for $\mathcal{K}_{\text{inf}}$. We agree that obtaining $10^4$ points is unrealistic in that application and that our considerations of computational cost could have been better illustrated on another setting.
>
> * **Finite-time/worst-case performance/$o$ in Theorem 1:** In the revision we will detail the scaling of the $o_{\epsilon}(.)$ term in Theorem 1, and include the following short discussion. In our proof, all its components are explicit, so the scaling in $\epsilon$ can be recovered: we obtain $\epsilon^{-6}$, that allows to get a sub-linear minimax bound of $\mathcal{O}(T^{5/6})$ if e.g. $\epsilon = \Delta_k/2$ is chosen, though not the optimal rate of $\mathcal{O}(\sqrt{T})$. We can make several remarks on this result. First, the same scaling is obtained for the vanilla MED (l. 517) with the proof techniques presented in [3]. Our fast implementations don't deteriorate that bound. This result is likely to be not tight, but improving it may require a much more involved analysis. A recent paper [2] established that MED is minimax optimal for Bernoulli distributions, but the analysis is very technical. Finally, to the best of our knowledge it remains open to prove that there exists an algorithm that is both minimax and asymptotically (instance) optimal in bounded bandits (i.e. with the true $\mathcal{K}_{\text{inf}}$ in the instance-dependent regret, not the $2\Delta^2$ or $\text{kl}$ approximation as for KL-MS [2]).
>
> * **Upper bound on the best mean:** The upper bound $\mu_{\text{max}}< B$ is necessary so that the bias of the portfolio algorithm does not become infinite, and some of the second-order terms of the regret scale in $\frac{1}{B-\mu_{\text{max}}}$. We first highlight that all existing optimal algorithms need to know the upper bound $B$, so this assumption is standard. On the other hand, deciding if knowing $\mu_{\text{max}}<B$ is a strong assumption or not boils down to determining if our bandit problem may contain distributions that are highly concentrated around $B$. In our example in agriculture this would be very unrealistic, and it is reasonable to expect that experts may provide quite well separated estimates of $B$ ($\approx$ highest ever recorded yield) and $\mu_{\text{max}}$ (how good can the yield be on average knowing that there will inevitably be poor seasons). In absence of such expert knowledge, the solution of artificially increasing the upper bound outside of the support is indeed valid.
>
> * **Points to clarify** $N_k(t)$ is defined l. 16 as the number of pulls of arm $k$, and l. 126 is direct from the definition of  as the KL-inf as an expectation over the empirical distribution (Eq. (4)). We use ``greedy duels'' as a convenient term to call duels where the algorithm chooses (without sampling) the arm with maximum empirical average (that would be the output of the greedy algorithm). These duels are necessary to guarantee a large enough sample size for the leader when updates of the online estimate of $\mathcal{K}_{\text{inf}}$ are performed.

---

> > ### Comment · Reviewer_gHyp · 2023-08-14
> >
> > I would like to thank the authors for the clarifying comments.

---

### Author Rebuttal · Authors · 2023-08-04

We thank the reviewers for their insightful comments on the paper, and appreciate their overall positive feedback.
Following the suggestions of all reviewers, our revision will mainly serve the purpose of providing additional intuitions to some technical parts of the paper. We also thank the reviewers for pointing some typos and specific points that required clarifications.

We provide a detailed response under each individual review, and would be delighted to provide further insights if needed during the discussion phase.

In our responses we refer to the following references:

[1] J. Bian and K. Jun. Maillard sampling. Boltzmann exploration done optimally. In Proceedings of the 25th International Conference on Artificial Intelligence and Statistics, AISTATS
2022, 2022.

[2] H. Qin, K. Jun, C. Zhang. Kullback-Leibler Maillard Sampling for Multi-armed Bandits with Bounded Rewards, 2023.

[3] D. Baudry, K. Suzuki, and J. Honda. A general recipe for the analysis of randomized multi-armed bandit algorithms, 2023.

[4] S. Agrawal, S. Juneja, and W. M. Koolen. Regret minimization in heavy-tailed bandits. In Conference
on Learning Theory, COLT 2021, 15-19 August 2021, Boulder, Colorado, USA, 2021.

---

### Decision · Program_Chairs · 2023-09-21

**Decision:**

Accept (poster)

**Comment:**

This paper makes a solid contribution for the nonparametric multi-armed bandit problem (or, noise-adaptive multi-armed bandits). The improvements on the computational efficiency is meaningful, though the guarantee on OMED/OIMED comes with an assumption.